# Taperin bundles F-actin at stereocilia pivot points enabling optimal lifelong mechanosensitivity

Inna A. Belyantseva[1]*, Chang Liu[2]*, Abigail K. Dragich[3], Takushi Miyoshi[1,5], Sayaka Inagaki[1], Ayesha Imtiaz[1], Risa Tona[1], Karen Sofia Zuluaga-Osorio[3,4], Shadan Hadi[3], Elizabeth Wilson[1], Eva Morozko[1], Rafal Olszewski[6], Rizwan Yousaf[1], Yuliya Sokolova[7], Gavin P. Riordan[1], S. Andrew Aston[8], Atteeq U. Rehman[1], Cristina Fenollar Ferrer[1], Jan Wisniewski[9], Shoujun Gu[6], Gowri Nayak[10], Richard J. Goodyear[10], Jinan Li[2], Jocelyn F. Krey[11,12], Talah Wafa[13], Rabia Faridi[1], Samuel Mawuli Adadey[1], Meghan Drummond[1], Benjamin Perrin[14], Dennis C. Winkler[7], Matthew F. Starost[15], Hui Cheng[8], Tracy Fitzgerald[13], Guy P. Richardson[10], Lijin Dong[16], Peter G. Barr-Gillespie[11,12], Michael Hoa[6], Gregory I. Frolenkov[3,17], Thomas B. Friedman[1], and Bo Zhao[2]

**Stereocilia are rod-like mechanosensory projections consisting of unidirectionally oriented actin filaments that extend into the inner ear hair cell cytoskeleton, forming dense rootlets. Taperin (TPRN) localizes to the narrowed-down base of stereocilia, where they pivot in response to sound and gravity. We show that TPRN-deficient mice have progressive deafness characterized by gradual asynchronous retraction and fusion of outer and inner hair cell stereocilia, followed by synaptic abnormalities. Stereocilia that lack TPRN develop warped rootlets with gradual loss of TRIOBP-5 and ANKRD24 from mechanosensory rows starting postnatally. In contrast, TPRN overexpression causes excessive F-actin bundling, extra rows, and over-elongation of stereocilia during development. Purified full-length mouse TPRN cross-links F-actin into bendable bundles reflecting in vivo data. This F-actin–bundling ability is attributed to the TPRN N-terminal region. TPRN interacts with the membrane receptor PTPRQ, connecting the F-actin core to the plasma membrane, stabilizing stereocilia. Thus, TPRN is a specialized F-actin bundler strategically located to augment stereocilia rootlet formation and their pivot point flexibility for sustained sound-induced deflections.**

## Introduction

Mechanosensory stereocilia on the apical surface of auditory hair cells are organized in rows of increasing height (Fig. 1 A). Tip links interconnect stereocilia of neighboring rows and gate the mechanoelectrical transduction (MET) channels that are located at the tips of shorter stereocilia and can sense sub-nanometer movements (Beurg et al., 2009; Fettiplace and Hackney, 2006; Pickles et al., 1984). Additional links between stereocilia sustain the bundle (Goodyear et al., 2005). When deflected by sound waves, stiff stereocilia pivot at their tapered

bases (Fettiplace and Hackney, 2006; Karavitaki and Corey, 2010) (Fig. 1 A), initiating MET (Hudspeth and Jacobs, 1979). Each stereocilium has a rigid paracrystalline core with hundreds to thousands of unidirectionally oriented and cross-linked actin filaments with barbed ends located at the tip and pointed ends at the base of a stereocilium (Tilney et al., 1980; Tilney and Tilney, 1988). Near the base of each stereocilium, some of the peripheral actin filaments are terminated and some are gathered together forming a tapered shape toward an insertion point that is

[1]Laboratory of Molecular Genetics, National Institute on Deafness and Other Communication Disorders, National Institutes of Health, Bethesda, MD, USA;   [2]Department of Otolaryngology-Head and Neck Surgery, Indiana University School of Medicine, Indianapolis, IN, USA;   [3]Department of Physiology, University of Kentucky, College of Medicine, Lexington, KY, USA;   [4]EIA University (Antioquia School of Engineering) and CES University, Medellín, Colombia;   [5]Division of Molecular and Integrative Physiology, Department of Biomedical Sciences, Southern Illinois University School of Medicine, Carbondale, IL, USA;   [6]Auditory Development and Restoration Program, National Institute on Deafness and Other Communication Disorders, NIH, Bethesda, MD, USA;   [7]Advanced Imaging Core, National Institute on Deafness and Other Communication Disorders, NIH, Bethesda, MD, USA;   [8]Data Science Core, National Institute on Deafness and Other Communication Disorders, NIH, Bethesda, MD, USA;   [9]Confocal Microscopy and Digital Imaging Facility, Experimental Immunology Branch, NCI, NIH, Bethesda, MD, USA;   [10]School of Life Sciences, University of Sussex, Brighton, UK;   [11]Oregon Hearing Research Center, Oregon Health & Science University, Portland, OR, USA;   [12]Vollum Institute, Oregon Health & Science University, Portland, OR, USA;   [13]Mouse Auditory Testing Core, National Institute on Deafness and Other Communication Disorders, NIH, Bethesda, MD, USA;   [14]Department of Biology, Indiana University Indianapolis, Indianapolis, IN, USA;   [15]Division of Veterinary Resources, Office of Research Services, NIH, Bethesda, MD, USA;   [16]Genetic Engineering Core, National Eye Institute, NIH, Bethesda, MD, USA;   [17]Department of Otolaryngology - Head and Neck Surgery, University of Kentucky, College of Medicine, Lexington, KY, USA.

Correspondence to Bo Zhao: zhaozb@iu.edu;   Thomas B. Friedman: friedman@nidcd.nih.gov;   Gregory Frolenkov: gregory.frolenkov@uky.edu

*I.A. Belyantseva and C. Liu share co-first authors.

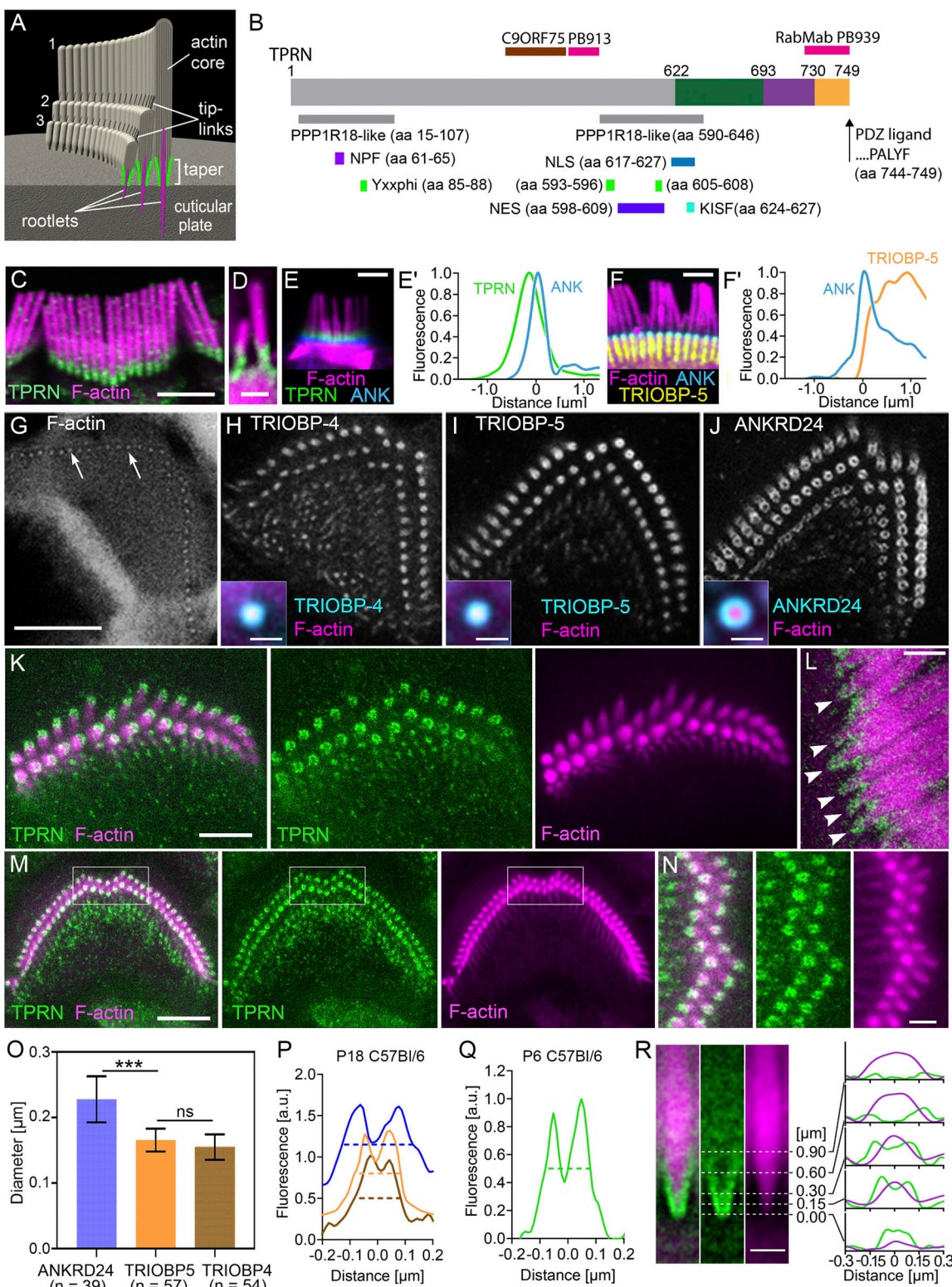

**Figure 1. TPRN protein domains, motifs, antibody epitopes, and localization in mouse cochlear hair cells. (A)** Illustration of a mechanosensory stereocilia bundle on the apical surface of a hair cell. The three rows of actin-based stereocilia are interconnected by tip links and anchored to the hair cell body by F-actin rootlets (magenta). The tapered region of stereocilia is indicated by a bracket and outlined by green lines. **(B)** Schematic of mouse TPRN protein encoded by four exons indicated by different colors and number of aa encoded by each exon. Below the TPRN schematic, colored bars indicate predicted domains using ELM (http://elm.eu.org) unless otherwise indicated. The gray bars indicate two small regions of aa sequence percentage identity to phostensin (aa 15–107 and 590–646). Purple bar is an EH ligand (aa 61–65) containing an NPF motif, the light green bars show three Y-based endocytic sorting motifs (Yxxphi) (aa 85–88, 593–596, and 605–608), blue bar is a NLS (aa 617–627, https://nls-mapper.iab.keio.ac.jp), light blue bar is a PP1-binding site with KISF motif (aa 624–627), violet bar is a nuclear export signal, NES (aa 598–609), and an arrow indicates a PDZ ligand (aa 744–749). Locations of epitopes of custom

rabbit polyclonal anti-TPRN antibodies (red bars above the schematic) PB913 and PB939 and a RabmAb against the same epitope to raise antiserum PB939. The brown bar—location of epitope of the commercial anti-TPRN (C9ORF75) antibody (HPA020899, RRID:AB_1845835; MilliporeSigma). **(C)** Confocal microscopy image showing TPRN localization (green) at bases of stereocilia spanning the taper region of all rows. **(D)** Stereocilia side view image. Unless otherwise stated, rhodamine-phalloidin (magenta) was used to counterstain F-actin in most panels. **(E)** Localization of TPRN (green) in relation to ANKRD24 (blue) in P20 WT OHC stereocilia. **(F)** Localization of ANKRD24 (blue) in relation to its interacting partner TRIOBP-5 (yellow) in P14 WT IHCs. **(E′–F′)** Line graphs of normalized fluorescence intensity for each protein measured from the stereocilia pivot point (0) to 1-µm down (1) and 1-µm up (−1) the stereocilium. ANKRD24 (cyan) concentrates at the pivot point of every stereocilium (magenta). A fainter signal of ANKRD24 (F′) distributes along the rootlet portion highlighted by TRIOBP-5 in (F). **(G–J)** STED super-resolution images at the pivot points of OHC stereocilia in P18 C57BL/6J mouse revealed rootlet insertions into cuticular plate stained with Star Red phalloidin (G) and surrounding ring patterns of TRIOBP-4 (H), TRIOBP-5 (I), and ANKRD24 (J). Inserts in H–J show mean intensity projection of aligned cross sections through the pivot point of row 1 OHC stereocilia double stained with F-actin (phalloidin, magenta) and TRIOBP-4 (*n* = 25), TRIOBP-5 (*n* = 52), and ANKRD24 (*n* = 64), correspondingly (turquoise). **(K)** A similar ring pattern at P6 stereocilia taper region just above the apical surface of IHC is revealed in STED image by a secondary nanobody together with anti-TPRN antibody (green). **(L)** Image along the longitudinal axis of P6 IHC stereocilia showing a funnel-like pattern (white arrowheads) of TPRN staining (green) at the taper region. **(M)** STED image of P6 OHC showing similar TPRN ring pattern (green). **(N)** Enlarged images of a rectangular area depicted in M of individual OHC stereocilia outlined by TPRN rings (green). **(O)** Diameters of ring staining for ANKRD24, TRIOBP-4, and TRIOBP-5 in the longest row (Row 1) stereocilia of OHCs. Number of stereocilia/cells: ANKRD24, *n* = 39/3; TRIOBP-5, *n* = 57/3; TRIOBP-4, *n* = 54/4. Asterisks show statistical significance of the differences between proteins: overall one-way ANOVA (P < 0.0001) followed by Tukey's post hoc comparison test (***P < 0.0001). The graph compares the average ring diameters for each protein using one-way ANOVA (P < 0.0001) followed by Tukey's post hoc comparison test (***P < 0.0001). **(P)** Examples of intensity profiles along a line through the center of a stereocilium rootlet cross section revealed by F-actin staining, which were used to determine the diameters of TRIOBP-4, TRIOBP-5, and ANKRD24 staining patterns in panel O. **(Q)** Similar plot of TPRN fluorescence intensity also confirms the ring appearance of TPRN staining in P6 stereocilia cross sections. **(R)** Left, mean intensity projection of 20 aligned longitudinal sections through the base of row 1 IHC stereocilia stained with F-actin (magenta) and TPRN (green). Right, intensity profiles of F-actin and TPRN staining across stereocilium at different distances from the pivot point (0). Images in C-E and F were obtained using Zeiss LSM880 Airyscan confocal system and in G–J using a FACILITY microscope system (Abberior Inc). Inserts in H–J and images in K–N and R were obtained using a STEDYCON super-resolution system (Abberior Inc) on an Eclipse Ti2 microscope (Nikon). Scale bars are 2 µm in C–J, 1 µm in K–N, 500 nm in L and R, and 200 nm in H–J inserts. Scale bar in G applies to G–J.

smaller in diameter than the diameter of a stereocilium (Furness et al., 2008; Song et al., 2020) (Fig. 1 A). Actin filaments descending through the insertion point are tightly packed together and penetrate the actin-rich cuticular plate meshwork to form a rootlet that anchors a stereocilium in the cell body (Fig. 1 A) (Katsuno et al., 2019; Kitajiri et al., 2010; Krey et al., 2022; Pacentine et al., 2020).

Rootlets occupy most of the space at stereocilia pivots, determine hair bundle mechanical stiffness and, hence, sensitivity to sound-induced forces (Furness et al., 2008; Kitajiri et al., 2010; Pacentine et al., 2020). To ensure the resilience of stereocilia for lifelong sound-induced deflections, rootlets must be both flexible and durable. Acoustic overstimulation can lead to mechanical damage of stereocilia F-actin at pivot points (Liberman, 1987; Pataky et al., 2004; Tilney et al., 1982). Several proteins localize to the taper region, including ANKRD24, GRXCR2, CLIC5, RDX, PTPRQ, TRIOBP, and taperin (TPRN) (Krey et al., 2022; Pacentine et al., 2020). However, their exact roles in optimizing mechanical properties and durability of stereocilia are unknown.

We reported previously that stereocilia that lack F-actin bundlers TRIOBP-4 and TRIOBP-5 do not form rootlets, cannot withstand repetitive deflections, and degenerate shortly after the onset of hearing (Kitajiri et al., 2010). TRIOBP-4 is responsible for the unique tight packing of F-actin along the entire length of rootlets (Katsuno et al., 2019; Kitajiri et al., 2010). TRIOBP-5 provides additional reinforcement in the lower half of rootlets and is necessary to maintain normal width and length of the entire rootlet (Katsuno et al., 2019; Kitajiri et al., 2010). TRIOBP-5 recruits ANKRD24 to rootlets, and in the absence of TRIOBP-5, rootlets are fragile, dysmorphic, splayed, or bent, resulting in progressive deafness (Katsuno et al., 2019; Krey et al., 2022).

Deleterious variants of human *TPRN* (Table S1) are associated with nonsyndromic prelingual hearing loss DFNB79 (Rehman et al., 2010) or progressive hearing loss (Bashir et al., 2013; Li et al., 2010). TPRN deficiency in mouse results in stereocilia degeneration (Chen et al., 2016; Liu et al., 2018; Men et al., 2019; Qi et al., 2024). In inner ear hair cells, TPRN is located near stereocilia insertion/pivot points spanning the tapered base of stereocilia (Rehman et al., 2010). Mouse *Tprn* encodes a protein of 749 aa (RefSeq: NP_780495.2) (Fig. 1 B). Mouse and human TPRN have two regions of sequence similarity to phostensin, a minus-end actin-capping protein that targets protein phosphatase 1 (PP1) to the F-actin cytoskeleton (Kao et al., 2007; Lai et al., 2009) (Fig. 1 B). TPRN also has a nuclear localization signal (NLS, Fig. 1 B). TPRN overexpression causes formation of filamentous actin structures in both nuclei and cytoplasm of COS-7 cells and over-elongation of stereocilia in hair cells (Liu et al., 2018; Rehman et al., 2010). Yet, the exact biochemical function of TPRN and the molecular mechanisms of stereocilia degeneration in *Tprn* mutants are unknown.

Using purified TPRN and newly engineered mice with total or partial deletions of *Tprn*, we studied the function of TPRN at the base of stereocilia and the molecular mechanisms of deafness due to a TPRN deficiency. We show that TPRN forms bendable F-actin bundles with smaller interfilamentous space than between F-actin bundled by espin isoform 1 (ESPN, RefSeq: NP_997570.1), which could be uniquely suited to enable flexibility at the stereocilia pivot point. TPRN has multiple actin-binding sites, but only the N-terminal portion of TPRN is essential and sufficient to bundle F-actin. Refined localization using super-resolution STED imaging shows that native TPRN encircles the F-actin core along stereocilia tapers forming funnel-like patterns similar to that reported using an HA antibody for HA-tagged endogenous TPRN (Qi et al., 2024). TPRN interacts with a membrane receptor PTPRQ, connecting stereocilia F-actin to the plasma membrane. In TPRN-deficient hair cells, TRIOBP-5 and ANKRD24 begin to disappear from rootlets of the shortest third

row stereocilia during postnatal development, leading to excessive pruning of stereocilia and dysmorphic rootlets. Later, these structural abnormalities of stereocilia bundles followed by synaptic rearrangement and excessive accumulation of endosome-like vesicular aggregates in inner hair cell (IHC) cytoplasm by 1 mo of age, indicative of affected MET and/or vesicular transport (Corns et al., 2018; O'Connor et al., 2024). The latter likely exacerbates the progression of hearing loss to profound by 2 mo of age. Based on these data, we propose a model that assumes that F-actin bundling by TPRN at stereocilia pivot points brings actin filaments together to reinforce rootlets, stabilizing stereocilia structure and optimizing mechanosensitivity and sound transduction for sustained sound-induced stimuli.

## Results

### Partially overlapping localizations of TPRN, TRIOBP, and ANKRD24 at stereocilia bases

Comprehensively validated commercial and custom anti-TPRN antibodies were used to clarify and refine the exact localization of TPRN at the base of stereocilia (Fig. 1 B and Fig. S1, A–C) (see also Materials and methods). Airyscan confocal microscopy shows TPRN immunoreactivity spanning the taper region at the bases of all stereocilia (Fig. 1, C and D), overlapping with the ANKRD24 immunofluorescence but extending along the taper above ANKRD24 (Fig. 1, E and E'). In contrast to TPRN, ANKRD24 concentrates at stereocilia insertion points, and a fainter ANKRD24 signal is also present along the rootlets in the cuticular plate overlapping with robust, homogeneously distributed TRIOBP-5 (Fig. 1, F and F'). Super-resolution STED imaging revealed previously unreported ring patterns of TRIOBP-4 in cross sections of the lower rootlet of the outer hair cells (OHCs) (Fig. 1, G and H), consistent with our model postulating that TRIOBP-4 "wraps around" actin filaments along the length of the rootlets (Katsuno et al., 2019; Kitajiri et al., 2010). Similar to our reported labeling in IHCs (Krey et al., 2022), TRIOBP-5 and ANKRD24 form rings around the rootlet F-actin in OHCs. TRIOBP-5 and TRIOBP-4 rings were similar in diameter (Fig. 1, I, O, and P), while ANKRD24 diameter is larger than the TRIOBP-5 diameter (Fig.1, J, O, and P). TPRN also forms rings around the rootlets at stereocilia bases of both OHCs and IHCs (Fig. 1, K–N and Q). However, TPRN localization extends throughout the entire taper region of stereocilia, forming a funnel-shaped structure (Fig. 1 L, arrowheads) enveloping the F-actin core (Fig. 1, L and R). We hypothesize that TPRN interacts with F-actin and influence localization and/or function of other proteins in its immediate proximity, such as ANKRD24 and TRIOBP-5.

### TPRN overexpression produces abnormal F-actin bundles

An EGFP-*Tprn* cDNA expression vector was transfected into organ of Corti (OC) explants from postnatal day 2–4 (P2–4) mice. In WT hair cells, exogenous mouse full-length TPRN (FL TPRN) targets the base of each stereocilium where native TPRN is located (Fig. 2, A and B). Overexpressed under a potent CMV promoter, EGFP-TPRN spreads to the tips of stereocilia, causing their excessive thickening (Fig. 2 B), over-elongation (Fig. 2, B

and C), and sometimes curled and thin F-actin extensions (Fig. 2, D–F). In transfected hair cells, EGFP-TPRN causes not only stereocilia bundle abnormalities but also induces abnormal F-actin structures in the cell cytoplasm and nucleus (Fig. 2, G–J). Overexpression of FL TPRN in COS-7 cells showed that TPRN first appears in the nucleus as dots (Fig. 2 K) and is then seen as short filaments that elongate, coalesce, and form whorl-like bundles. These filaments contain both TPRN and F-actin, as visualized by both EGFP-tagged TPRN and fluorescently labeled phalloidin (Fig. 2 L and Video 1). In COS-7 cells transfected with a TPRN fragment that lacks the first 260 residues of TPRN but has a predicted NLS (aa 617–627, Fig. 1 B), we observed a diffuse accumulation of mouse EGFP-TPRN$^{260–749}$ in the nucleus without formation of the whorl-like F-actin bundles (Fig. 2 M). In agreement, it was previously reported that a truncated human TPRN (RefSeq: NP_001121700.2) translated from methionine-codon 307 (corresponding to mouse TPRN methionine-codon 347; NP_780495.2) does not form filamentous structures but produces diffuse staining in the nuclei of HeLa cells (Ferrar et al., 2012). We also found that FL TPRN with a disabled NLS was localized along cytoplasmic F-actin in COS-7 cells and was not observed in nuclei (Fig. 2 N). These data indicate that the predicted NLS is functional in mouse TPRN, and the first 260 N-terminal residues of TPRN are necessary for F-actin bundling. Interestingly, EGFP-TPRN$^{1–260}$ did not show nuclear localization and specific targeting to stereocilia bases. Instead, it localized along the length of stereocilia causing their over-elongation (Fig. 2, O and P). In nonsensory cells of the OC, EGFP-TPRN$^{1–260}$ overexpression caused elongation and thickening of apical microvilli, forming stereocilia-like bundles (Fig. 2, Q and R). These data also argue for actin cross-linking ability in the first 260 TPRN residues that are absent in TPRN$^{260–749}$. In contrast, a C-terminal fragment EGFP-TPRN$^{730–749}$ was uniformly distributed along stereocilia but did not influence stereocilia bundle morphology, similar to overexpression of control EGFP alone (Fig. 2, S and T). These data suggest that a sequence targeting and/or retaining TPRN at stereocilia tapers is localized between residues 260 and 730, while the first 260 residues at TPRN N-terminus are necessary and sufficient to bundle F-actin.

### Purified recombinant full-length mouse TPRN bundles F-actin in vitro

To investigate if TPRN alone can bundle F-actin, we fused mouse FL TPRN with a thioredoxin (TRX) tag, purified TRX-TPRN from *Escherichia coli* (Fig. 3, A; and Fig. S1, H and I), and used it in actin-bundling assays in vitro. Fluorescent microscopy showed that actin filaments are not bundled when polymerized with TRX alone (Fig. 3 B) but are assembled into prominent bundles in the presence of TRX-TPRN (Fig. 3 C). As in COS-7 cells (Fig. 2, K, L, and N), TRX-TPRN was localized along the length of actin bundles (Fig. 3 C, right) or thinner filamentous structures (Fig. 3 D), likely representing single filaments with lengths of 10.27 ± 0.42 μm ($n$ = 55) comparable with those of single actin filaments shown in the left panel of Fig. 3 B (10.34 ± 0.36 μm, $n$ = 96). This localization pattern differs from that of phostensin, a minus-end actin-capping protein that shares some sequence

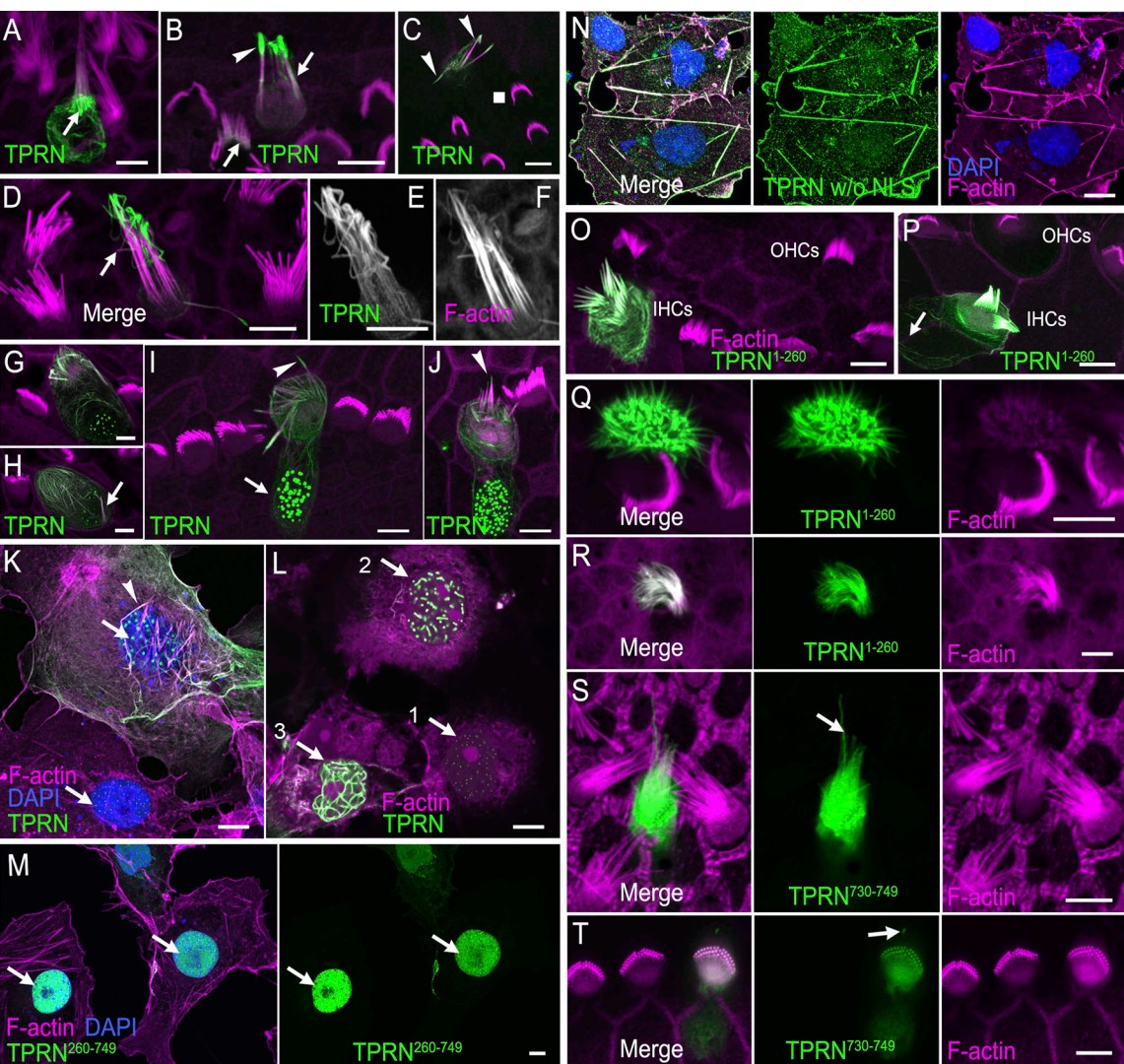

Figure 2. **Exogenous EGFP-TPRN results in F-actin abnormalities in WT IHCs and in COS-7 cells. (A and B)** Similar to endogenous TPRN, transfected FL mouse EGFP-TPRN (green) is localized to stereocilia bases of P2–P4 hair cells (*n* = 2 cells, upward pointing arrows). **(C and G–J)** However, excessive expression of EGFP-TPRN transfected into hair cells leads to mislocalization of EGFP-TPRN at stereocilia tips (*n* = 9 cells) causing abnormal actin remodeling, over-elongation (*n* = 12 cells), thickening (*n* = 10 cells), and degeneration of stereocilia (*n* = 14 cells) (arrowheads in B, C, and G–J). **(D–F)** Thin abnormally long filamentous structures extend from tips of P4 OHC stereocilia (B, down-pointing arrow), and abnormally long, curvy filaments emanate from the tips of stereocilia of some transfected P3 vestibular hair cells (*n* = 5 cells) (D–F). **(E)** Confocal channel showing EGFP-TPRN alone. **(F)** Phalloidin-stained F-actin. The curled filaments contained EGFP-TPRN stained by phalloidin, indicating abnormal actin remodeling at stereocilia tips. **(G–J)** Mislocalization of EGFP-TPRN to stereocilia tips in P3 IHC and OHCs (*n* = 8 cells) (I and J, arrowheads), hair bundle degeneration (*n* = 14 cells) (G–J), and abnormal actin bundling in hair cell cytoplasm (*n* = 6 cells) (H, arrows), as well as dotted pattern of EGFP-TPRN in nuclei of hair cells (*n* = 7 cells) (G, I, and J, arrows) when FL TPRN is over-expressed. **(K)** Similar patterns of EGFP-TPRN expression showing abnormal actin bundles in the cytoplasm (arrowhead) and dotted pattern in nuclei of transfected COS-7 cells (arrows). Note, the dots of TPRN in the nucleus are smaller when TPRN expression is low and become prominent with higher ex-pression levels of EGFP-TPRN. **(L)** Three COS-7 cells show different stages of EGFP-TPRN expression (1, 2 and 3): initially dots of TPRN are observed in nuclei (arrow 1). TPRN dots then become short filaments (arrow 2), which coalesce into a long and thick whorl-like bundles (arrow 3), as also shown in Video 1. **(M)** Overexpressed TPRN$^{260–749}$ (green) localizes to nuclei (blue DAPI staining, arrows) of COS-7 cells (magenta, phalloidin). Right panel shows TPRN$^{260–749}$ (green) channel only. **(N)** FL TPRN (green) co-localizes with F-actin in the cytoplasm of COS-7 cells when NLS of *Tprn-GFP* cDNA is mutated by replacing Lys(K) at positions 621, 622, and 624 to Ala(A), so the protein sequence was changed from "GSSRKKMKISF" to "GSSRAAMAISF". **(O)** In P3 IHCs, EGFP-TPRN$^{1–260}$ localizes along the stereocilia length and over-elongates stereocilia (*n* = 3 cells). **(P)** No nuclear staining is observed (arrow) since TPRN$^{1–260}$ lacks NLS motif. **(Q and R)** Overexpression of a truncated EGFP-TPRN$^{1–260}$ in non-sensory cells of P3 OC explant causes elongation and thickening of apical microvilli (Q), and in the internal sulcus cell area induces additional bundles of microvilli (*n* = 3 cells) reminiscent of stereocilia bundles of nascent vestibular hair cell (R), which appear limp and lacking the staircase architecture of WT stereocilia bundles. **(S and T)** Overexpression of a C-terminal fragment EGFP-TPRN$^{730–749}$ does not alter WT stereocilia bundle morphology and was uniformly distributed along the entire length of P4 stereocilia, the kinocilium (arrow), and cell body (*n* = 5 cells) of vestibular hair cells (S) and P3 auditory hair cells (T) similar to expression of control EGFP alone in hair cells. In all panels phalloidin (magenta) was used to visualize F-actin. Scale bars are 5 µm.

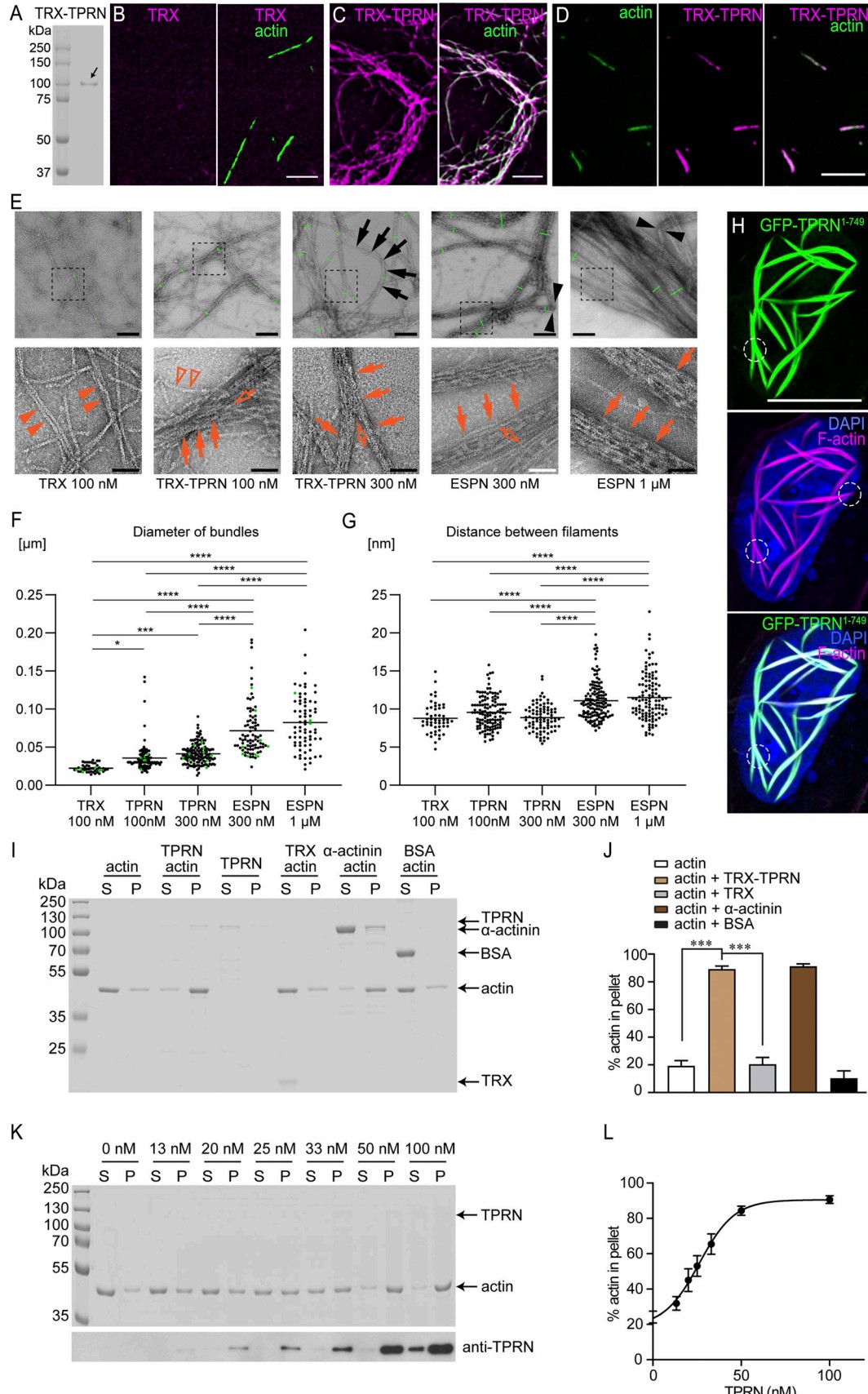

Figure 3. **TPRN bundles F-actin filaments in vitro. (A)** Coomassie blue-stained SDS-PAGE gel showing purified TPRN after expression in *E. coli*. A TRX tag was added to the N terminus in-frame with mouse TPRN, which improves its solubility and stability. The first lane is the protein ladder. The second lane is the

purified TRX-TPRN protein (arrow), the size of which is ~100 kDa. **(B–D)** Actin filaments (green) in the presence of control TRX tag only (B) or TRX-TPRN (magenta) (C and D). Actin (3 µM) was incubated with 0.1 µM TRX or TRX-TPRN at RT for 1.5 h. Using an antibody against TRX, the localization of TRX-TPRN (magenta) along F-actin filamentous structures (green) was visualized by fluorescence microscopy. Note, TRX tag alone (magenta) did not bind to F-actin filaments as shown in B. **(D)** TRX-TPRN (magenta) was distributed along the length of F-actin filamentous structures (green) and not concentrated just at the ends of F-actin. Scale bars in B–D: 5 µm. **(E)** Negative stain TEM images of F-actin polymerized with 100 nM TRX, 100 nM TRX-TPRN, 300 nM TRX-TPRN, 300 nM ESPN, or 1 µM ESPN. Images acquired at a low magnification (upper panels) and high magnification (lower panels). The short green lines across the bundles show the examples of bundle diameter measurements. ESPN is used at 300 nM or above since only a few bundles appear at 100 nM. TRX-TPRN bundles show bending (black arrows), while ESPN bundles occasionally appear kinked (black arrowheads). TRX-TPRN and ESPN bundle actin filaments (red arrows) with bifurcations (open arrows). Filaments bundled by 100 nM TRX-TPRN show single filaments branched from bundles (open red arrowheads). TRX samples occasionally show filaments neighboring one another (red arrowheads), which are analyzed as pseudo-bundles in F and G. Scale bars: 200 nm (E, upper panels) and 50 nm (E, lower panels). **(F)** Average diameter of F-actin bundles. Compared with the diameter of control 100 nM TRX (0.022 ± 0.005 µm, average ± SD, $n$ = 46), the bundle diameter is significantly larger for 100 nM TRX-TPRN (0.035 ± 0.018, $n$ = 102), 300 nM TRX-TPRN (0.041 ± 0.013 µm, $n$ = 138), 300 nM ESPN (0.071 ± 0.034 µm, $n$ = 87), and 1 µM ESPN (0.082 ± 0.035 µm, $n$ = 73). One-way ANOVA shows P < 0.0001. Post hoc multiple comparisons by Tukey (****P < 0.0001; ***P = 0.0001; *P < 0.05; $n.s.$: P ≥ 0.05). **(G)** Average distances between filaments increasing in the order of TRX, TRX-TPRN, and ESPN. The concentration of TRX-TPRN and ESPN does not affect the distances between filaments. One-way ANOVA shows P < 0.0001 (8.7 ± 1.9 nm, $n$ = 52 for 100 nM TRX; 9.5 ± 2.0 nm, $n$ = 122 for 100 nM TRX-TPRN; 8.8 ± 1.7 nm, $n$ = 86 for 300 nM TRX-TPRN; 11.1 ± 2.4 nm, $n$ = 134 for 300 nM ESPN; 11.5 ± 3.0 nm, $n$ = 111 for 1 µM ESPN). Post hoc multiple comparisons by Tukey (****P < 0.0001; $n.s.$: P ≥ 0.05). **(H)** Actin-bundling by EGFP-TPRN in COS-7 cell nucleus. EGFP-TPRN (green), phalloidin (magenta), DAPI (blue). Note the forked/conjoined areas of actin bundles outlined by circles resemble the bifurcated bundles of purified TRX-TPRN in D and E. Scale bars, 10 µm. **(I)** A low-speed (13,500 × $g$) co-sedimentation assay was used to determine the cross-linking activity of TPRN. Actin (3 µM) was incubated alone or with 0.1 µM TRX-TPRN or 6 µM TRX protein or 3 µM α-actinin or 3 µM BSA under polymerization conditions, followed by centrifugation. Equivalent amounts of supernatant (S) and pellet (P) were separated using SDS-PAGE and stained with Coomassie blue. The experiments were repeated at least three times. **(J)** Quantification of the percentage of actin in the pellet is shown in G. Data are represented as mean ± SEM. ***P < 0.001 by unpaired two-sided $t$ test ($n$ = 4). **(K)** Coomassie blue–stained protein gel showing 3 µM actin co-sedimented with TPRN of increased concentrations (13–100 nM). Immunoblotting detected TPRN in supernatant and pellet. **(L)** Percentage of actin in pellet is shown in I ($n$ = 3). 100 nM TRX-TPRN was sufficient to saturate the binding sites of 3 µM F-actin. Data are represented as mean ± SEM. TPRN in I–L is tagged with TRX at its N terminus as indicated in A. Source data are available for this figure: SourceData F3.

similarity with TPRN (Kao et al., 2007; Lai et al., 2009). Thus, purified mouse TRX-TPRN has a similar F-actin–bundling activity in vitro as EGFP-TPRN expressed in WT mammalian hair cells.

Transmission electron microscopy (TEM) of purified FL TPRN incubated with F-actin reveals that negatively stained F-actin only showed bundling when copolymerized with purified TRX-TPRN (Fig. 3 E, second and third columns) and not when incubated with TRX alone (Fig. 3, E, first column). In many instances, we observed TPRN-actin bundles to be curved (Fig. 2 L, arrow 3 and Fig. 3, E and H) and branched (Fig. 3, E and H, black arrows and red open arrows) compared with those bundled by ESPN, which sometimes appeared kinked (Fig. 3 E, fourth and fifth columns, black arrowheads). Actin branching was also reported with fimbrin (Glenney et al., 1981). Branching versus coalescing is hard to discriminate in static images. However, curved TPRN-induced bundles were seen coalescing in COS-7 cell nuclei during a time-lapse recording (Video 1) and in static images (Fig. 3 H, white circles). The average diameter of TRX-TPRN–induced bundles (0.035 ± 0.018 µm for 100 nM, 0.041 ± 0.013 µm for 300 nM) is smaller than ESPN-induced bundles (0.071 ± 0.034 µm for 300 nM, 0.082 ± 0.035 µm for 1 µM) but still larger than the "pseudo-bundles" formed on the grids with 100 nM of TRX protein (0.022 ± 0.005 µm) (Fig. 3 F). The distances between actin filaments increase in the order of the TRX control (8.7 ± 1.9 nm; average ± SD), TRX-TPRN (9.5 ± 2.0 nm for 100 nM, 8.8 ± 1.7 for 300 nM), and ESPN (11.1 ± 2.4 nm for 300 nM, 11.5 ± 3.0 nm for 1 µM) (Fig. 3 G). We speculate that TPRN can bundle actin filaments more tightly than ESPN and slightly less tightly than TRIOBP-4 (Kitajiri et al., 2010), although TPRN bundles are not as straight as ESPN bundles.

Next, the F-actin–bundling activity of purified TRX-TPRN was assessed using low-speed (13,500 × $g$) co-sedimentation assays. Under this centrifugal force, actin is detected in the pellet only if bound to a protein capable of bundling or cross-linking actin filaments (Huang et al., 2005). Negative control experiments using actin alone, TRX alone with actin, or BSA with actin showed that in the absence of TPRN, only ~20% of actin was pelleted (Fig. 3, I and J). However, adding TRX-TPRN caused a significant increase of F-actin in the pellet, similar to the effect of recombinant α-actinin, a well-studied actin–cross-linking protein (Matsudaira, 1991; Sjoblom et al., 2008) we used as a positive control (Fig. 3, I and J). Furthermore, TRX-TPRN increased the percentage of pelleted actin in a dose-dependent manner, elevating it from ~20% to over 90% (Fig. 3, K and L). These data indicate that purified FL mouse TRX-TPRN directly interacts with and bundles F-actin in vitro.

### F-actin–binding sites of TPRN

To identify F-actin–binding sites in mouse TPRN, we fused different TPRN fragments with an in-frame Myc-tag at its N-terminus and individually co-expressed them in HEK293 cells with HA-tagged β-actin (Fig. 4 A). Full-length and several Myc-TPRN fragments efficiently co-immunoprecipitated (Co-IP) HA–β-actin, revealing that TPRN has multiple actin-binding sites (Fig. 4, B and C; and Fig. S1, J–L). Since some actin bundlers, such as α-actinin, function as dimers (Matsudaira, 1991; Ribeiro Ede et al., 2014), we used a Co-IP assay to determine if TPRN can oligomerize. C-terminal EGFP-tagged TPRN (TPRN-EGFP) was expressed alone or together with Myc-TPRN. TPRN-EGFP efficiently Co-IP Myc-TPRN, demonstrating that TPRN can homo-oligomerize, although the stoichiometry is unknown (Fig. 4 D). In addition, we co-expressed TPRN fragments with Myc-, EGFP-, or HA-tags in HEK293 cells. Co-IP assays revealed that multiple regions of TPRN mediate these oligomeric interactions (Fig. 4, E–J).

Figure 4. **TPRN interaction with actin and oligomerization. (A)** Diagram of the TPRN constructs used for biochemical experiments. **(B and C)** TPRN interacts with β-actin. HEK293 cells were transfected with the constructs indicated for each panel. Immunoprecipitations were carried out with Myc antibody, followed by western blotting to detect co-expressed proteins. The upper rows show Co-IP results, and the lower rows show input protein. IP: Myc shows IP of all the constructs with Myc tag. **(B)** HA–β-actin is pulled down by FL Myc–TPRN, indicating its interaction with TPRN. No HA–β-actin was pulled down by Myc antibody without TPRN. **(C)** Each contiguous region of TPRN, (1–170, 171–410, 411–622, and 623–749 [RefSeq: NP_780495.2]) can mediate interactions with F-actin. Please see additional results in Fig. S1, J–L. **(D)** Co-IP shows homomeric interactions between FL TPRN–EGFP and Myc–TPRN. **(E–J)** Homomeric interactions of TPRN fragments illustrated by Co-IP. TPRN$^{1–410}$-EGFP and Myc-TPRN$^{1–410}$, TPRN$^{411–749}$-EGFP and Myc-TPRN$^{411–749}$, TPRN$^{171–410}$-EGFP and Myc-TPRN$^{171–410}$, TPRN$^{411–622}$-EGFP and Myc-TPRN$^{411–622}$, TPRN$^{623–749}$-EGFP and Myc-TPRN$^{623–749}$, and HA-TPRN$^{1–170}$ and Myc-TPRN$^{1–170}$ can oligomerize. Molecular weight markers (kDa) are shown on the left side of each blot. Source data are available for this figure: SourceData F4.

To identify regions of TPRN that can bundle F-actin, β-actin was polymerized in vitro with FL TRX-TPRN and four TRX-tagged TPRN fragments (residues 1–170, 1–300, 1–400, and 301–749, RefSeq: NP_780495.2) and stained with FITC-phalloidin. Of these, FL TRX-TPRN and three N-terminal fragments bundled F-actin as revealed by fluorescence light microscopy (Fig. 5 A). However, TPRN[301–749] did not show detectable F-actin–bundling activity (Fig. 5 A), although it can bind β-actin (Fig. 4 C). The bundling/cross-linking activities of TPRN fragments were quantified using low-speed F-actin co-sedimentation assays (Fig. 5, B–G). FL TPRN has the strongest F-actin cross-linking activity (Fig. 5 G). TRX-TPRN[1–300] and TRX-TPRN[1–170] effectively bundle F-actin; nevertheless, the rightward shift of the curves indicates that the cross-linking activity of purified TPRN is reduced as the length of TPRN is decreased (Fig. 5, G and H). TRX-TPRN[1–63] and TRX-TPRN[301–749] exhibit minimal actin–cross-linking activity (Fig. 5, E–G), consistent with in vitro fluorescence microscopy data showing that TRX-TPRN[301–749] does not bundle F-actin (Fig. 5 A). Similarly, in vivo EGFP-TPRN[260–749] expressed in COS-7 cells does not form whorl-like F-actin bundles in the nuclei as compared with the FL EGFP-TPRN but rather shows a diffuse nuclear signal (Fig. 2 M). In contrast, EGFP-TPRN[1–260] in hair cell stereocilia causes thickening and elongation of stereocilia and microvilli (Fig. 2, O–R), which correlates with in vitro bundling ability of N-terminal TRX-TPRN fragments. Taken together, these data suggest that only residues 1–260 of TPRN harbor the F-actin–bundling activity, while additional actin-binding sites in the remaining TPRN sequence may boost actin–cross-linking activity.

## TPRN is essential for inner ear function

To characterize the expression of *Tprn* mRNA in the inner ear, RNAscope in situ hybridizations (ISH) of P3 mouse inner ear cryosections were performed using two probes, one spanning 840 nucleotides and the other 954 nucleotides of *Tprn m*RNA (Fig. S2 A). In WT mice, *Tprn* mRNA was detected in auditory hair cells and to a lesser extent in supporting cells of the OC. A prominent ISH signal was also observed in the basal cell layer of the stria vascularis (SV) (Fig. S2 B). To a lesser extent, *Tprn* mRNA was expressed in SV marginal and intermediate cells (Fig. S2 B). The expression of *Tprn* mRNA in these inner ear cell types was consistent with single-cell RNAseq data and our RNAseq datasets showing SV *Tprn* expression (https://umgear.org/).

TPRN is ubiquitously expressed in many tissues in mouse (Li et al., 2010; Rehman et al., 2010). To investigate the importance of TPRN in general and for the development and maintenance of inner ear hair cells, we used CRISPR/Cas9 to engineer a ~7.5-kb deletion encompassing all four coding exons and introns of mouse *Tprn*. This mouse is designated *Tprn[em1F3ibTF]* but referred to here as *Tprn[−/−]*. The RNAscope probes did not detect a signal in *Tprn[−/−]* OC and SV cells (Fig. S2 B). Furthermore, immunostaining of the inner ear tissues of WT and *Tprn[−/−]* littermates shows that TPRN is detected only in WT mice (Fig. 6 A), confirming specificity of RNAscope probes and a TPRN rabbit monoclonal antibody (RabmAb).

TPRN immunoreactivity was also present in actin-based apical processes of supporting cells of *Tprn[+/+]* mice (Fig. 6 A″,

arrow), where we previously detected TRIOBP-4 and TRIOBP-5 (Katsuno et al., 2019; Kitajiri et al., 2010). TPRN was not detected in *Tprn[−/−]* supporting cells (Fig. 6 A, right column of images). Western blot analysis of adult mouse inner ear and brain tissue samples revealed the presence of an ~80-kDa protein band in WT (*Tprn[+/+]*) tissue samples, corresponding to the deduced molecular weight of 80.1 kDa for FL TPRN. This ~80-kDa band was absent in tissues from *Tprn[−/−]* littermates, confirming a lack of TPRN and establishing antibody specificity (Fig. S1 D).

Next, we characterized defects caused by the absence of TPRN. Matings of heterozygotes (*Tprn[+/−]*) yielded 309 offspring, showing a 1:2:1 Mendelian ratio for the three genotypes (Chi² = 0.087, P = 0.96) and no significant difference in numbers of males and females for the three genotypes. *Tprn[−/−]* males and females are fertile. Gross body morphology, front and hind limbs, including the joints, bones, and skeletal muscles of *Tprn[−/−]* mice were indistinguishable from WT. Although TPRN is widely expressed in the body, male and female mice at P60–P65 (6 *Tprn[−/−]* mice, 6 *Tprn[+/−]* mice, and 6 *Tprn[+/+]* mice) show no histological abnormalities of 51 different tissues examined blind to genotype. However, inner ear cross sections of *Tprn[−/−]* mice show degeneration, including loss of IHCs and OHCs in the basal turn of the cochlea, often accompanied by loss of phalangeal cells (Fig. S3). The spiral ganglion had moderate to marked loss of neurons at the base and sometimes toward the middle turn. These data indicate that TPRN has an obvious nonredundant function mainly in the inner ear.

Since TPRN is expressed in the SV, we measured the endocochlear potential (EP). EP is generated by SV and is a driving force for positive ions entering hair cells through stereocilia MET channels in response to sound stimuli. In P60 *Tprn[+/+]*, *Tprn[+/−]*, and *Tprn[−/−]* littermates (Fig. S4, A–C), EPs were in the normal range (P = 0.1932, one way ANOVA), and SV morphology was not affected in any of the three genotypes (Fig. S4). These data indicate that TPRN is not necessary to establish SV cellular organization or maintain the EP.

*Tprn[−/−]* mice show progressive deafness with high frequencies affected earlier than low frequencies and are profoundly deaf at P60 (Fig. 6 B; Fig. S5 A; and Tables S2, S3, S4, and S5). *Tprn[−/−]* mice have residual hearing at P19–P21, which is similar to that seen in our *Tprn[N259/N259]* mice, which have a deletion of TPRN residues 260–749 (Fig. S5 B; and Tables S6 and S7) and the *Tprn[in103/in103]* mouse (Liu et al., 2018, Fig. S5 C; and Tables S8 and S9). The thresholds at all frequencies in these mice gradually increase and result in profound deafness by P60 (Fig. 6 B; Fig. S5, A–C; and Tables S2, S3, S4, S5, S6, S7, S8, and S9). An absence of distortion product otoacoustic emissions (DPOAEs, data not shown) in *Tprn[−/−]* mice indicates OHC malfunction. None of the *Tprn*-deficient mice show an obvious qualitative vestibular phenotype, such as circling or head bobbing. Quantitative measurement of vestibular sensory evoked potential (VsEP) shows no significant differences between *Tprn[+/+]* and *Tprn[−/−]* littermates at P30 or P60, except for a difference in VsEP threshold at P60, which was likely attributable to expected test-retest variation between time points (Fig. S5, D–F; and Tables S10 and S11). Thus, TPRN is necessary for auditory hair cell function and, at least in the

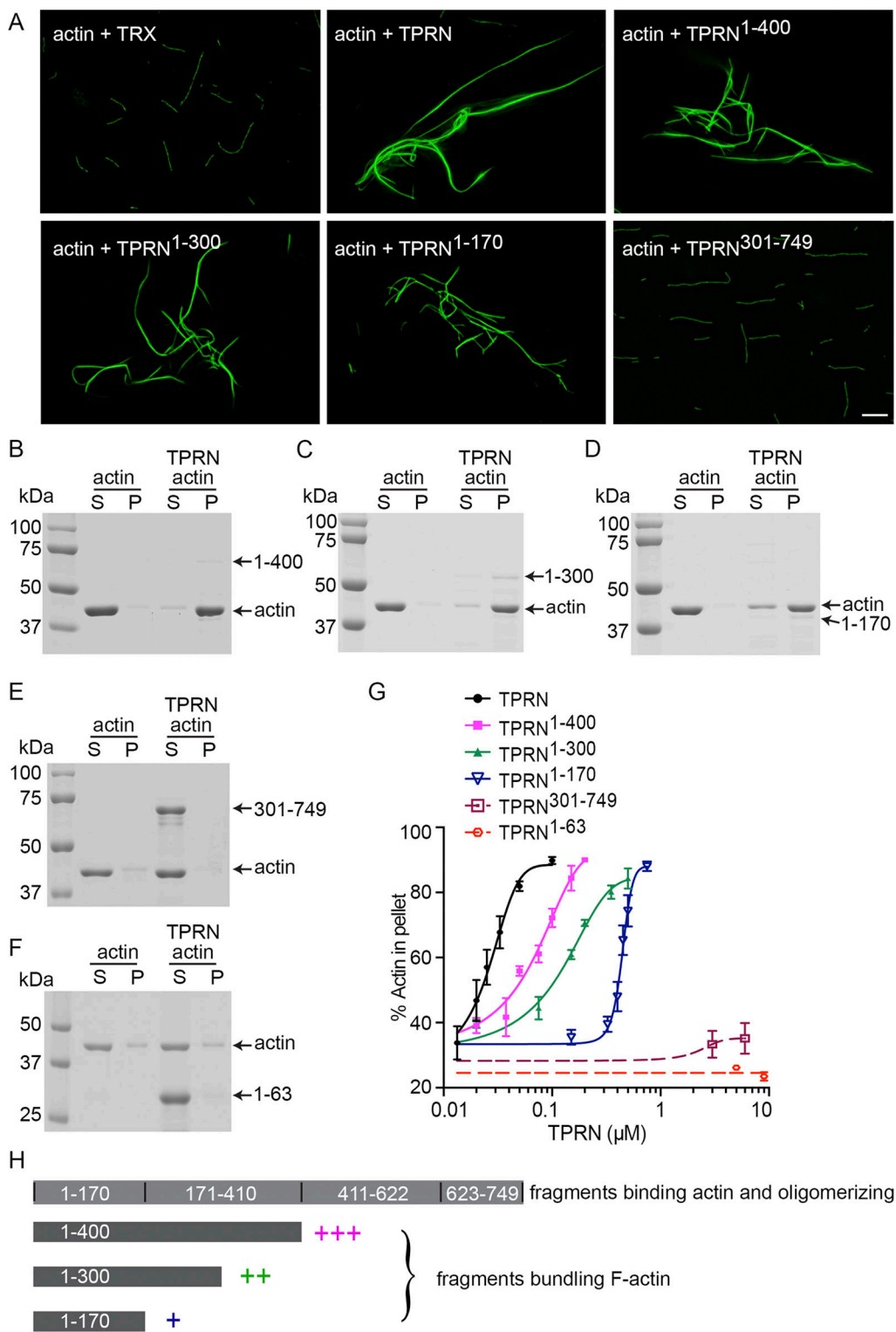

Figure 5. **Characterization of actin-bundling ability of TPRN.** All TPRN fragments and FL TPRN have a TRX tag at the N-terminus. **(A)** Actin filaments in the presence of a TRX tag, FL TPRN, or fragments of TPRN. Actin (3 μM) was incubated with 3 μM TRX, 0.1 μM FL TPRN, 0.15 μM TPRN$^{1-400}$, 0.5 μM TPRN$^{1-300}$, 0.75 μM TPRN$^{1-170}$, or 1.5 μM TPRN$^{301-749}$ at RT for 1.5 h, followed by labeling with fluorescent FITC-phalloidin. The experiment was performed more than three times, yielding consistent results. Scale bars: 5 μm. **(B–F)** Low-speed co-sedimentation assays were used to determine the cross-linking activity of TPRN fragments. Actin (3 μM) was incubated with buffer or 0.2 μM TPRN$^{1-400}$ (B), 0.5 μM TPRN$^{1-300}$ (C), 0.75 μM TPRN$^{1-170}$ (D), 3 μM TPRN$^{301-749}$ (E), or 9 μM TPRN$^{1-63}$ (F) at RT for 1.5 h, followed by centrifugation. Equivalent amounts of supernatant (S) and pellet (P) were separated using SDS-PAGE and stained with Coomassie blue. Arrows with numbers on the right-hand side of each gel indicate corresponding TPRN fragment length in aa used in each experiment. All experiments were repeated at least three times. **(G)** Actin was incubated with varying amounts of FL TPRN or fragments of TPRN. Then, low-speed co-sedimentation assays were performed. The percentage of actin in the pellet was quantified ($n \geq 3$). Data are represented as the mean ± SEM. **(H)** The TRX-TPRN schematics illustrating actin-bundling activities of various TPRN fragments indicated by a bracket based on co-sedimentation data showing that bundling relies on the N-terminal part of TPRN, while all TPRN fragments can bind actin and oligomerize. Source data are available for this figure: SourceData F5.

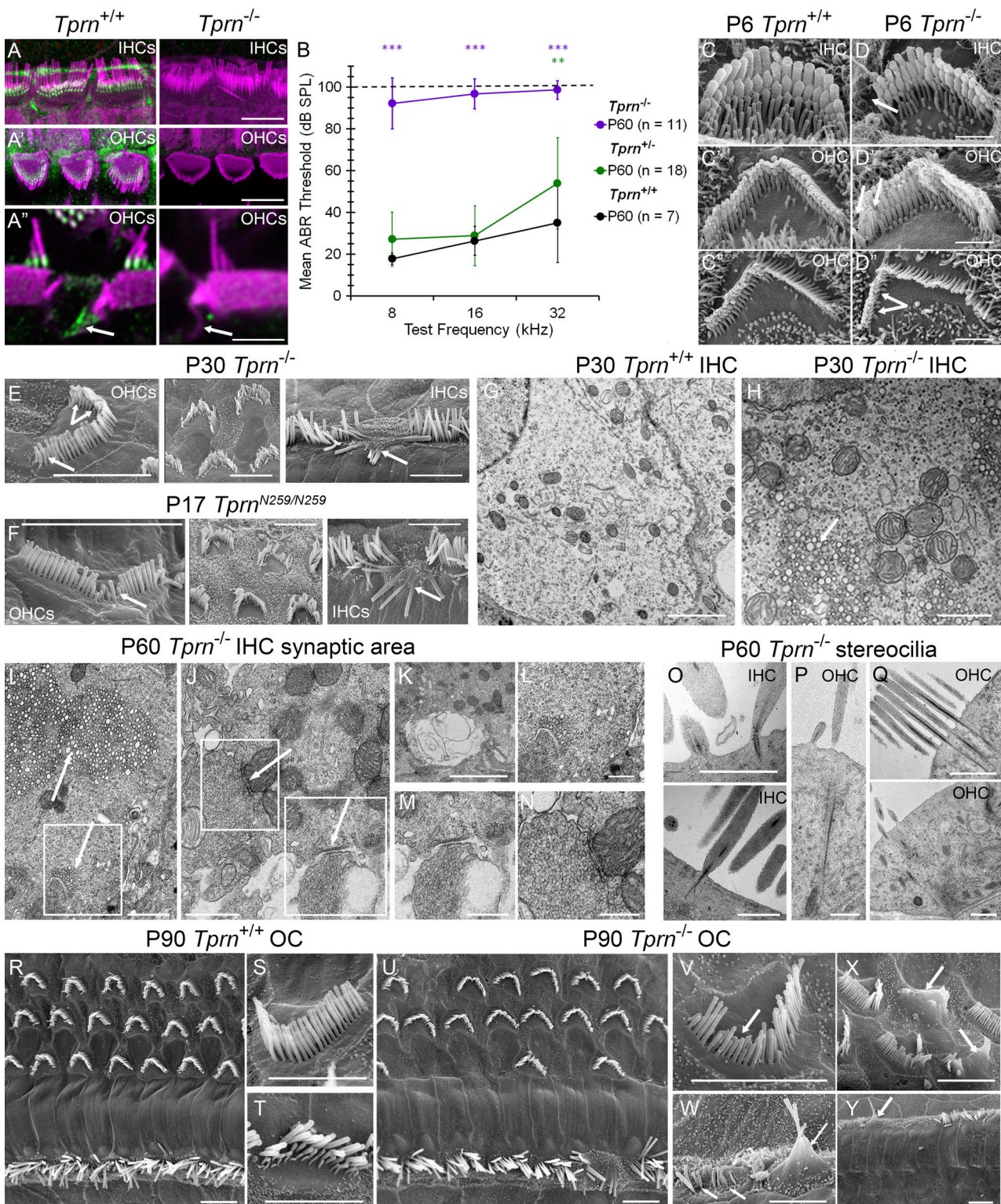

**Figure 6.** ***Tprn*<sup>−/−</sup> mice are profoundly deaf at P60 and show abnormalities of IHCs and OHCs. (A–A")** Airyscan confocal images showing TPRN RabmAb immunoreactivity (green) and F-actin (magenta) in P20 WT (*Tprn*<sup>+/+</sup>) and TPRN null (*Tprn*<sup>−/−</sup>) mouse IHC (A) and OHC stereocilia (A′) and in P10 supporting cells surrounding IHCs and OHCs (A″). TPRN is localized at the base of *Tprn*<sup>+/+</sup> stereocilia and associated with actin cytoskeleton of supporting Deiters' cells but not detected in *Tprn*<sup>−/−</sup> hair cells and supporting cells. **(A″)** Enlarged side view of hair cell stereocilia showing TPRN localization at the base of OHC stereocilia in all three rows and co-localized with F-actin in ascending processes of Deiters' cells. **(B)** Mean ABR thresholds at P60 of *Tprn*<sup>+/+</sup>, *Tprn*<sup>+/−</sup>, and *Tprn*<sup>−/−</sup> littermates at 8, 16, and 32 kHz. Using linear mixed effects regression, we found that *Tprn*<sup>−/−</sup> mice exhibited profound deafness at P60 at all frequencies (Tables S4 and S5), progressive deafness from P18 to P30 and P60 (Fig. S5 A; and Tables S2 and S3) and more pronounced early hearing loss at high frequencies (Fig. S5 A and Table S5). The graph displays mean ± SD. **, ***—significant difference in ABR threshold compared with *Tprn*<sup>+/+</sup> animals at P < 0.01 and P < 0.001, respectively. Color of asterisks indicate the group showing the difference. **(C–D″)** Representative SEM images of IHC and OHC stereocilia bundles of P6 *Tprn*<sup>+/+</sup>

(C–C'') and P6 *Tprn*⁻/⁻ mouse (D–D''). Note, a *Tprn*⁻/⁻ IHC (D) has some stereocilia missing from the first row (arrow) and a less developed third row of stereocilia with more prominent pruning of thin stereocilia/microvilli as compared with a WT IHC stereocilia bundle (C). *Tprn*⁻/⁻ OHC stereocilia bundle from apical turn (D') also has a stereocilium missing from the first row and a shortened stereocilium in the second row (arrows), while all stereocilia in *Tprn*⁺/⁺ apical OHC (C') are present and are of normal length. *Tprn*⁻/⁻ OHC stereocilia bundle from the basal turn of the cochlea (D'') shows multiple missing stereocilia (arrows) from the third row and an altered V shape of the bundle, while the entire third row stereocilia are present in the WT V-shaped OHC from the basal turn (C''). **(E and F)** Stereocilia abnormalities in P30 *Tprn*⁻/⁻ hair cells (E) are similar but less prominent than in P17 *Tprn*^N259/N259 hair cells (F). Shortening and disappearance of the third row stereocilia (E, left panel, forked arrow) and shortening of selective stereocilia from longer rows (E and F, left panels, arrows). Middle panels in E and F show OHCs with abnormal hair bundles. Right panels show fusion of IHC stereocilia (arrows). **(G and H)** TEM images of the synaptic area of a *Tprn*⁻/⁻ IHC (H) show abnormal accumulation of endosome-like vesicular structures (arrow) when compared with a *Tprn*⁺/⁺ IHC (G). **(I–Q)** TEM micrographs of P60 *Tprn*⁻/⁻ hair cells. **(I)** Accumulation of endosome-like vesicles in the cell cytoplasm (arrow) and presence of axosomatic efferent contacts with accumulation of vesicles at the postsynaptic sites (boxed area with an arrow). **(J)** Swollen and damaged IHC efferent presynaptic terminal. Boxed areas in I and J are enlarged in L–N, correspondingly. **(K)** Another example of swollen efferent terminal like in M. **(O)** IHC stereocilia show rootlet fragmentation and breakage at the stereocilia insertion point (top image), splayed rootlets within the cuticular plate, or multiple electron dense spots within stereocilia cores (bottom image). **(P and Q)** OHC stereocilia show long, prominent rootlets, sometimes penetrating abnormally deep into cytoplasm below the cuticular plate (P) and sometimes showing a hollow center of the rootlet structure (top image) and accumulation of the electron dense spots within the cell body nearby long splayed rootlets (bottom image) (Q). **(R–Y)** SEM images of P90 *Tprn*⁺/⁺ OC (R), showing three rows of OHC and one row of IHC from the middle turn of the cochlea with representative normal structure OHC hair bundle (S) and IHC hair bundle (T). **(U)** There are missing OHCs and fused IHC hair bundles in the middle turn of the P90 *Tprn*⁻/⁻ OC. **(V)** Characteristic OHC hair bundle abnormalities with shortened stereocilia of all rows. **(W)** Common abnormalities of IHC stereocilia: fusion (arrow) and abnormally thin taper with some stereocilia absent likely as a result of breakage at the taper (two adjacent arrows). **(X)** OHCs from the upper basal turn also show stereocilia fusion (arrow). **(Y)** Lower basal turn shows complete absence of OHCs and only a few remaining IHCs, some with fused stereocilia bundles (arrow). Scale bars in A, A', E, F, R, U, and Y are 5 μm, and in A'', G, and N are 2 μm. Scale bars in C–C'', D–D'', H, I, K–M, P, and Q are 1 μm, and in J and O are 500 nm.

first few months of life, is not necessary for vestibular hair cell function.

## TPRN null mice show asynchronous shortening of stereocilia and synaptic alterations

TPRN-deficient hair cells were examined using scanning electron microscopy (SEM) and TEM. During the first postnatal week, *Tprn*⁻/⁻ mice show a few missing and/or shortened stereocilia or misshapen hair bundles (Fig. 6, C and D). These abnormalities become prominent in 1- to 2-mo-old *Tprn*⁻/⁻ mice and include fusion of IHC stereocilia, shortening, and loss of individual stereocilia from different rows of OHC bundles, as well as loss of the majority of stereocilia from the shortest third row of OHC (Fig. 6 E). Similar changes were observed in stereocilia of homozygous *Tprn*^N259/N259 mice (Fig. 6 F), with most stereocilia shortening and loss occurring in mechanotransducing stereocilia rows. The progressive degeneration of mechanotransducing stereocilia rows is expected to perturb MET currents in adult *Tprn*⁻/⁻ mice.

In addition to SEM observations, TEM analyses of P30 *Tprn*⁻/⁻ IHCs show numerous membranous vesicles clustered mostly in the cytoplasm below nuclei (Fig. 6, G and H) and at postsynaptic sites neighboring efferent contacts with IHC membranes at P60 (Fig. 6, I–N), indicating a possible vesicular turnover or transport abnormality near the synaptic pole of IHCs. Moreover, in P60 *Tprn*⁻/⁻ mice, we observed axosomatic contacts of efferent endings with IHCs (Fig. 6, I and J, boxed areas), which are normally present only transiently in immature IHCs (Katz and Elgoyhen, 2014; Roux et al., 2011; Simmons, 2002; Simmons et al., 1996). In adult mice, these axosomatic IHC contacts from efferent fibers were reported only after damage to the cochlea, in aging mice, or in mutants with no MET currents (Corns et al., 2018; Lauer et al., 2012; Ruel et al., 2007; Zachary and Fuchs, 2015).

Additional ultrastructural TEM observations of P60 *Tprn*⁻/⁻ mice also reveal stereocilia rootlet abnormalities of IHCs and OHCs. IHCs have broken rootlets at stereocilia insertion sites and/or splayed rootlets. Some rootlets appear "hollow" or have

multiple electron-dense fragments inside stereocilia F-actin cores, although the significance of these findings is unclear (Fig. 6 O). Occasional abnormally long rootlets spanning the cuticular plate and penetrating deep into the cell cytoplasm were observed in P60 *Tprn*⁻/⁻ OHCs (Fig. 6, P and Q). SEM observations of OC from P90 *Tprn*⁻/⁻ mice showed progressive degeneration of stereocilia bundles, as compared with P60–P90 WT mice (Fig. 6, R–T), manifested by absence of third row OHC stereocilia and shortening of some stereocilia from rows 1 and 2 (Fig. 6, U and V). Some stereocilia of either IHCs or OHCs were fused (Fig. 6, W–Y, arrows). This degeneration resulted in missing OHC bundles in the cochlea middle turn at P90 (Fig. 6 X) and their complete absence in the basal turn (Fig. 6 Y), confirming histopathological observations (Fig. S3 and Fig. S4 D). Similar loss of hair bundles was observed for IHCs. At P90, all IHC stereocilia bundles were absent or severely degenerated in basal turns (Fig. 6, U, W, and Y).

## TPRN interacts with membrane receptor PTPRQ

PTPRQ is localized at the base of stereocilia and is necessary for the maintenance of WT stereocilia architecture (Goodyear et al., 2003). In *Ptprq*-null mice (Fig. 7, A, B, F, and G), TPRN was still present at stereocilia tapers. However, the number of fluorescent puncta was significantly reduced in OHCs (Fig. 7, A–C) but not in IHCs (Fig. 7, F–H), while puncta fluorescence intensity level was significantly reduced in IHCs (Fig. 7, I and J) but not in OHCs (Fig. 7, D and E), indicating that PTPRQ may be involved in retention of TPRN at stereocilia bases. Stereocilia within each row of homozygous *Ptprq* KO mice are uneven in length by P15, indicating an asynchronous receding resembling the phenotype of stereocilia in *Tprn*⁻/⁻ hair cells, albeit more severe. Next, we determined if TPRN interacts with PTPRQ using a NanoSPD protein interaction assay (Bird et al., 2017). We found that an mCherry-MYO10-HMM-TPRN fusion protein expressed in COS-7 cells binds to and moves EGFP-PTPRQ to the tips of filopodia (Fig. 7 K). In a control experiment, mCherry-MYO10-HMM localizes to the tips of filopodia but does not transport

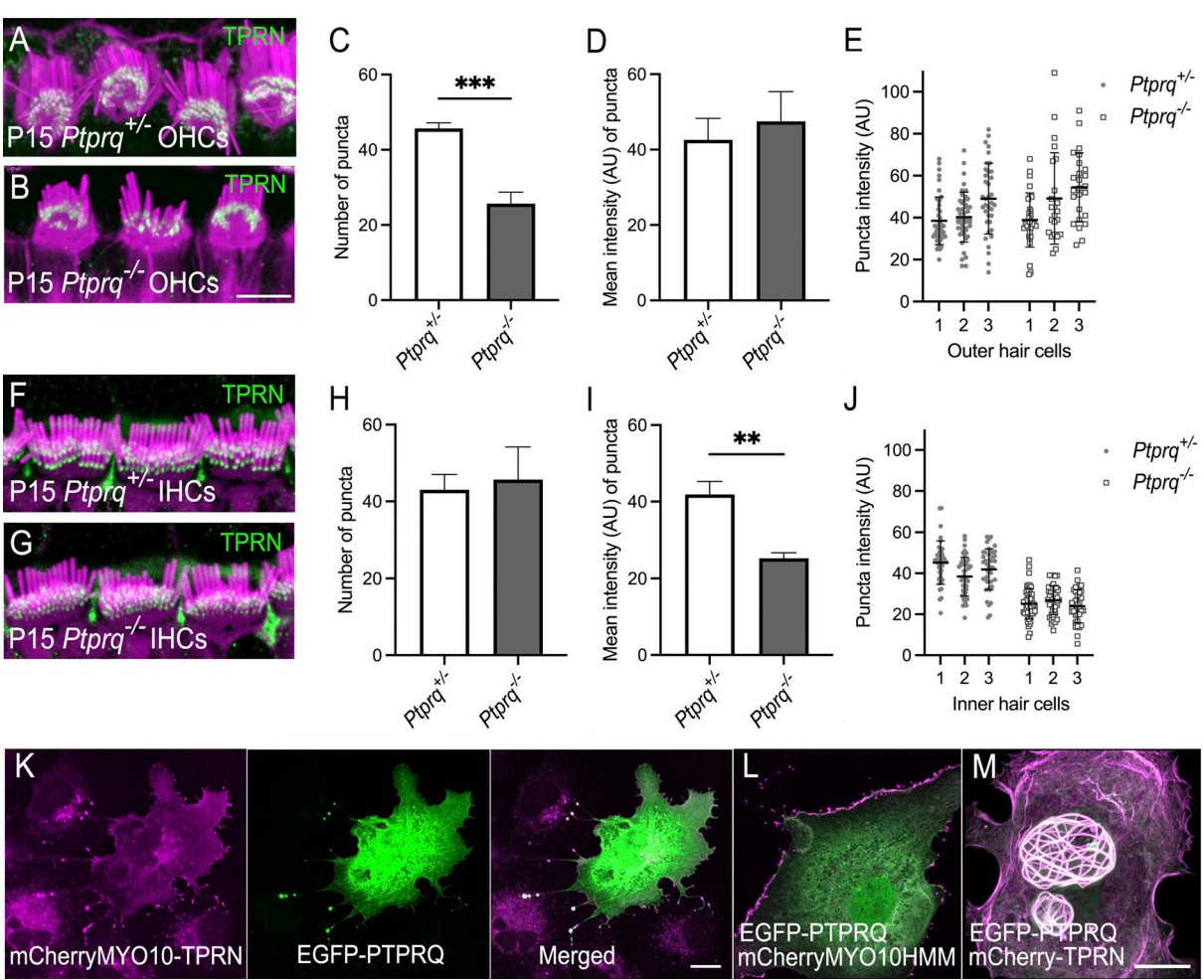

Figure 7. **Localization of TPRN in hair cells lacking PTPRQ and interaction of TPRN and PTPRQ in live cells. (A and B)** Localization of TPRN in P15 normal hearing heterozygous littermate (*Ptprq*$^{+/-}$) and deaf *Ptprq* knockout mouse (*Ptprq*$^{-/-}$) OHCs was examined using anti-TPRN antibodies PB913 and C9ORF75. Both antibodies gave identical staining showing TPRN is concentrated at the base of OHC stereocilia in both *Ptprq*$^{+/-}$ (A) and *Ptprq*$^{-/-}$ (B) mouse OC and visualized as green puncta. **(C)** Bar graph showing quantification of number of TPRN puncta in OHCs for both genotypes. *Ptprq*$^{+/-}$ n = 137/3 puncta/cells and *Ptprq*$^{-/-}$ n = 77/3 puncta/cells. Number of TPRN puncta at bases of OHC stereocilia was significantly reduced in P15 *Ptprq*$^{-/-}$ mice compared with *Ptprq*$^{+/-}$ controls as revealed by unpaired two-sided *t* test (***P = 0.0005). **(D and E)** Bar graph of mean fluorescence intensity of individual puncta showing no statistically significant differences between genotypes by unpaired two-sided *t* test (P = 0.4338). Imaris software spot function was used to calculate the number of puncta per cell for each genotype. ImageJ (Fiji) was used to calculate mean fluorescence intensity value of puncta for each genotype to compare the differences in TPRN expression in *Ptprq*$^{+/-}$ control and *Ptprq*$^{-/-}$ knockout mice. **(F and G)** TPRN localization at the base of IHC stereocilia in P15 *Ptprq*$^{+/-}$ control (F) and *Ptprq*$^{-/-}$ knockout mouse (G). **(H–J)** Quantification of number of TPRN puncta (H) and mean fluorescence intensity of puncta (I and J) in IHCs for both genotypes. Mean TPRN fluorescence intensity of TPRN puncta was significantly reduced in P15 *Ptprq*$^{-/-}$ knockout mouse IHCs (**P = 0.0015 by unpaired two-sided *t* test) but no significant differences were found in number of TPRN puncta in IHC stereocilia, 129/3, number of puncta/cells (*Ptprq*$^{+/-}$) and 137/3 number of puncta/cells (*Ptprq*$^{-/-}$) between the genotypes (P = 0.6489 by unpaired two-sided *t* test). Data are represented as mean ± SD. Puncta from three cells per genotype and three for each hair cell type were analyzed. **(K)** mCherry-MYO10-TPRN fusion construct (magenta) expressed in COS-7 cells targets filopodia tips and co-localizes there with EGFP-PTPRQ (green). **(L)** Control mCherry-MYO10HMM does not target PTPRQ protein to filopodia tips when co-expressed together, confirming that PTPRQ is delivered to the filopodia tips due to its interaction with TPRN (K). **(M)** Overexpressed mCherry-TPRN (magenta) interacts with EGFP-PTPRQ (green), transports PTPRQ to the nucleus, and forms a whorl-like bundle where both TPRN and PTPRQ co-localize. The experiments were performed at least three times, yielding consistent results. All scale bars are: 5 µm; scale bar in B applies to A, B, F, and G; scale bar in M applies to L and M.

EGFP-PTPRQ protein to filopodia tips (Fig. 7 L). Overexpressed mCherry-TPRN concentrates in the nuclei of COS-7 cells and forms whorl-like bundles of F-actin that can interact with PTPRQ and translocate PTPRQ to nuclei, where both are associated with actin bundles (Fig. 7 M). These data suggest that TPRN may connect the actin core to the cell membrane at the base of stereocilia by interacting with membrane receptor protein PTPRQ. Furthermore, TPRN could regulate the actin cytoskeleton at the base of stereocilia in response to environmental signals detected by PTPRQ, a protein known to be important for maintaining stereocilia architecture (Goodyear et al., 2003).

## Absence of TPRN or its overexpression in vivo disrupts stereocilia bundle development

To elucidate molecular mechanisms of stereocilia rootlet abnormalities, we first investigated if localization of two key rootlet proteins, TRIOBP-5 and ANKRD24, is affected in *Tprn*$^{-/-}$ mice. TRIOBP-5 and ANKRD24 are localized in the lower half of

rootlets as early as P0 in WT IHCs (Katsuno et al., 2019; Kitajiri et al., 2010; Krey et al., 2022). Remarkably, the number of TRIOBP-5– and ANKRD24-stained rootlets visualized as puncta at stereocilia insertion points was significantly decreased in P7 $Tprn^{-/-}$ (Fig. 8, A–B″) and $Tprn^{in103/in103}$ mice (Fig. 8 C and Fig. S5, G–J′). The loss of TRIOBP-5 (Fig. 8, A–C; and Fig. S5, G and H) and ANKRD24 (Fig. S5, I–J′) occurs mainly at the base of shorter stereocilia (rows 3–5) in immature OHC and IHC bundles. The number and fluorescence intensity of TRIOBP-5 puncta in taller rows (rows 1–2) were not significantly affected (Fig. 8, A–C; and Fig. S5, G and H). A decrease of TRIOBP-5 puncta was detected in the shortest third row stereocilia of TPRN-deficient IHCs as early as P1–P3 (Fig. 8, D–G). ESPN and GPSM2, which are essential for stereocilia elongation and localized at tips of tallest row stereocilia (Tadenev et al., 2019), were not significantly affected in TPRN-deficient mice (Fig. S5, K–L′). SEM observations confirm that rows 1 and 2 appear normal, but the number of shorter-row stereocilia in TPRN-deficient $Tprn^{in103/in103}$ mice were significantly reduced in early postnatal hair cells (Fig. 8, H–J). Hence, TPRN is essential for the development and subsequent maintenance of third row sound-transducing stereocilia.

To investigate the effect of TPRN overexpression on development of stereocilia, we used adeno-associated virus (AAV) to deliver cDNAs encoding an N-terminal HA-tag fused to FL TPRN (HA-TPRN) or to TPRN fragments into P2 WT mouse inner ears via the posterior semicircular canal. Cochleae were fixed at P7 for immunostaining. Overexpression of FL HA-TPRN by AAV infection resulted in degeneration of hair bundles and an over-elongation of some stereocilia (Fig. 8, K–L′), likely due to excessive actin-bundling activity of overexpressed HA-TPRN. These changes are consistent with stereocilia abnormalities due to overexpression using injectoporation (Liu et al., 2018) and gene gun transfection of TPRN (Fig. 2, A–J). Consistent with TPRN fragments having reduced actin-bundling activity compared with FL TPRN (Fig. 5 G), overexpression of HA-TPRN[1–400], HA-TPRN[1–300], or HA-TPRN[1–170] did not result in obvious over-elongation of stereocilia (Fig. 8, M–O), although occasionally some hair cells had overgrown stereocilia (Fig. 8 S). Strikingly, overexpression of these fragments significantly increased the number and fluorescence intensity of TRIOBP-5–containing rootlets in the shorter row stereocilia (Fig. 8, M–T) and the number of shorter rows of stereocilia (Fig. 8, M–O, R, and R′), revealing a stabilizing function of TPRN that counteracts developmental pruning. Expression of control tdTomato or HA-TPRN[1–63], which cannot bundle actin filament in vitro (Fig. 5, F and G), did not affect expression of TRIOBP-5 or the morphology of stereocilia (Fig. 8, P, Q, and T). Together, these data indicate that either an excess of TPRN or TPRN deficiency disrupt normal development of stereocilia.

## MET currents and stereocilia bundle stiffness in TPRN-deficient young postnatal OHCs

To determine whether MET currents in auditory hair cells are affected by a TPRN deficiency, whole-cell patch-clamp was used to record MET currents produced by fluid-jet deflection of postnatal OHC bundles (P4–P7) located in the middle of the cochlear apical turn (Fig. 9 A). We observed broadly similar MET responses to step deflections of stereocilia in $Tprn^{+/+}$, $Tprn^{-/-}$, and $Tprn^{N259/N259}$ OHCs (Fig. 9, B–D). However, maximal MET current was decreased relative to the WT control in $Tprn^{N259/N259}$ but not $Tprn^{-/-}$ OHCs (Fig. 9 E). The changes of the open probability of MET channels at resting bundle position ($P_{OPEN}$) in $Tprn^{N259/N259}$ OHCs did not reach statistical significance (Fig. 9 F). Thus, selective retraction of shorter stereocilia in $Tprn^{-/-}$ mice observed in Fig. 6, D–D″ is not a consequence of a reduced MET current at P4–P7 but rather represents an intrinsic effect of TPRN on stereocilia stability.

To explore the effects of TPRN deficiency on hair bundle stiffness, we performed high-speed video recordings of stereocilia movements during ramp-like deflections of stereocilia with a fluid-jet in P4–P7 OHCs. The amplitude of this ramp stimulus in each cell was determined from MET current recordings in the same cell (as in Fig. 9, B–D) and was adjusted to saturate the MET current. In contrast to similar experiments with the auditory hair cells of TRIOBP-4/5–deficient mice (Kitajiri et al., 2010), we did not observe evidence of stereocilia breakage even after 50 repetitive intense deflections with a fluid-jet. The relationship between hair bundle deflection (ΔX) and fluid-jet pressure (P) was calculated. A linear fit to this ΔX(P) relationship is proportional to the mechanical compliance of the stereocilia bundle, a parameter reciprocal to stiffness. Although we did observe a smaller average compliance in $Tprn^{-/-}$ OHCs compared with that in WT control and $Tprn^{N259/N259}$ OHCs, variations of compliance values fit into their overall dependence on the area of the tip of fluid-jet pipette (Fig. 9 G). Our data suggest that any potential effects of TPRN-mediated F-actin cross-linking at stereocilia pivot points on mechanical stiffness of hair bundles may be too subtle to be detected by a fluid-jet technique or not present in young postnatal hair cells before the onset of hearing. We note that the loss of the third row stereocilia in TPRN-deficient OHCs (Fig. 6, D–D″) is expected to decrease stiffness and increase hair bundle compliance, but this loss is asynchronous and differentially affects even neighboring young postnatal hair cells.

## 3D focused ion beam SEM reveals rootlet ultrastructural abnormalities

To explore ultrastructural changes produced by a loss of TPRN, we performed serial sectioning with a focused ion beam SEM (FIB-SEM) of fast-frozen freeze-substituted OCs from P17–P34 mice. FIB-SEM revealed several ultrastructural abnormalities in IHC stereocilia of $Tprn^{-/-}$ and $Tprn^{N259/N259}$ mutants, including F-actin breakages at stereocilia pivot points, warped or "kinked" rootlets anchored in the cuticular plate, and "branching" at their lower ends (Fig. 10 A). Partial breakage of F-actin near the insertion points may explain why we observe collapsed TPRN-deficient stereocilia and their fusion with the hair cell apical plasma membrane (Fig. 6, E and F, right panels). Quantification of the percentage of stereocilia rootlets exhibiting these abnormalities showed that $Tprn^{-/-}$ IHCs have a somewhat different phenotype compared with $Tprn^{N259/N259}$ IHCs (Fig. 10 B). The percentage of stereocilia with F-actin breakages at the pivot points and the percentage of "branched" rootlets was significantly increased in $Tprn^{-/-}$ IHCs compared with WT IHCs, while

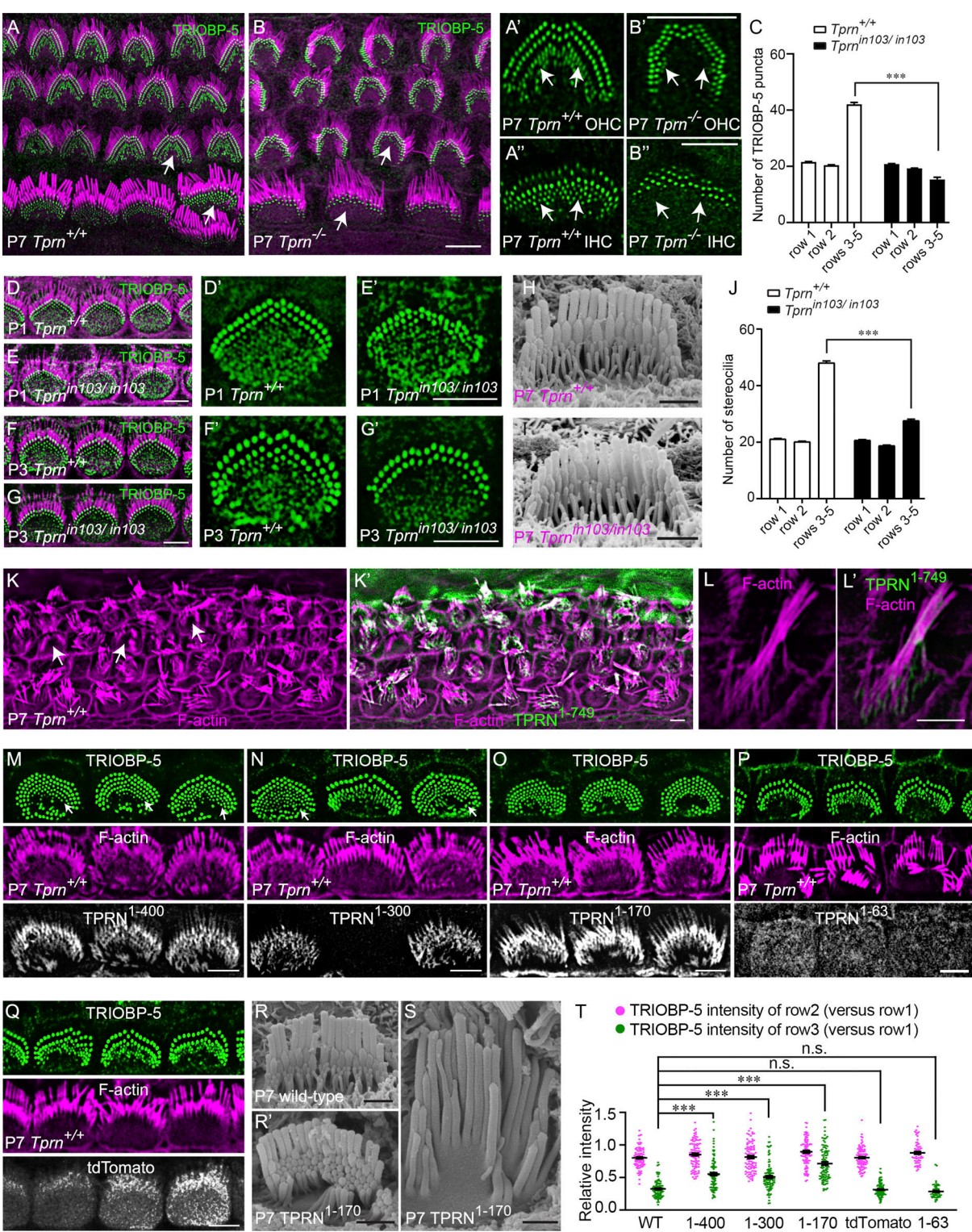

Figure 8. **Abnormal stereocilia development due to constitutive absence or long-term overexpression of TPRN in vivo. (A and B)** Whole mounts of cochlea from P7 WT or *Tprn*⁻/⁻ mice were stained with antibodies against TRIOBP-5 (green) and phalloidin (magenta) to reveal stereocilia. **(A′–B′)** Enlarged images of the OHC hair bundles indicated by arrows in A and B. **(A″–B″)** Enlarged images of the IHC hair bundles indicated by arrows in A and B. Number of TRIOBP-5 puncta at bases of immature short-row stereocilia was significantly reduced in P7 *Tprn*-deficient hair cells. Two arrows point to immature stereocilia rows in A′ compared with B′ and in A″ compared with B″. **(C)** Quantification of TRIOBP-5 puncta at the bases of different rows of stereocilia in P7 *Tprn*⁺/⁺ and in *Tprn*-deficient mouse model *Tprn*^in103/in103^. Fifty-six WT and *Tprn*-deficient IHCs from at least three mice were analyzed per group. Data are means ± SEM. ***P < 0.001 by unpaired two-sided *t* test. **(D–G)** Cochlear whole mounts from P1 (D and E) or P3 (F and G) of *Tprn*⁺/⁺ and *Tprn*^in103/in103^ mice immunostained for TRIOBP-5 and counterstained for F-actin using phalloidin. **(D′–E′)** Enlarged images of TRIOBP-5–stained IHC bundles in D and E. **(F–G′)** As early as P3, there was a reduction of TRIOBP-5 puncta at the base of short-row stereocilia of *Tprn*-deficient mice, better appreciated on enlarged images (F′–G′). **(H and I)**

Representative SEM images of IHC stereocilia bundles from the middle turn of the cochlea of P7 WT (H) and *Tprn*[in103/in103] (I) mice. **(J)** Quantification of the number of stereocilia in different rows. At least 48 IHCs from three mice in each group were analyzed. Data are represented as mean ± SEM. ***P < 0.001 by unpaired two-sided *t* test. **(K and K')** A low-magnification image showing stereocilia of P7 WT hair cells infected at P2 with AAVs expressing FL HA-TPRN. On average, 64.9 ± 6.5% of IHCs exhibited excessive stereocilia elongation. A total of three mice were analyzed, and 27–64 IHCs were assessed in the apical region of the cochlea of each mouse. Arrows indicate cells with apparently normal stereocilia, likely due to the absence of AAV transduction. **(L and L')** Degeneration of P7 WT IHC stereocilia infected at P2 with AAVs expressing FL HA-TPRN. **(K–L')** Hair cells are counterstained with phalloidin to reveal F-actin and an anti-HA antibody to detect HA-TPRN (green). **(M–Q)** P7 WT hair cells infected at P2 with AAVs expressing HA-TPRN[1–400] (M), HA-TPRN[1–300] (N), HA-TPRN[1–170] (O), TPRN[1–63] (P), or tdTomato (Q). Note, multiple rows of TRIOBP-5 puncta in hair cells expressing TPRN[1–400], TPRN[1–300], or TPRN[1–170]. (N) Extra rows of TRIOBP-5 puncta are visualized at the apical surface of infected cells (arrows point to row 5 stereocilia) as compared with a noninfected cell in the middle of the image in N or cells expressing TPRN[1–63] or tdTomato (P and Q). **(R–S)** SEM images of P7 WT noninfected (R) and infected with AAVs expressing HA-TPRN[1–170] (R' and S) hair cells. On average, 5.5 ± 2.1% of IHCs exhibited negligible changes to the hair bundle, 87.4 ± 2.7% exhibited supernumerary rows of short stereocilia with increased thickness (R'), and 7.0 ± 2.9% showed over-elongated stereocilia (S). A total of five mice were analyzed, and 32–41 IHCs were assessed in the apical region of the cochlea of each mouse. Scale bars: 5 μm in A–Q, except 1 μm in H, I, R, and S. **(T)** Relative intensity of TRIOBP-5 in rows 1–3. Significantly increased intensity of TRIOBP in row 3 stereocilia in hair cells expressing TPRN[1–400], TPRN[1–300], or TPRN[1–170]. Data are represented as mean ± SEM. ***P < 0.001 by unpaired two-sided *t* test. tdTomato: *n.s.* P = 0.4854. TPRN[1–63]: *n.s.* P = 0.0504. WT, TPRN[1–400], and TPRN[1–300] n = 47/3 cells/mice; TPRN[1–170] and tdTomato n = 50/3 cells/mice; and TPRN[1–63] n = 30/3 cells/mice.

*Tprn*[N259/N259] IHCs had an intermediate phenotype (Fig. 10 B, left and right). Interestingly, *Tprn*[N259/N259] rootlets appear to "kink" at a similar angle within the cuticular plate (Fig. 10, A and B, middle). In addition, *Tprn*[N259/N259] rootlets were significantly thinner than WT and *Tprn*[−/−] rootlets, at least in first and second row stereocilia of IHCs (Fig. 10 C). Thus, TPRN deficiency results in abnormalities of stereocilia rootlets in both *Tprn*[−/−] and *Tprn*[N259/N259] mice (Fig. 10 D). Some differences between the

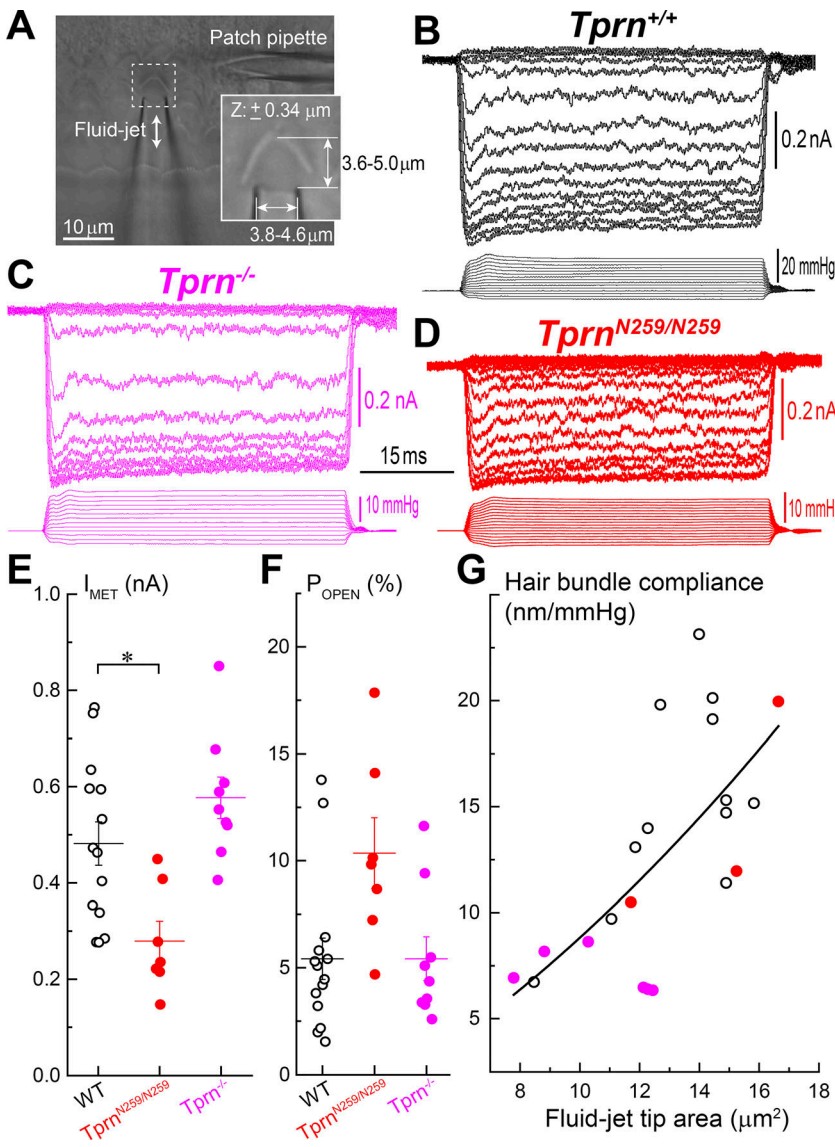

Figure 9. **TPRN deficiency has little effect on MET currents and stereocilia bundle stiffness in young postnatal OHCs. (A)** Bright field image on an OC explant with a patch-clamp pipette (right) and fluid-jet deflecting the OHC stereocilia bundles (bottom). Inset shows a representative frame from a high-speed video recording sequence used to quantify stereocilia bundle movement and to determine positioning of the fluid-jet relative to the hair bundle (see Materials and methods). **(B–D)** Representative MET currents to fluid-jet stimuli evoked by step-like voltage commands in OHCs of control *Tprn*[+/+] (B), *Tprn*[−/−] (C), and *Tprn*[N259/N259] (D) mice. **(E and F)** Maximal MET current (E) and open MET channel probability at resting bundle position, $P_{OPEN}$ control *Tprn*[+/+] (open black circles), *Tprn*[−/−] (closed magenta circles), and *Tprn*[N259/N259] (closed red circles) mice. Data from individual cells and mean ± SEM are shown. Asterisk shows statistical significance: *P < 0.05; unpaired two-sided *t* test. **(G)** Relationship between stereocilia bundle compliance and the tip area of the fluid-jet pipette that determines the force applied to the bundle. Hair bundle compliances were calculated from linear fits of relationships between hair bundle displacement and fluid-jet pressure around resting bundle positions (0–100 nm deflections). All data show the same parabolic function. Age of the cells: P4–P7, location along the cochlea: middle of the apical turn, number of tested cells/mice: WT 12/8; *Tprn*[−/−] 6/2; *Tprn*[N259/N259] 3/3.

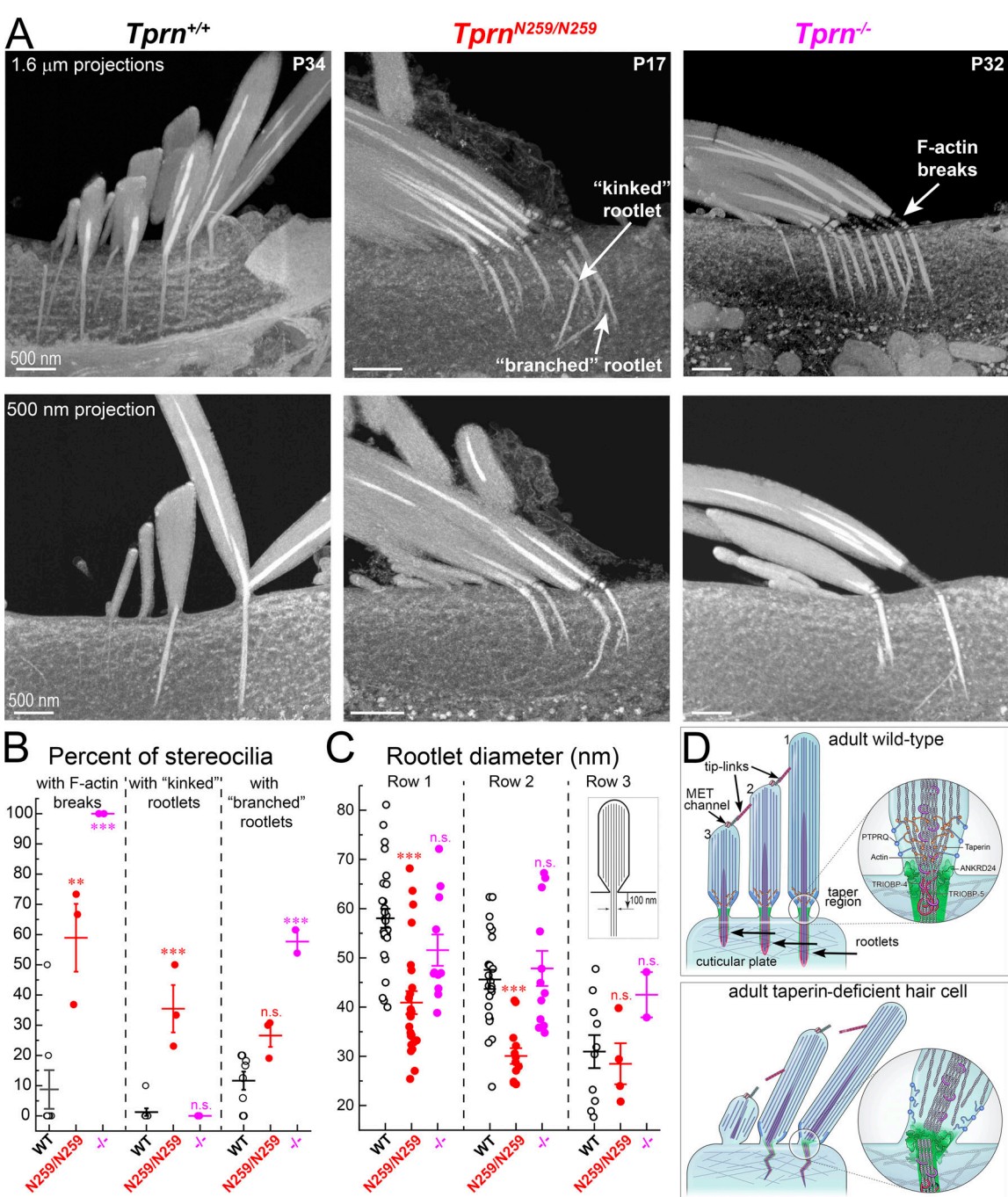

Figure 10. **Abnormalities of stereocilia rootlets and taper regions evoked by TPRN deficiency. (A)** Ultrastructural features of stereocilia tapers and rootlets in control (left), *Tprn*$^{N259/N259}$ (middle), and *Tprn*$^{-/-}$ (right) IHCs. Images represent maximum intensity projections of the stacks of individual FIB-SEM sections with ∼2 × 2 × 20-nm resolution through different overall thicknesses: 1.6 µm (top) and 400 nm (bottom). Arrows point to three typical abnormalities of the rootlets in TPRN-deficient IHCs: (1) "kinks" in the lower portion of the rootlet within cuticular plate, (2) branching at the lower end of the rootlet, and (3) F-actin breaks at stereocilia pivot points. All scale bars are 500 nm. The age of IHCs (P17, P32, or P34) is indicated. **(B)** Percentages of stereocilia with F-actin breaks at the pivot points (left), abnormal kinked rootlets (middle), and branched rootlets (right) in WT (*n* = 8/3, cells/mice, black open circles), *Tprn*$^{N259/N259}$ (*n* = 3/2, cells/mice, red closed circles), and *Tprn*$^{-/-}$ (*n* = 2/1, cells/mice, magenta closed circles) IHCs. In all three categories (F-actin breaks, kinked and branched rootlets), the difference between genotypes was highly significant (P < 0.001, one-way ANOVA). Asterisks show significance of the differences from the WT (post hoc Bonferroni test: **P < 0.01; ***P < 0.001; *n.s.*, not significant). **(C)** Diameters of the lower portion of rootlets were quantified by measuring at the level of 100-nm down from IHC stereocilia pivot points (schematic in the inset of C) in the same WT, *Tprn*$^{N259/N259}$, and *Tprn*$^{-/-}$ IHCs as in B. Each point is one rootlet. The decrease in diameters of rootlets from in rows 1 and 2 of *Tprn*$^{N259/N259}$ stereocilia was significantly different from WT (P < 0.0001, one-way ANOVA, post hoc Bonferroni test: ***P < 0.001; *n.s.*, not significant). P17 *Tprn*$^{N259/N259}$ and P32 *Tprn*$^{-/-}$ IHCs were processed and analyzed in parallel with corresponding WT IHCs (*n* = 4 for each group). Since no statistically significant differences were found between P17 and P34 for the WT, the data for WT were combined. **(D)** Drawings illustrate adult WT (top) and adult TPRN-deficient (bottom) stereocilia with magnified views of their rootlets and tapers and localization of stereocilia base proteins discussed in this study. The enlarged view of a WT stereocilia taper region is shown in a circle of top panel. In the bottom panel, an enlarged image shows details of structural disruptions at the taper region in the absence of TPRN.

rootlet phenotypes in $Tprn^{-/-}$ and $Tprn^{N259/N259}$ hair cells may reflect residual actin-bundling capability of just the first 259 TPRN residues (Fig. 5 G) presumably expressed in $Tprn^{N259/N259}$ mice.

## Discussion

Here, we show that purified mouse TPRN directly bundles F-actin in vitro reflecting structures observed when TPRN is overexpressed in vivo. TPRN is located at the stereocilia taper immediately above the lower half of the rootlet that is encircled by TRIOBP-4, TRIOBP-5, and ANKRD24 and is strategically located to bring actin filaments of the stereocilia core closer together to traverse the pivot point (Fig. 10 D). We hypothesized that TPRN brings actin filaments together based on two observations: (1) in vitro, the distance between actin filaments bundled by purified TPRN is smaller compared with a distance between F-actin bundled by ESPN (Fig. 3 E). (2) In the presence of TPRN, actin filaments merge in live transfected COS-7 cells (Video 1 and Fig. 3 H, white circles). Without TPRN, the central F-actin filaments of a stereocilium can still form a rootlet, perhaps through bundling by TRIOBP-4/5. However, these TPRN-deficient rootlets are not durable, prone to breakages and branching/kinking (Fig. 10), reflecting altered interactions with the proteins anchoring the rootlet to the actin meshwork of the cuticular plate. The effects of TPRN deficiency on stereocilia bundles could be seen as early as P1–P7 (Fig. 6, C and D; Fig. 8, B–J; and Fig. S5, G–J′), which argues in favor of abnormal development of the rootlets rather than their accelerated damage, although the latter supposition cannot be excluded as we observed increased F-actin breakages at P17 $Tprn^{N259/N259}$ and P32 $Tprn^{-/-}$ stereocilia (Fig. 10 B). Our data also suggest an interaction of TPRN with the membrane receptor protein PTPRQ (Fig. 7, K–M), which could affect stereocilia actin core stability. The loss of F-actin bundling by TPRN in mice lacking TPRN also results in a gradual disappearance of TRIOBP-5 and ANKRD24 and retraction of the shortest stereocilia rows during postnatal development (Fig. 8, A–G′; and Fig. S5, I and J, correspondingly). All these effects of a TPRN deficiency in postnatal hair cells before the onset of hearing are MET independent, e.g., occur without significant loss of MET current in $Tprn^{-/-}$ hair cells (Fig. 9). The effect of TPRN deficiency on mechanical properties of stereocilia bundles is different from the reported effects of TRIOBP-4/5 or TRIOBP-5 deficiencies (Katsuno et al., 2019; Kitajiri et al., 2010) since it is not associated with significant loss of mechanical stiffness (Fig. 9 G). Our data indicate the direct involvement of TPRN in the formation and proper organization of F-actin filaments at stereocilia pivot points.

### Stereocilia bundle development requires tight control of TPRN expression

Early in postnatal development, rodent auditory hair cells form excessive numbers of stereocilia-like projections, most of which are subsequently retracted, leaving only rows 1 through three in a mature hair cell (Hadi et al., 2020; Kaltenbach et al., 1994; Krey et al., 2023; Waguespack et al., 2007). The mechanism of this developmental refinement of the hair bundle architecture is

unknown. Our data show a role of TPRN in this process. TPRN deficiency results in abnormal pruning of third row stereocilia, whereas overexpression of N-terminal TPRN fragments causes retention of multiple supernumerary stereocilia rows (Fig. 8). This retraction cannot be a secondary effect of the loss of MET current (Velez-Ortega and Frolenkov, 2019), since MET is largely preserved in young postnatal auditory hair cells at least in $Tprn^{-/-}$ hair cells (Fig. 9). Thus, a controlled level of endogenous $Tprn$ expression is essential for dose-dependent regulation of the number of stereocilia retained in a hair bundle. Inability to restore precisely a WT level of TPRN may explain a negative result of the first $Tprn$ gene therapy attempt in mouse (Qi et al., 2024). Our data show that TPRN overexpression is cytotoxic to hair cells. In 90% of transfected hair cells (Fig. 2, B–F), we observed excessive actin bundling leading to overelongated, thickened, and misshaped stereocilia and subsequent hair cell degeneration. In addition, 67% of hair cells with bundle abnormalities also had abnormal nuclear F-actin structures (Fig. 2, G–J). Elevated TPRN expression is shown to be prognostic for renal and cervical cancer (The Human Protein Atlas, https://www.proteinatlas.org/ENSG00000176058-TPRN/pathology/). However, loss of TPRN function causes only deafness in mouse (Li et al., 2010; Liu et al., 2018; Men et al., 2019; Qi et al., 2024) and human (Table S1) (Rehman et al., 2010).

### Possible function of TPRN in hair cell nuclei

A truncated version of human TPRN that begins with methionine-307 ($TPRN^{307–711}$), mistaken at the time for FL TPRN, was located in nuclei when transfected into HeLa cells and involved in DNA damage repair (Ferrar et al., 2012). We reported that chromodomain helicase DNA-binding protein 4, CHD4, was a partner of TPRN (Bird et al., 2017). However, we did not detect endogenous FL TPRN in the nucleus of a WT mouse hair cell using validated TPRN antibodies. When overexpressed, FL TPRN was found in nuclei of hair cells and heterologous cells, and this localization depends on the presence of a NLS in TPRN (Fig. 2). We speculate that TPRN's NLS is masked in healthy WT hair cells but under stress TPRN enters the nucleus. Future studies may reveal a "damage sensing" mechanism for TPRN in auditory hair cells.

### TPRN role in vesicular transport and synaptic function in IHCs

In P30–P60 $Tprn^{-/-}$ mouse hair cells, there is an abnormal accumulation of aggregated endosome-like vesicles, which was previously observed only in 8-mo-old aging C57BL/6J mice (Stamataki et al., 2006), and the presence of efferent axosomatic IHC contacts (Fig. 6). There are several possible explanations for these phenomena. First, a progressive shortening and loss of stereocilia in P30 $Tprn^{-/-}$ mice likely deteriorate MET currents later in development, which is known to result in similar synaptic abnormalities of IHCs (Corns et al., 2018; Lauer et al., 2012; Ruel et al., 2007; Zachary and Fuchs, 2015). Second, there might be a direct role of TPRN in synaptic function. TPRN has a conserved NPF motif (Fig. 1 B) which is known to interact with EH domains of proteins that function in endocytosis and vesicular trafficking (de Beer et al., 2000). TPRN also possesses three

predicted YXXphi motifs implicated in endosomal and secretory pathways (Fig. 1 B). In addition, TPRN has a PP1 KISF docking motif and was identified as a member of the PP1 interactome (Esteves et al., 2012). In human brain, PP1 is involved in regulating synaptic transmission and plasticity (Allen et al., 2004). Finally, there is continuous vesicle trafficking between the apex and base of an IHC (Ashmore, 2004). Profound changes of the F-actin structures at the apex of adult IHCs may disrupt this traffic in TPRN-deficient IHCs.

**TPRN–PTPRQ interaction links F-actin to the membrane**
In this study, super-resolution STED imaging refined our reported localization of TPRN (Rehman et al., 2010) showing that native TPRN localizes along the taper region of stereocilia forming a funnel-like shape. This pattern is similar to TPRN-HA visualized with an anti-HA antibody (Qi et al., 2024). We speculate that TPRN envelopes peripheral F-actin bringing them together and incorporating them into the lower half of rootlets. Interestingly, in guinea pig, the peripheral actin filaments at the base of the stereocilium shaft curve and contact the central rootlet and do not terminate near the membrane (Furness et al., 2008). At the taper region of the stereocilium, in rat, mouse, and guinea pig, rootlets have a central core surrounded by a concentric dense ring associated with the peripheral actin filaments. This ring of unknown nature described by Furness and coauthors that narrows and converges with the central core of rootlets toward the stereocilium insertion point may consist of TPRN that bundles actin filaments destined to converge into the rootlets. PTPRQ has a similar localization pattern to TPRN at stereocilia tapers (Goodyear et al., 2003). Bundling of F-actin by TPRN and its interaction with membrane receptor PTPRQ may be important to relay extracellular signals to the F-actin core. The TPRN immunofluorescence is diminished in PTPRQ knockout mouse, indicating that PTPRQ may be important for TPRN retention at the base of stereocilia in a similar manner to GRXCR2 (Liu et al., 2018). Interestingly, the PTPRQ knockout mouse also shows an asynchronous retraction of shorter row stereocilia indicative of stereocilia F-actin instability (Goodyear et al., 2003). We speculate that interaction of TPRN with PTPRQ is essential to stabilize stereocilia.

**Novel function of TPRN as an F-actin–bundling protein at stereocilia pivot points**
Some F-actin cross-linkers, like FSCN1, are monomeric and rely on multiple actin-binding domains (Lamb and Tootle, 2020), while other actin bundlers, like α-actinin, function as dimers (Matsudaira, 1991; Ribeiro Ede et al., 2014). We found that TPRN oligomerizes with an unknown stoichiometry. When TPRN is divided into four consecutive fragments, each of them binds to actin, indicating multiple actin-binding regions (Fig. 4 C). However, only the N-terminus of TPRN exhibits actin-bundling activity in vitro and in vivo. The sequence of the N-terminal 1–260 residues of TPRN shows no similarity to the actin-interacting motifs reported for FSCN1, TRIOBP, members of the plastin family, α-actinin, spectrin, ESPN, filamin, dystrophin, scruin, dematin, or villin. How TPRN bundles F-actin is unknown and awaits studies of the atomic structure of TPRN

cross-linked to F-actin using cryo-EM tomography or crystallography. Nevertheless, from negative staining TEM, we observed that purified mouse TPRN organizes F-actin into tightly packed flexible bundles with smaller actin interfilamentous distance as compared with more stiff F-actin bundles induced by ESPN or fascin-1 (Chen et al., 1999; Jansen et al., 2011). TPRN-induced F-actin bundles resemble those bundled by α-actinin (Claessens et al., 2006) or PLS1 (fimbrin) in vitro (Glenney et al., 1981) for which branching was reported (Glenney et al., 1981). Tighter bundling of actin filaments by purified TPRN than ESPN in vitro suggests that TPRN can merge F-actin at stereocilia tapers to bring them closer together to traverse the insertion points. In addition, TPRN–F-actin bundles are often curved (Fig. 3, E–H and Video 1) and subsequently coalesce into a whorl-like structures. The coalescence of actin filaments/small bundles into larger bundles observed with the time-lapse recordings of actin bundle assembled by overexpressed TPRN in the nuclei of live COS-7 cells agrees with negative stained TEM data showing split and coalescing bundles (Video 1). We posit that TPRN produces flexible bundles that lessen damage during sustained acoustical stimulation, augmenting stereocilia rootlet formation and reinforcing the stereocilia pivot points, enabling long-lasting hearing function.

As an F-actin bundler at the base of stereocilia, TPRN is well suited for controlling actin/myosin-based transport of various stereocilia proteins, including components of MET machinery. However, our data show that MET currents are largely unaffected at least in $Tprn^{-/-}$ OHCs. It is still possible, nonetheless, that $Tprn^{N259/N259}$ OHCs have decreased MET current due to abnormal F-actin bundling in rootlets caused by truncated TPRN. The alternative explanation for the decreased maximal MET current in $Tprn^{N259/N259}$ hair cells would be an increased fragility of their hair bundles. However, this is not supported by the fact that we were unable to detect any obvious effects of TPRN deficiency on pivot stiffness of stereocilia, at least in young postnatal OHCs. Furthermore, in contrast to our previous observations in TRIOBP-4/5–deficient hair cells (Kitajiri et al., 2010), TPRN-deficient stereocilia of young postnatal OHCs are not excessively fragile and can withstand intensive repetitive deflections. Thus, independent of the differences in MET currents of $Tprn^{-/-}$ and $Tprn^{N259/N259}$ OHCs, our finding indicates that TPRN function at the base of stereocilia is different but complementary to that of TRIOBP-4/5. It is possible that bundling of actin filaments by TPRN somehow optimizes pivot stiffness of stereocilia after the onset of hearing and later in adult, and the effect in young postnatal hair cells may be too subtle to be revealed by a fluid-jet technique. This optimization is not a trivial task, since an IHC has to balance the requirements for best sensitivity to sound-induced stimuli and insensitivity to Brownian motion (Denk et al., 1989). In OHCs, the balance is even more crucial since the proper stiffness of OHC bundles is crucial for cochlear amplification (Legan et al., 2000).

**TPRN is crucial for the macromolecular complex at a stereocilium pivot point**
In summary, our data show that in vitro, purified TPRN alone and its N-terminal fragments can merge F-actin filaments into

flexible and curved bundles with smaller interfilamentous distance than the distance between F-actin in ESPN-induced bundles. In vivo, TPRN overexpression produced similar F-actin bundles. Furthermore, our data indicate that F-actin bundling by TPRN and its interaction with PTPRQ might be essential to stabilize stereocilia cores and rootlets, two different F-actin compartments. At the base of stereocilia, TPRN is strategically positioned to gather and converge actin filaments toward the narrowed insertion point and incorporate them into tightly packed rootlets. We posit that TPRN organizes actin filaments for proper rootlet formation in development and to allow durable deflections of stereocilia by sound throughout a lifetime.

## Materials and methods

### Immunostaining and antibodies

Inner ears from mice at different postnatal ages, ranging from P0 to P65 were dissected and fixed for 2 h with 4% (wt/vol) formaldehyde in PBS at RT and used for immunostaining. Tissues were washed with 1× PBS, microdissected, and permeabilized in 0.5% (wt/vol) Triton X-100 for 15–20 min at RT. The samples were blocked with 2% BSA and 5% normal goat serum in PBS and incubated with primary antibodies at 5 µg/ml, followed by Alexa Fluor–conjugated secondary antibodies at 2–5 µg/ml. Samples were washed in 1× PBS three times, 10 min each, and mounted using ProLong Gold antifade reagent (Thermo Fisher Scientific) and imaged at RT with an LSM880 Airyscan confocal microscope, and a 63× 1.4 NA oil immersion objective using ZEN software with Zeiss Airyscan processing using auto setting and Z-stack acquisition. Maximum intensity projections were obtained for some images as indicated in figure legends. Gamma adjustments were not used for any of the images.

A commercial antibody against TPRN was purchased from Sigma-Aldrich (C9ORF75, HPA020899; RRID: AB_1845835; Sigma-Aldrich). Custom rabbit polyclonal antibodies against TPRN were developed with help of Covance using antigenic peptides corresponding to residues 726–749 and 516–535 as epitopes. The anti-TPRN RabmAb was prepared by immunizing three rabbits with a peptide corresponding to residues 726–749 of mouse TPRN (RefSeq: NP_780495.2). Several hundreds of clonal ascitic fluids were prepared by Epitomics and tested in immunolocalization experiments. Monoclonal antibodies from selected ascitic fluid clones were then affinity purified. These antibodies were first validated using both TPRN overexpression in COS-7 cells (CRL-1651, RRID:CVCL_0224; ATCC), which showed a strong signal mimicking localization of an introduced TPRN construct (Fig. S1, E–G), and the absence of a signal for TPRN in inner ear hair cells from a mouse, which was engineered with a 7.5-kb genomic DNA deletion that removes all four protein-coding exons and intronic sequence of mouse *Tprn* (*Tprn^emIF3ibTF^*). This mouse is on a C57BL/6J congenic genetic background and, for simplicity, is referred to here as *Tprn^−/−^*. Another anti-TPRN custom-made antibody to the residues 730–749 of mouse TPRN was made and evaluated for the specificity previously (Liu et al., 2022). In addition, the following primary antibodies were purchased and used for the experiments: anti-TRIOBP-5 (16124-1-AP, RRID:AB_2209237;

Proteintech) that we validated previously (Krey et al., 2022), anti-TRIOBP-4/5, which was generated using a peptide common to both TRIOBP-4 and TRIOBP-5 and predominantly recognizes TRIOBP-4 (Kitajiri et al., 2010); anti-HA (2367, RRID: AB_10691311; Cell Signaling Technology), anti-TRX (71542-3, RRID:AB_10808058; MilliporeSigma), anti-ESPN (sc-393469, RRID:AB_2905522; Santa Cruz Biotechnology), anti-ANKRD24 (sc-241811, RRID:AB_10851440; Santa Cruz Biotechnology, discontinued), anti-GPSM2 (HPA007327, RRID:AB_1849941; MilliporeSigma), anti-MYO7A (25–6790, RRID:AB_10015251; Proteus Biosciences), and anti-KCNJ10 (NBP1–70371, RRID:AB_11006134; Novus Biologicals). We also used the following secondary antibodies: Alexa Fluor 488 goat anti-rabbit (A-11070, RRID: AB_2534114; Thermo Fisher Scientific), Alexa Fluor 546 goat anti-rabbit (A-11071, RRID:AB_2534115; Thermo Fisher Scientific), Alexa Fluor 546 donkey anti-goat IgG pAb (A-11056, RRID:AB_2534103; Thermo Fisher Scientific), donkey anti-goat Alexa Fluor 488 (A-11055, RRID:AB_2534102; Thermo Fisher Scientific). Phalloidin Atto 390 (50556; MilliporeSigma), rhodamine phalloidin (R415; Thermo Fisher Scientific), Alexa Fluor 546-phalloidin (A22283; Thermo Fisher Scientific), Alexa Fluor 488-phalloidin (A12379; Thermo Fisher Scientific), Alexa Fluor 568-phalloidin (A12380; Thermo Fisher Scientific), and Alexa Fluor 647-phalloidin (A22287; Thermo Fisher Scientific) were used to visualize F-actin. Samples were mounted using ProLong Gold antifade reagent (P36934; Thermo Fisher Scientific) or ProLong Gold antifade with DAPI reagent (P36935; Thermo Fisher Scientific).

### STED microscopy and immunostaining

Cochleae from P6 to P13 C57BL/6J mice were fixed in 4% PFA for 1 h at RT, washed in 1X PBS three times each for 10 min, permeabilized using 0.2% Triton X-100 for 15 min at RT, and after a brief wash, incubated in blocking solution containing 2% BSA in 1X PBS and 0.1% Tween 20 for 1 h at RT. At the end of the incubation, primary antibody–nanobody complexes were prepared by mixing 1 µg of the primary antibody with 0.3–1.0 µg of the nanobody conjugated to Star Red Abberior dye (43354-1 mg; Sigma-Aldrich) in a total volume of 20 µl of 1X PBS in a fresh tube. The antibody–nanobody complexes were incubated for 40 min at RT and then diluted to a final staining volume in 1X PBS (1 µg of the primary antibody in 100 µl of 1X PBS) per sample. The samples were washed once in PBS briefly, and the antibody–nanobody complexes were added to the tissue samples and incubated for 1 h at RT in a humid chamber protected from light. Then, samples were washed 3X 5 min in 1X PBS and Star 580 Phalloidin (ST580; Abberior, Inc.) was applied at a 1:100 dilution (1 U/ml) to counterstain F-actin and incubated for 1 h at RT. Following the counterstaining, the samples were washed three times 5 min each in 1X PBS and mounted using No 1.5 glass coverslip with ProLong Gold antifade mountant (P36934; Thermo Fisher Scientific). Images were acquired using Abberior FACILITY microscope system (Abberior, Inc.) or STEDYCON super-resolution system (Abberior, Inc.) on an Eclipse Ti2 microscope (Nikon) equipped with Nikon Plan-Apo λ D 100× 1.45 N.A. Oil DIC objective (Nikon). Raw images were acquired at RT with Abberior's Lightbox software for image acquisitions and

analyses or STEDYCON software (version 9.0.815-gfd315e86). ImageJ (RRID:SCR_003070; NIH) and custom macros were used for stereocilia alignment and fluorescence distribution analyses. For longitudinal analysis, "Analysis of stereocilia" macro (https://github.com/janwisn/Analysis_of_stereocilia) performed the following sequence of operations: (1) selection of multiple images of stereocilia parallel to the focal plane (after marking tip location and XY axis orientation) from a set of STED images, (2) creating an average of all selected regions after an alignment (translation and rotation of 5x upscaled images) and noise reduction (median filter), (3) applying FFT, followed by unsharp mask to enhance large-scale pattern, and (4) measurement of signal profile across stereocilium at different distances from its tip. Relative intensity graphs were created in Excel. For cross-sectional analysis, "Analysis of rootlet cross-sections at stereocilia pivot points" macro (https://github.com/janwisn/Analysis_of_rootlet_cross-sections_at_stereocilia_pivot_points) allows (1) selection of regions centered on pivot points (after 5x upscaling) and (2) averaging selected images after slight Gaussian blur (to reduce pixelation artefacts). No noise filters were used. Brightness and contrast of final images were adjusted in Adobe Photoshop 2025 (26.2.0 release) or Corel Photo-Paint 2019 (version 21.3.0.755). No gamma adjustments were used.

### Gene gun transfection of inner ear explants

Gene gun transfection of inner ear explants was performed as described (Belyantseva, 2016; Belyantseva et al., 2003, 2005). In brief, sensory epithelia of the OC were dissected from P0–P3 C57BL/6J WT mice and cultured overnight in DMEM/F12 medium supplemented with 7% FBS at 37°C and in 5% $CO_2$. Epithelial explants were then transfected with EGFP-TPRN, EGFP-TPRN[1–260], and EGFP-TPRN[730–749] using 1-µm gold microcarriers (1652263; BioRad) coated with cDNA and delivered via Helios gene gun–mediated transfection (1652431; BioRad) using helium at 110 psi. After 24 h of transfection, sensory epithelial explants were washed two times in 1X PBS and fixed in 4% PFA for 2 h at RT or overnight at 4°C, permeabilized in 0.1% Triton X-100 for 15 min at RT, and counterstained with TRITC-phalloidin (R415; Thermo Fisher Scientific) or Alexa Fluor 546–phalloidin (A22283; Thermo Fisher Scientific) at 1:100 dilution in 1X PBS for 15–20 min at RT, followed by three washes of 10 min each in 1X PBS. Transfected and counterstained sensory epithelial explants were removed from the culture dish using a fine needle and mounted on a glass slide using Prolong Gold Antifade reagent (Thermo Fisher Scientific). Images were obtained using a Zeiss LSM880 Airyscan confocal microscope equipped with a 63×, 1.4 N.A. objective.

### COS-7 cell culture and transfection

COS-7 cells were obtained from the American Type Culture Collection (CRL-1651, RRID:CVCL_0224; ATCC). Cells were expanded and cultured at 37°C and 10% $CO_2$ in DMEM supplemented with 10% heat-inactivated FBS (Atlanta Biologicals) and GlutaMAX (Thermo Fisher Scientific). To accommodate imaging, COS-7 cells were seeded in glass-bottom culture dishes (#1.5; MatTek, Inc.) and transfected in situ with Lipofectamine 3000 (Thermo Fisher Scientific) using the manufacturer's protocol.

Transfected cells were incubated for 24–48 h before fixation with 4% PFA for 30 min at RT, counterstained with phalloidin (Thermo Fisher Scientific), and observed using LSM880 Airyscan confocal equipped with 63×, 1.4 N.A. objective and Zen software (Zeiss). For live-cell imaging, transfected cells were incubated for 24 h after seeding and then imaged overnight at 37°C and 10% $CO_2$ using PerkinElmer Spinning Disk confocal with temperature, humidity, and $CO_2$ controlled environmental chamber.

### Purification of mouse TRX-TPRN expressed in *E. coli*

Several purification strategies for mouse FL TPRN were tried and were unsuccessful using plasmids expressed in *E. coli* (70235-4; MilliporeSigma), Sf9 insect cells in Sf-900 II SFM (11496015; Thermo Fisher Scientific), and human kidney HEK293 cells (CRL-1573, RRID:CVCL_0045; ATCC). A variety of epitope tags, including GST and MBP, in-frame either at the N terminus or C terminus of TPRN were not helpful. A range of yields of TPRN was obtained, and except for a few expression constructs, TPRN was unstable, and yields were minimal. However, expression in *E. coli* of mouse TPRN protein (RefSeq: NP_780495.2) tagged with TRX (109 aa residues) at the N terminus stabilized recombinant TPRN protein. The coding sequence of mouse *Tprn* (RefSeq: NM_175286) was cloned into the pET-32a vector (69015; MilliporeSigma). The plasmid was then transformed into *E. coli* BL21 (DE3) competent cells (70235-4; MilliporeSigma) and cultured overnight at 37°C in Luria–Bertani media. Cells were induced for 16 h at RT by adding of 1 µM isopropyl-β-D-thiogalactopyranoside following subculturing in fresh media and growth at 37°C for 6 h until the OD at 600 nm reached 0.8. The cells were collected and lysed by sonication using a Qsonica Sonicator for 360 cycles of a 1-s (s) pulse with 4-s intervals between pulses at 60% maximal power. FL and various truncations of TPRN tagged with TRX at the N terminus, and TRX protein alone were purified using Ni-NTA beads (L00223-25; GenScript) according to the manufacturer's instruction. TRX tag removal using thrombin protease (69-022-3; Thermo Fisher Scientific) failed to obtain untagged FL TPRN, suggesting that untagged TPRN is unstable in vitro. Coomassie blue staining was performed using Coomassie Brilliant Blue R-250 (BP101-50; Thermo Fisher Scientific), and silver staining was performed using the Pierce Silver Stain Kit (PI24600; Thermo Fisher Scientific), following the manufacturer's protocols.

### Actin co-sedimentation assays

Low-speed co-sedimentation assays were performed using an Actin Binding Protein Spin-Down Assay Kit following the manufacturer's instruction (BK001; Cytoskeleton, Inc.). In brief, in a 100 µl reaction volume, 3.0 µM G-actin was incubated for 1.5 h at RT with various concentrations of TPRN in 1X actin polymerization buffer (10 mM Tris, pH 7.5, 1 mM ATP, 50 mM KCl, and 2 mM $MgCl_2$). The samples were then centrifuged at 4°C at 13,500 × $g$ for 30 min. 100 µl of supernatant was collected and mixed with 33.3 µl of 4X protein loading buffer (50 mM Tris-HCl, pH 6.8, 8% SDS, 2% bromophenol, and 40% glycerol). The pellet was mixed with 133.3 µl of 1X protein loading buffer

(12.5 mM Tris-HCl, pH 6.8, 2% SDS, 0.5% bromophenol, and 10% glycerol). Equal volumes of pellet and supernatant samples were separated using 10% SDS-PAGE gel and stained with Coomassie Brilliant Blue R-250 (BP101-50; Thermo Fisher Scientific). Using Adobe Photoshop (RRID:SCR_014199), the quantity of actin in the pellet and supernatant was measured.

**Fluorescence microscopy of actin filaments**
Fluorescence microscopy was performed according to a published protocol (Huang et al., 2005). In brief, a 100 µl reaction volume contained 1X actin polymerization buffer (10 mM Tris pH, 7.5, 1 mM ATP, 50 mM KCl, and 2 mM $MgCl_2$), 3.0 µM G-actin (APHL99; Cytoskeleton, Inc), which was incubated for 1.5 h at RT with various concentrations of TPRN, and 5 µM FITC-phalloidin (P5282; MilliporeSigma). The polymerized F-actin was diluted 400 times using fluorescence buffer containing 10 mM imidazole, pH 7.0, 50 mM KCl, 1 mM $MgCl_2$, 100 mM DTT, 100 µg/ml glucose oxidase, 15 mg/ml glucose, 20 µg/ml catalase, and 0.5% methylcellulose. Actin filaments were observed at RT on a DM6 FS automated deconvolution Leica microscope using a HCX PL APO 100×/1.40–0.70 oil objective, DFC3000 G camera, and LAS X life science microscope software (Leica).

**Negative staining of F-actin**
To observe actin filaments and bundles using negative stain TEM, 3 µM of β-actin from human platelets (APHL99; Cytoskeleton, Inc) was polymerized in 100 µl of a solution containing 100 nM TRX, 100 nM TRX-TPRN, 300 nM TRX-TPRN, 300 nM ESPN, or 1 µM ESPN. Using mouse *Espn* cDNA (RefSeq: NM_207687.3), recombinant FL ESPN was subcloned with BamHI and NotI restriction enzymes into a pGEX-6P-1 vector containing a GST tag (28954648; Cytiva). Using the described method (Miyoshi et al., 2021), GST-ESPN was expressed and purified using a bacterial expression system. In brief, GST-ESPN was expressed in BL21 (DE3) cells (EC0114; Invitrogen) and purified using a Glutathione Agarose Resin (P306010; Ubpbio) and PreScission protease (Z02799; GenScript) in a buffer containing 20 mM Tris-HCl, pH 7.4, 150 mM NaCl, and 1 mM DTT.

For incubation with TRN and TPX-TPRN, β-actin was polymerized in a buffer containing 20 mM Tris-HCl, pH 7.4, 50 mM KCl, 50 mM NaCl, 2 mM $MgCl_2$, 1 mM ATP, 1 mM DTT, and 1.5 µM unconjugated phalloidin (P2141; Sigma-Aldrich), of which 10 mM Tris-HCl, pH 7.4 and 50 mM NaCl were derived from the buffer dissolving the TRX-TPRN or TRX. For β-actin incubated with 300 nM and 1 µM ESPN, the polymerization buffer contained 15 and 50 mM NaCl, respectively. After incubating for 1.5 h at RT, 4 µl of the solution was applied to 400 mesh copper grids with a continuous carbon film (CF400-Cu-50; Electron Microscopy Sciences), pre-treated using a Fischione Instruments Model 1020 Plasma Cleaner for 13 s to make them hydrophilic. Grids were incubated for 30 s at RT, washed briefly in distilled water three times, and stained with 0.5% uranyl acetate dissolved in molecular biology grade water (351-029-101; Quality Biological) for 30 s at RT. Images were acquired on a JEM 2100 transmission electron microscope (JEOL) using a OneView Camera (Gatan) at nominal magnifications of 20,000 × and

100,000 ×. Diameters of bundles were measured using the line tool on the Fiji platform (Schindelin et al., 2012) drawing cross sections for each bundle. As described (Volkmann et al., 2001), distances between filaments were measured detecting the center line of each filament using a bandpass filter on the Fiji platform (Schindelin et al., 2012), the profile_line function of scikit-image (https://scikit-image.org/), and the interp1 and find_peaks functions of SciPy (https://scipy.org/). For 100 nM TRX-TPRN, diameters and distances between filaments were measured, avoiding the segments where actin filaments coalesce. For 100 nM TRX, these parameters were measured using pseudo-bundles consisting of at least two filaments running in contact with each other.

**Plasmid constructs**
*Tprn* cDNA (RefSeq: NM_175286.4) was amplified from a mouse inner ear cDNA library. FL and various truncations of *Tprn* were subcloned into pEGFP-N3 vector (Clontech) used for Co-IP experiments or subcloned into an AAV expression vector (# 67634; Addgene) used for AAV production or pET-32a vector (69015-M; MilliporeSigma) used for protein purification. The FL HA-tag-*Tprn* or various truncations of HA-tag-*Tprn* were inserted into an AAV expression vector (# 67634; Addgene), replacing the GFP-coding sequence. A translation start codon and a Myc epitope tag (MEQKLISEEDL; 5′-ATGGAACAAAAACTCATCTCAGAA GAGGATCTG-3′) were added at the 5′ end of *Tprn* cDNA sequence. The FL Myc-tag-*Tprn* or various truncations of Myc-tag-*Tprn* were inserted into pEGFP-N3 vector using XhoI and NotI endonuclease restriction sites, where they replaced the EGFP-coding sequence, and then used in Co-IP experiments. Primers 5′-GGCTCCTCGAGAGCAGCGATGGCGATCTCCTTCAAT-3′ and 5′-ATTGAAGGAGATCGCCATCGCTGCTCTCGAGGAGCC-3′ were used to generate NLS-mutant TPRN (K621A, K622A, and K624A). Platinum SuperFi DNA Polymerase (12351010; Thermo Fisher Scientific) was used for cloning. pAAV-CAG-tdTomato plasmid used to purify AAV-expressing tdTomato was obtained from the Addgene (59462). All plasmids were Sanger sequenced. mCherry-tagged MYO10HMM (residues 1–941; RefSeq: NP_062345) and mCherry-tagged MYO10-TPRN (residues 1–749, RefSeq: NP_780495.2) used in co-transfection experiments with EGFP-PTPRQ were prepared as described previously (Bird et al., 2017) and were deposited at https://www.addgene.org. Plasmid DNA encoding the cytoplasmic intracellular domain of chicken *Ptprq* (RefSeq: XP_040515960.1; bases 7,024–8,184) with EGFP fused to its C terminus was cloned into the backbone of a pEGFP-Actin vector (Clontech) from which the sequences encoding EGFP and β-actin had been removed by restriction digestion with NheI and BamHI.

The EGFP-*Tprn* full-length (RefSeq: NM_175286.4) and corresponding cDNA constructs of *Tprn* fragments used in our gene gun transfection experiments were PCR amplified from postnatal day 5 mouse inner ear cDNA using LA Taq DNA polymerase (Takara Mirus), cloned into XL-TOPO vector (Invitrogen), and transformed into DH5α *E. coli* strain (Invitrogen). A 2,250-bp cDNA product encoding the *Tprn* open reading frame was amplified with Pfu Ultra DNA polymerase (Stratagene) and ligated into the EcoRI and SalI sites of pAcGFP1-C2 vector

(Clontech). All expression plasmids were purified using Qiagen EndoFree Plasmid Maxi Kit (Qiagen). Both strands of the cDNA inserts were verified by Sanger sequencing.

## Co-IP

Co-IP was carried out as described (Krey et al., 2022; Liu et al., 2018). In brief, 2 days after transfection using Lipofectamine 2000 (11668019; Thermo Fisher Scientific), HEK293 cells (CRL-1573, RRID: CVCL_0045; ATCC) were collected and lysed using RIPA buffer at pH 7.4 (50 mM HEPES, 150 mM NaCl, 1% Triton X-100, 1 mM EDTA, and protease inhibitors). After centrifugation at 16,600 × $g$ for 15 min at 4°C, supernatant was collected. Immunoprecipitations were then carried out using anti-Myc (E6654, RRID:AB_10093201; MilliporeSigma) or anti-HA agarose beads (E6779, RRID:AB_10109562; MilliporeSigma). The immunocomplexes were washed with RIPA buffer, resuspended in Laemmli sample buffer, and then analyzed by western blotting. The protein input was detected by blotting HEK293 cell lysate using anti-Myc or anti-HA antibody, followed by ECL detection using Pierce ECL 2 Western Blotting Substrate (PI80196; Thermo Fisher Scientific) and HyBlot Autoradiography film (1159T41; Thomas Scientific). Autoradiography films were then scanned using an Epson Perfection V600 Photo Scanner and analyzed using ImageJ (RRID:SCR_003070; NIH). Each experiment was carried out at least three times. The antibodies used included anti-Myc (sc-40, RRID:AB_627268; Santa Cruz Biotechnology), anti-HA (2367s, RRID:AB_10691311; Cell Signaling Technology), anti-HA (11867423001, RRID:AB_390918; Roche), and anti-EGFP (sc-9996, RRID:AB_627695; Santa Cruz Biotechnology).

## AAV production and injection

AAVs were produced according to a published protocol (Grieger et al., 2006). In brief, pHelper (Cell Biolabs, Inc.), pUCmini-iCAP-PHP.eB (#103005; Addgene) (Chan et al., 2017) and AAV expression plasmids were transfected in an equimolar ratio into HEK293T cells by polyethyleneimine (Chan et al., 2017). Four days after transfection, cells were harvested and lysed. Density gradient ultracentrifugation was performed by loading ~4 ml cell lysis onto 4.5 ml of 60%, 4.5 ml of 40%, 6 ml of 25%, and 8 ml 15% iodixanol layers and then centrifuged at 350,000 × $g$ for 1.5 h at 4°C. Iodixanol in the AAV-containing fractions was removed through buffer exchange using 100-kD Amicon ultra-15 centrifugal filter unit (UFC9100; MilliporeSigma). AAV titer was measured by qRT-PCR using QuantStudio 3 Real-Time PCR system (Thermo Fisher Scientific) and PowerUp SYBR Green Master Mix kit (A25741; Thermo Fisher Scientific). Primers used for qRT-PCR were: 5′-TGTTCCCATAGTAACGCCAAT-3′ and 5′-CCATAAGGTCATGTACTGGGC-3′. The qRT-PCR program was set as 2 min at 95°C, 40 amplification cycles of 1-s denaturation at 95°C, and 30-s annealing and extension at 60°C. A melting curve analysis was performed to verify primer specificity as 1 s at 95°C, 20 s at 60°C, and 1 s at 95°C. AAVs (~ 1 × 10¹⁰ GC) were injected into P2 pups via the posterior semicircular canal according to a published protocol (Isgrig and Chien, 2018).

## Germ line deletion of the mouse *Tprn* using CRISPR/Cas9

Experiments with mice were approved by the NIDCD/NIH Animal Care and Use Committee (ACUC), animal protocol #1263 to TBF, the NEI/NIH ACUC, protocol # NEI-626 to LD, the ACUC at the University of Kentucky, protocol # 2019–3414 to GF, and at the Indiana University School of Medicine, ACUC protocol # 22069 to BZ. A C57BL/6J mouse line with a recessive null allele of the *Tprn* gene (RefSeq: NM_175286.4) was created using CRISPR/Cas9 technology by deleting ~7.5 kb of genomic DNA, including 715-basepairs upstream of a predicted promoter of exon 1, all four exons, including the last protein-coding exon of *Tprn* and 102 bp of the 3′ UTR. The pair of gRNAs (for SpCas9, PAM = NGG) flanked the deleted region of the genomic DNA from the C57BL/6J mouse strain. gRNAs were selected based on their positions near target sites, ranked by the online gRNA selection tool (https://www.crisprscan.org/) and synthesized with T7 in vitro transcription as described (Varshney et al., 2015). gRNAs were tested for in vitro cleavage activity and then in a mouse embryonic fibroblast (MEF) cell line carrying an engineered Tet-inducible Cas9 expression cassette through a Surveyor assay (https://www.idtdna.com/pages/products/crispr-genome-editing/alt-r-genome-editing-detection-kit) for indel-inducing activity as briefly described below. For the in vitro cleavage assay, a genomic PCR product containing the target sites of selected gRNAs was incubated with SpCas9 protein (New England Biolabs) by following the manufacturer's suggested protocol and analyzed on a 2% agarose gel stained with ethidium bromide. Ratios of cleavage bands with expected sizes compared with FL PCR products are measured with a fluorescence scanner and used as an estimate of in vitro cleavage efficiency of the gRNAs tested. gRNAs were further tested for their efficiency in inducing indels at target sites in an immortalized MEF cell culture (Rosa26 Tet-Cas9 MEF) custom engineered to carry a tet-inducible Cas9 expression cassette. Upon confirmation of efficient target cleavage activity in MEF cells, two gRNAs were selected, mixed with SpCas9 protein (PNA Bio), and microinjected into C57BL/6J mouse zygotes as described (Wang et al., 2013). The combination of the two gRNAs used to generate the 7.5-kb deletion allele were *Tprn*_gR_up1 5′-GGG ATACTTGGCAGGACCAT-3′ and *Tprn*_gR_down2 5′-GGGGGA ACTGCCTGTGGGAC-3′. F0 mice (total 13) were screened for the presence of the 7.5-kb deletion by PCR using the following primers: forward primer 5′-GACAGGGTTTCTCTGTATAGCCCT-3′ and reverse primer 5′-CCTTGTTACAGTGCTGTTTATTGA-3′. Among all F0 mice screened, one F0 mouse (Carl-614) out of 13 F0 mice screened had a single PCR product of 493 base pairs indicative of the desired deletion allele, while others had additional bands of 600–700 base pairs in size indicating more complex mosaic patterns of genomic insertion and deletions (Indel) at the target site. PCR products from five F0 mice were cloned into pCR4-TOPO TA vector (Invitrogen) vector, and eight individual clones from each individual TA cloning experiment were sequenced for confirmation. Five different deletions (chr2: 25,269,896-25,261,908; chr2: 25269519-25262042; Chr2: 25269519-25262053; Chr2: 25269519-25261884, and Chr2: 25269616-25262044) were uncovered ranging in size from 7.466 to 7.988 kb. However, no partial deletion of the ORF was detected indicating genome editing was successful for both gRNAs in each instance and resulted in a deletion of the entire region between both gRNAs. All deletion alleles included all

protein-coding exons of the annotated *Tprn* gene (RefSeq: NM_ 175286.4), and the largest deletion allele included 720 flanking bases pairs. The 7.5-kb deletion allele included the whole *Tprn* open reading frame and the 3′UTR but retained the initial 368 bp of the 5′UTR from male founder #Carl-614 and was transmitted to the germline and selected for further studies. Male founder #Carl-614 was mated with a C57BL/6J female mouse (Jackson Laboratory) for germline transmission of the 7.5-kb deletion allele. Offspring from the germline transmission mating were then genotyped and mice carrying the desired deletion allele were selected and further backcrossed to C57BL/6J mice for at least eight generations. The resulting mouse strain was named according to JAX nomenclature rules as *C57BL/6J-Tprn^{emlibTBF}* but is here referred to as *Tprn^{-/-}*. To genotype the *Tprn^{-/-}* mouse, PCR primers were used to amplify the WT allele: 5′-AGCTCACACCTGCCAGTCAGAAC-3′ and mutant allele 5′-TGCAGCGATACCCAAAGAAACTT-3′ in combination with a common reverse primer 5′-CCAGACCAGCAAGCTCAAAGT GT-3′. The WT and mutant alleles were amplified by separate PCR reactions. Touchdown PCR from 65 to 54°C were run as follows: initial denature for 2 min at 94°C, then 10 cycles of 95°C for 15 s, 65°C for 30 s, and extending for 1 min 20 s at 72°C, where starting at cycle 2, the annealing temperature decreased by 1°C each cycle; then denaturing at 95°C for 15 s, annealing at 54°C for 30 s, and extending for 1 min 20 s at 72°C for 28 cycles; followed by a final extension of 10 min at 72°C. Using these cycling parameters, the WT allele is amplified at 655 bp and the mutant allele is amplified around 922 bp.

The *Tprn^{inl03/inl03}* mouse model was genotyped as described by Liu et al. (2018). In brief, the primers 5′-CTGGAAACGGGA GATCCTTG-3′ and 5′-GAAGCCTGGCGCTGACTC-3′ were used to amplify both WT and mutant alleles using Taq DNA Polymerase (New England Biolabs). PCR was performed with the following program: initial denaturation at 94°C for 5 min, followed by 35 cycles of 94°C for 30 s, 60°C for 30 s, and 68°C for 50 s, with a final extension at 68°C for 5 min. The WT allele amplimer is 274 bp and the mutant amplimer is 377 bp. *Tprn^{N259}* mice were derived from KOMP2566 clone (University of Michigan Transgenic Animal Model Core). The *Tprn^{N259}* mouse has a partial deletion of *Tprn* retaining sequence encoding an N-terminal portion from residues 1–259 (RefSeq: NP_780495.2). Although there is no TPRN detected in *Tprn^{N259}* mice using antibodies shown in Fig. 1 B, we cannot exclude the possibility that the N-terminal fragment is translated as a truncated peptide of TPRN. *Tprn^{N259}* mice were backcrossed to C57BL/6J (Jax Lab) for more than eight generations in the NIDCD animal facility. A multiplex PCR assay using a WT primer 5′-TGAACACCCAGATCACCAAA-3′ and mutant allele 5′-GACCTGCAGCCCCTAGATAA-3′ with a common reverse primer 5′-AGGGCTCCTGGGTATTGACT-3′ were used in the same PCR reaction to amplify the mutant allele at 126 bp and WT allele at 233 bp. A quick two-step thermocycler program was run to amplify these as follows: initial denature for 3 min at 94°C, then denature for 20 more seconds at 94°C, annealing for 20 s at 60°C for 35 cycles, and a final extension step for 3 min at 72°C. Since these two mouse models were used here mainly for data comparison with our complete knockout of *Tprn* gene (*Tprn^{-/-}*), the hearing phenotypes represented by auditory

brainstem response (ABR) data for *Tprn^{-/-}*, *Tprn^{N259/N259}*, and *Tprn^{inl03/inl03}* mouse models are shown in Fig. S5, A–C.

## Comprehensive phenotyping numerous tissues from *Tprn^{-/-}* mice and controls

Fifty-three mouse tissues and mouse skeletons from males and females of the three genotypes (*Tprn^{+/+}*, *Tprn^{+/-}* and *Tprn^{-/-}*) were examined after tissue sections on slides were prepared for histology. Phenotyping using conventional light microscopy after hematoxylin-eosin staining included examination of middle and inner ears (for each temporal bone, 15 step sections were cut at 20-μm interval), skin, tongue, pancreas, salivary glands, lacrimal glands, lymph nodes, thymus, adrenal glands, thyroid glands, pituitary gland, peripheral nerves, eyes, stomach, small intestine, cecum, colon, rectum, anus, esophagus, trachea, lungs, liver, gallbladder, spleen, kidneys, heart, the female reproductive tract (ovary, fallopian tube, uterus, cervix, and vagina) or the male reproductive tract (bulbourethral gland, preputial glands, penis, seminal vesicles, coagulating glands, prostate gland, ductus deferens, epididymis, and testes), urinary bladder, brain, spinal cord, spinal column, tail, head (oral cavity, nasal cavity, and skull), and the front and hind limbs, including joints, bones, and skeletal muscles.

## ABR, DPOAE, and VsEP measurements

Auditory function was evaluated using the ABR and DPOAEs at 18, 30, and 60 days of age. Mice were anesthetized via an intraperitoneal injection of a solution of ketamine (56 mg/kg) and dexmedetomidine (0.375 mg/kg). DPOAEs and ABRs were measured using Tucker-Davis Technologies instruments (TDT) hardware (RZ6 Multi I/O processor, MF-1 speakers) and software (BioSigRz, v. 5.7 or 5.7.1). To maintain body temperature, mice were placed inside a sound-proof booth (Acoustic Systems) resting on a heating pad at 37°C connected to a temperature controller and rectal probe (ATC-2000; World Precision Instruments). DPOAEs were recorded using an ER-10B⁺ microphone (Etymotic) coupled to the mouse's ear using a modified 10-μl pipette tip trimmed to 10 mm in length and 2 mm in the diameter at the tip opening. DPOAE ($2f_1–f_2$) levels were obtained in response to tone pairs with the lower tone (f1) at 65-dB sound pressure level (SPL) and the higher tone (f2) at 55-dB SPL, with f2 varied from 4 to 40 kHz (5 points/octave) and f2/f1 = 1.25. The mean noise floor was calculated from three points sampled above and three points sampled below the DPOAE. DPOAE signals were converted to dB SPL according to the calibration of an ER-10B⁺ microphone provided by the manufacturer. ABR thresholds were determined by presenting Blackman-gated tone-burst stimuli (3-ms duration, 29.9/s, alternating polarity) at 8, 16, and 32 kHz. Subdermal needle electrodes were placed at the cranial vertex and beneath each pinna, with the non-test ear serving as the ground. Stimuli were delivered via a TDT MF-1 speaker in a closed-field configuration. PVC tubing provided with the speaker was coupled to the ear using a modified pipette tip of the same dimensions as the one used with the DPOAE microphone. Responses were amplified (20×), filtered (0.3–3 kHz), and digitized at 25 kHz. Thresholds were determined through visual inspection of stacked waveforms (average of

512–1,024 artifact-free responses per waveform). Stimuli were presented first at 80-dB SPL and then decreased in 10- to 20-dB steps until the ABR waveform disappeared. The stimulus was then increased and decreased in 5-dB steps until the lowest stimulus level that produced a repeatable waveform was determined, which was designated as the ABR threshold. At least two waveforms (1,024 responses each) were obtained for stimulus levels at and near the ABR threshold to ensure repeatability of the response. If no response was evident at 80-dB SPL, the stimulus level was increased to 90-dB SPL. When no repeatable ABR waves were detectable at the highest stimulus level tested (90-dB SPL), the threshold was designated as 100-dB SPL for subsequent analyses.

VsEPs were obtained as previously described (Tona et al., 2019). Subcutaneous electrodes were placed at the nuchal crest, at the left ear, and at the left hip. Mice were placed in the supine position on the heating pad with the head placed in a custom, noninvasive head clip attached to an electromechanical shaker (ET 132-2; Labworks, Inc.). The shaker delivered linear pulses (2-ms duration, 17 pulses per second) to the head in the naso-occipital axis. An accelerometer attached to the shaker was used to calibrate the stimulus in jerk (i.e., first derivative of acceleration) at the start of each test session. Stimuli were delivered in 3-dB steps from +6 dB to –18 dB re:1.0 g/ms (1.0 g = 9.8 m/s2). Four waveforms were measured at each stimulus level: two in the positive (occipital toward nasal) and two in the negative (nasal toward occipital) direction (256 responses per waveform). Pairs of the positive and negative waveforms were averaged together to produce two averaged waveforms for analyses. The two averaged waveforms were superimposed and compared to ensure reproducibility. To ensure no contribution of an auditory response to the waveform, testing began with presentation of the highest stimulus level (+6 dB), both in quiet and in the presence of a masking noise (50–50,000 Hz, 90-dB SPL). If the unmasked response peak amplitude changed by >50% in the presence of the masker, the test was performed with the masker on. We defined the threshold measured as the stimulus level (dB) halfway between the lowest stimulus level producing a response and the next stimulus level at which no response was measured.

### Statistical analysis

We used a linear mixed effects regression model to assess the effect of manipulations of the *Tprn* gene on hearing and vestibular function in mice. All models included a random effect for subjects and fixed effects for age, genotype, the interaction of age and genotype, and frequency (only for ABR threshold models). The youngest age of testing *Tprn*[+/+] mice and the lowest frequency tested were set as reference levels for their respective variables. Post hoc pairwise comparisons used Tukey's adjustment for multiple comparisons. For post hoc analyses assessing hearing effects at specific frequencies, a separate model was fit for measurements of ABR threshold at each frequency. In the case of *Tprn*[in103/in103], individual mice were only assessed at a single time point with the control group only being assessed at P60. Consequently, we fitted a mixed effects linear regression model with a random effect for subject ID and a fixed effect for

genotype/age group and frequency that compared ABR threshold from the three distinct *Tprn*[in103/in103] age groups to that of *Tprn*[+/+] mice at P60. Linear mixed effects models were run in R version 4.4.2 (https://www.r-project.org/) using the lme4 and emmeans packages. All graphs display means ± SD.

### ISH using RNAscope probes

To reveal the expression of *Tprn* mRNA in different inner ear cell types, ISH was performed using RNAscope probes Mm-Tprn-01 (target region: 72–912 nucleotides, RefSeq: NM_175286.4), Mm-Tprn-02 (target region: 1,701–2,655 nucleotides, RefSeq: NM_175286.4), Probe-Mm-Myo7a-C2 (target region: 1,365–2,453 nucleotides, RefSeq: NM_001256081.1), and Probe-Mm-Cldn11-C3 (target region: 214–1,156 nucleotides, RefSeq: NM_008770.3) (Advanced Cell Diagnostics). Cochleae from *Tprn*[−/−] mice and WT littermate at P3 were fixed in 4% PFA in 1X PBS overnight at 4°C, cryopreserved in 15% sucrose in 1X PBS overnight, and then in 30% sucrose overnight at 4°C. Each cochlea was embedded and frozen in Super Cryoembedding Medium (Section-lab). Cryosections of 12-μm thickness were obtained using a Leica CM3050S cryostat microtome. An RNAscope Multiplex Fluorescent v2 assay (Advanced Cell Diagnostics) was used to visualize RNA molecules, and images were obtained using a Zeiss 40×, 1.4 N.A. objective on a Zeiss LSM880 confocal microscope.

### EP measurements

Mice were anesthetized with 2,2,2-tribromoethanol (T4842; Sigma-Aldrich) at 0.35 mg/g body weight. EP measurements were made using glass microelectrodes made of borosilicate glass capillaries (1B100F-4; Kwik-Fil) pulled using Sutter P97 puller to have an inner one-micron diameter opening. The microelectrode was filled with 1 M KCl for conductivity and inserted into the round window and through the basilar membrane of the first turn of the mouse cochlea. For EP measurements, males and females were used for each genotype. Anoxic-state EP was accomplished by intramuscular injection of succinylcholine chloride (0.1 μg/g, NDC-0409-6629-02; Pfizer) after establishing deep anesthesia, followed by an injection of 2,2,2-tribromoethanol. Anoxic-state EP is an indicator of the sensory hair cell function. When hair cells are functional, the anoxic-state EP is negative, whereas the EP becomes 0 if the hair cells are not functional or dead. Data were recorded digitally (Digidata 1440A and AxoScope 10; Axon Instruments) and analyzed using Clampfit10 (RRID: SCR_011323; Molecular Devices).

### Western blot analyses

Inner ears from four mice of each of three genotypes (*Tprn*[+/+], *Tprn*[+/−], and *Tprn*[−/−]) were dissected. The tissue was flash frozen in liquid nitrogen and pulverized in a CP02 Automated Dry Pulverizer (Covaris). The crushed powered tissue was solubilized in 50 mM Tris-HCl, pH 8.5, 1% Igepal (non-denaturing detergent), 1 mM EDTA, 0.1 mM PMSF, 1% N-ethylmaleimide, and 10 mM iodoacetamide separately for each genotype. Samples were centrifuged at 20,000 × *g*, 30 min at 4°C, and the supernatant was collected. Protein concentration was measured at OD_{280} using a NanoDrop One[C] spectrophotometer (Thermo Fisher Scientific). Each protein sample was diluted to 1 μg/μl.

Then, 15 µl of supernatant of each genotype was mixed with 15 µl of 2x Laemmli Sample Buffer (1610737; BioRad) to load one lane (15 µg protein/lane). Electrophoresis and western blotting were performed as described (Sanchez-Laorden et al., 2006) with some modification. Briefly, 4–20% Mini-PROTEAN TGX Precast Protein Gels (Bio-Rad) were used and electrophoresis was performed at 200V for 45 min. Bio-Rad Trans-Blot Turbo Transfer System and Trans-Blot Turbo transfer packs with 0.2-µm PVDF membrane were used with the default program for mixed molecular weight protein transfer with a run duration of 7 min. To block further binding of proteins to the membrane, as well as to dilute the primary and secondary antibodies, an Invitrogen iBind Flex Solution Kit (Invitrogen) was used according to the manufacturer instructions. An iBind Automated Western System performed the transfer of proteins from gel to the membrane. The membrane was then incubated in a custom made RabmAb TPRN primary antibody to the mouse TPRN epitope KEVMLTPASQNDLSDFRSEPALYF (residues 726–749, RefSeq: NM_175286.4; Fig. 1 B). The RabmAb was diluted 1:200 in blocking solution, and the membrane was incubated for ~3 h at RT, washed several times in wash buffer, and then incubated in HRP-conjugated goat anti-rabbit secondary antibody (A0545, RRID:AB_257896; Sigma-Aldrich) diluted 1:10,000 in blocking buffer. Similar loading of protein in each well was verified by stripping and probing the blots with an anti–β-actin antibody (A5441, RRID:AB_476744; Sigma-Aldrich) as a loading control. Pierce ECL Western Blotting Substrate (Thermo Fisher Scientific) was used to incubate the blot at RT for 1 min. Excess liquid was drained off the membrane, and an image of the membrane was obtained using a Bio-Rad ChemiDoc MP (Image Lab Touch Software version 3.0.1.14).

### Quantification of immunofluorescent puncta using Imaris and ImageJ

The images were analyzed using the Imaris software version 9.8 (Oxford Instruments). The spot function was used to label each punctum in the ImageJ. The number of puncta as well as the mean intensity of each punctum in each hair cell were calculated. The average of the mean intensities was calculated for each cell and compared between the genotypes using unpaired *t* test. Mean intensity of hair cells puncta was measured using ImageJ. The "set measurement" function in Fiji was setup to limit mean gray value measurements to threshold. This allows the software to examine the mean intensity at specified areas with fluorescent signal and ignoring the regions without signal. With this, the mean intensity value for the puncta (collectively) in each hair cell was calculated and compared within the genotypes using unpaired *t* test.

### TEM

The temporal bones of mice were dissected out, and cochleae were fixed overnight at 4°C in a combination of 2.5% glutaraldehyde and 1.6% PFA diluted in 0.1 M cacodylate buffer supplemented with 3.4% sucrose and 2 mM $CaCl_2$. The OC was locally perfused through the oval and round windows, microdissected, and washed in 0.1 M cacodylate buffer and in distilled water. After decalcification in 2.5% EDTA in 0.1 M cacodylate buffer with supplements, cochleae were either processed as a whole or bisected to expose OC or microdissected to isolate OC apical, middle, and basal turns. Postfixation in 1% osmium tetroxide was followed by dehydration in ascending ethanol series starting from 35 to 50%, staining with 1% uranyl acetate in 50% ethanol, and continued with dehydration in 65%, 75%, 85%, 95%, 99%, and 100% ethanol, followed by propylene oxide and embedding in Epon LX-112 (21210 - LX 112; Ladd Research) or EMbed 812 kit, (EMS 14120; Electron Microscopy Sciences) or PolyBed812 (08792; Polysciences, Inc.). Thin 60–80-nm sections were prepared using Ultracut UC7 (Leica), stained with 1% uranyl acetate and 0.3% lead citrate, and examined in JEM 2100 transmission electron microscope (JEOL) using a OneView Camera (Gatan). All reagents, except where indicated, are from Electron Microscopy Sciences.

### FIB-SEM

Cochleae were extracted from temporal bones and gently perfused through the oval window with a solution containing 2.5% glutaraldehyde, 2% PFA in 0.1 M cacodylate buffer, pH 7.4 (15960-01; Electron Microscopy Science) supplemented with 2 mM $CaCl_2$ and 1% tannic acid (21710; Electron Microscopy Sciences). The cochleae were kept in this fixative overnight at 4°C, and then the fixative was diluted 1:5 with 0.1 M cacodylate buffer for storage at 4°C until dissection. OC were dissected in distilled water. Then, they were either high-pressure frozen (Leica EM ICE) or plunge frozen (Leica EM CPC) in liquid $N_2$. The frozen samples were freeze substituted and stained in the Leica EM AFS2 machine using the following protocol: (1) 33-h incubation at –90°C in 100% methanol supplemented with 1% uranyl acetate; (2) slow warming at a speed of 4°C/h to –45°C in the same methanol/uranium acetate solution; and (3) 24 h incubation at –45°C in 100% methanol. After freeze substitution, the samples were low-temperature embedded in Lowicryl HM-20 resin (14345; Electron Microscopy Sciences) by gradually increasing Lowicryl concentration in methanol to 50% for 2 h, 75% for ~21 h, and 100% for ~24 h. To ensure that there was no residual methanol, the samples were additionally incubated with fresh 100% Lowicryl for 1 h, transferred into a flat mold with 100% Lowicryl, and incubated for additional 24 h. Then, the samples were polymerized with UV light at –45°C for 27 h, at 0°C for additional 40 h, and at 20°C for 48 h. Resin blocks were mounted on low profile 45/90° SEM holder, trimmed to reach a desired sample distance of 20–50 µm from the upper surface of the block (Leica UC7 ultramicrotome), and sputter coated with platinum (91016-PT; Electron Microscopy Sciences) at a thickness of 25 nm (EMS150T ES; Electron Microscopy Science). The samples were imaged with a backscattered electron detector of a field-emission scanning electron microscope (Helios Nanolab 660; FEI) equipped with a FIB. Serial sections of 20 nm were milled by FIB from the block face while alternating imaging with the electron beam with a resolution of 1.5–2.0 nm/pixel.

### Rootlet diameter analysis

FIB-SEM image analysis was conducted in Fiji (Image J) software by importing all image slices into tiff stacks. Minimum intensity

projections of varying thickness (~500–1,000 nm) were made to view the entire rootlet of each stereocilium in multiple rows, but before adjacent stereocilia would begin to overlap. Then, a 300-nm–long line was drawn perpendicular to the rootlet at a distance of 100 nm below the pivot point of a stereocilium, and intensity profile was obtained along this line. Since rootlets are cylindrical in nature, the minimum intensity projection makes a rootlet appear darkest in the center and lighter along the edges. Thus, the intensity profile across the rootlet nicely follows a Gaussian distribution. Intensity profiles from each rootlet were exported into Origin Pro 2023b software (OriginLab Corporation) and fit with a Gaussian curve. Rootlet diameters were determined as the width of the fitted Gaussian curves (one standard deviation, or width at 66%) and plotted for each row, grouped by genotype. One-way ANOVA with Bonferroni post hoc *t* tests were conducted between control and mutants, separately for each stereocilia row.

### OC explants for whole-cell patch-clamp recordings
OC explants were prepared from acutely dissected apical turns of the cochlea from $Tprn^{+/+}$, $Tprn^{+/N259}$, and $Tprn^{N259/N259}$ littermate mice at P4–P7. The explants were secured in a custom-made glass-bottom chamber with two strands of dental floss as previously described (Galeano-Naranjo et al., 2021). Briefly, explants were transferred to a custom chamber made from glass-bottom Petri dishes (Cat # 70673-02; Electron Microscopy Sciences). The walls of these dishes were cut by half to ~3.5 mm. Stack of two ~10-mm square plastic cover slips and ~0.5-mm wires were mounted to the glass-bottom with a silicon glue for holding the explant on top of plastic cover slips with two strands of dental floss. The chamber was filled with Leibovitz-15 cell culture medium (11415064; Thermo Fisher Scientific) for whole-cell patch-clamp recordings.

### Whole-cell patch-clamp recordings
OC explants were visualized using an upright light microscope (BX51WI; Olympus) with a 100× 1.00 N.A. water immersion objective. Perfusion of Leibovitz-15 medium (11415064; Thermo Fisher Scientific) was performed with a pipette placed near by the tissue, and fresh medium allowed access to the tissue over the course of the experiment. A micropipette with a tip diameter of ~4 μm containing Leibovitz-15 cell culture medium was used to make a hole in the freshly isolated OC tissue to gently separate out rows of hair cells in preparation for whole-cell patch-clamping. Patch pipettes were made from borosilicate glass (1B100F-4; WPI) and typically held resistances of 3–7 MΩ in the bath, with an intrapipette solution containing 147 mM CsCl, 2.5 mM MgCl$_2$, 1 mM EGTA, 2.5 mM K$_2$-ATP, and 5 mM HEPES in distilled water. All whole-cell patch-clamp recordings were conducted at RT (20–23°C). MET currents from auditory hair cells were recorded at a sampling rate of 250 kHz using an Axon MultiClamp 700B amplifier, Axon DigiData 1400B digitizer, and pClamp 10 software (Molecular Devices). MET currents were filtered using a low-pass, 10-kHz analog filter (Bessel type, Stanford Research Systems). Hair cells were held at −60-mV resting potential and stepped to −90 mV during deflections of the bundles. Since TPRN deficiency could make the hair bundles

more fragile resulting in increased cell-to-cell variability in the MET current amplitudes, our criteria for a "technically successful" recording did not include a minimal MET current. We included in statistical analysis all recordings from hair cells that had uncompensated series resistance of <25 MOhm and membrane resistance of >250 MOhm (determined by pClamp "membrane test").

### Fluid-jet deflection of stereocilia bundles
Stereocilia bundles were deflected with fluid-jet (Fig. 9 A) driven by the first generation (analog) high-speed pressure-clamp system (ALA Scientific Instruments) that was controlled by pClamp software. Fluid-jet pipettes were pulled from a thin-wall borosilicate capillaries (TW150F-4) with P-1000 Flaming/Brown style micropipette puller (Sutter Instruments). Despite all our efforts to obtain the pipettes with identical diameters, environmental factors (such as RT and humidity) resulted in variability of the tip diameters of 3.8–4.6 μm in different experiments. The fluid-jet pipette was filled with Leibovitz-15 cell culture medium. Before starting recordings, the fluid-jet pipette was positioned in front of a small debris, and holding pressure was adjusted to eliminate any movement of the debris. This procedure compensates for the capillary force in the pipette and ensures that the steady-state pressure at the tip of the fluid-jet is zero, so the hair bundle remains in its resting position when not stimulated. Before the experiment, the fluid-jet was positioned in front of a hair bundle at distance of ~4.5 μm and in such a way that the diameter of the pipette was aligned in Z axis with the image of the tip of the bundle (see Fig. 9 A, inset). This ensured Z positioning of the fluid-jet with an error of ±340 nm, the half-width of the point spread function of the objective in Z direction. During the experiment, high-speed video recordings of the bundle movements always included the image of the fluid-jet pipette (as shown in Fig. 9 A, inset), which allow us to measure the actual distance between the pipette and the bundle (3.6–5.0 μm throughout all experiments). As we previously reported, the force generated by this fluid-jet depends linearly on the applied pressure (Kitajiri et al., 2010).

### Measurements of stereocilia bundle deflections and mechanical compliance
Time-lapse video recordings of stereocilia movements at ~2,000 frames/sec were quantified using a custom-written MATLAB script. First, a rectangular region of interest covering a hair bundle was manually chosen. Then, at each X point, the intensity profiles along Y axis were plotted, interpolated 50 times to allow sub-pixel resolution, and the position of the peak (a stereocilium) was determined. Then, frame-by-frame sub-pixel shifts of the peaks of intensity profiles were determined at each X point of the region of interest. At the end, all these frame-by-frame shifts of intensity profiles were averaged together to obtain an average displacement of stereocilia in Y direction. Note that our fluid-jet has been always oriented in Y direction. Hair bundle compliance (reciprocal to the stiffness) was calculated from linear fits of relationships between hair bundle displacement and fluid-jet pressure around resting bundle positions (0–100-nm deflections). Within variabilities of fluid-jet stimulation parameters in our

experiments, we did not find a correlation of hair bundle compliance with the fluid-jet pipette-to-bundle distance (Pearson coefficient = 0.39, P = 0.08) but did find a strong correlation with the area of the fluid-jet pipette (Pearson coefficient = 0.66, P = 0.001). Therefore, the hair bundle compliance was presented as a function of the fluid-jet area (Fig. 9 G).

## Online supplemental material

Table S1 shows the reported pathogenic or likely pathogenic variants of human *TPRN* and associated phenotypes. Table S2 shows the statistical analysis of ABR data for $Tprn^{-/-}$, $Tprn^{+/-}$, and $Tprn^{+/+}$ mice. Table S3 shows the comparisons between time points of $Tprn^{-/-}$, $Tprn^{+/-}$, and $Tprn^{+/+}$ ABR data within genotypes. Table S4 shows the statistical analysis of $Tprn^{-/-}$, $Tprn^{+/-}$, and $Tprn^{+/+}$ ABR data at each frequency. Table S5 shows the comparisons of $Tprn^{-/-}$, $Tprn^{+/-}$, and $Tprn^{+/+}$ ABR data between genotypes at each frequency and time point. Table S6 shows the statistical analysis of ABR data from $Tprn^{N259/N259}$, $Tprn^{+/N259}$, and $Tprn^{+/+}$ mice. Table S7 shows the comparisons between time points of $Tprn^{N259/N259}$, $Tprn^{+/N259}$, and $Tprn^{+/+}$ ABR data within genotypes. Table S8 shows the statistical analysis of ABR data from $Tprn^{in103/in103}$ and $Tprn^{+/+}$ mice. Table S9 shows the comparisons of ABR data from $Tprn^{in103/in103}$ and $Tprn^{+/+}$ mice at each frequency between genotype and age groups. Table S10 shows the statistical analyses of $Tprn^{-/-}$, $Tprn^{+/-}$, and $Tprn^{+/+}$ VsEP threshold, P1 amplitude, and P1 latency measures for each genotype and time point. Table S11 shows the comparisons of $Tprn^{-/-}$, $Tprn^{+/-}$, and $Tprn^{+/+}$ VsEP threshold, P1 amplitude, and P1 latency between genotypes at specific time points. Fig. S1 shows the validation of anti-TPRN antibodies used in the study. Fig. S1 also shows the purified TRX-TPRN purity and instability without TRX tag and additional examples of Co-IP of TPRN and actin. Fig. S2 shows the T*prn* mRNA expression in WT and *Tprn* null mouse OC and SV using RNAscope probes. Fig. S3 shows the histological evaluation of the OC progressive degeneration in $Tprn^{-/-}$ mice. Fig. S4 shows that the SV morphology and EP are normal in $Tprn^{-/-}$ mice. Fig. S5 shows the ABR thresholds of $Tprn^{-/-}$, $Tprn^{N259/N259}$, and $Tprn^{in103/in103}$ mice and VsEP measurements in $Tprn^{-/-}$ mice. ANKRD24, TRIOBP-5, GPSM2, and ESPN localization in WT and $Tprn^{in103/in103}$ postnatal hair bundles and hair bundle morphology of TPRN-deficient mice. Video 1 shows that the overexpression of EGFP-TPRN in a COS-7 cell induces F-actin-bundle formation in the nucleus.

## Data availability

Original data, proteins, plasmids, and antibodies are available from the corresponding authors upon reasonable request.

# Acknowledgments

We are grateful to Jim McGehee, Sherly Michel, and Alexander Callahan for their help with mouse breeding and genotyping. We thank Mhamed Grati and Kevin Isgrig for critical reading of our manuscript and helpful suggestions, Dr. Mary Grace Velasko and John Waka for assistance with STED imaging using Abberior Facility Line microscope, and Dr. Erich Boger and Mr. Spencer M. Goodman for technical assistance.

This research was supported (in part) by the Intramural Research Program of the National Institutes of Health (NIH), National Institute on Deafness and Other Communication Disorders (NIDCD) DC000039 to TBF. This research was also supported by NIDCD/NIH grants: R01DC017147 and R01DC018785 (to B. Zhao) and R01DC014658 and R01DC019054 (to G. Frolenkov). This study is also partially supported by funding to the Mouse Auditory Testing Core, NIDCD (ZIC DC-000080), and NIH grant S10OD025130 for FIB-SEM study (to G. Frolenkov) performed at the University of Kentucky Electron Microscopy Center at the National Science Foundation NNCI Kentucky Multiscale Manufacturing and Nano Integration Node, supported by ECCS-1542174.

Author contributions: I.A. Belyantseva: conceptualization, data curation, formal analysis, investigation, methodology, project administration, validation, visualization, and writing—original draft, review, and editing. C. Liu: conceptualization, formal analysis, investigation, methodology, visualization, and writing—original draft, review, and editing. A.K. Dragich: conceptualization, formal analysis, investigation, methodology, resources, visualization, and writing—original draft, review, and editing. T. Miyoshi: formal analysis, investigation, visualization, and writing—review and editing. S. Inagaki: conceptualization, formal analysis, investigation, methodology, and writing—review and editing. A. Imtiaz: formal analysis, methodology, and writing—review and editing. R. Tona: data curation, formal analysis, validation, visualization, and writing—review and editing. K.S. Zuluaga-Osorio: data curation, formal analysis, investigation, software, and validation. S. Hadi: investigation. E. Wilson: data curation, formal analysis, investigation, resources, visualization, and writing—review and editing. E. Morozko: investigation and writing—review and editing. R. Olszewski: investigation. R. Yousaf: investigation, methodology, and writing—review and editing. Y. Sokolova: investigation. G.P. Riordan: investigation and methodology. S.A. Aston: formal analysis, visualization, and writing—review and editing. A.U. Rehman: conceptualization and writing—review and editing. C. Fenollar Ferrer: investigation and writing—review and editing. J. Wisniewski: investigation, resources, software, visualization, and writing—review and editing. S. Gu: software. G. Nayak: resources and writing—review and editing. R.J. Goodyear: investigation and visualization. J. Li: methodology and writing—review and editing. J.F. Krey: resources and validation. T. Wafa: investigation and writing—review and editing. R. Faridi: data curation, investigation, resources, validation, and writing—review and editing. S.M. Adadey: investigation and writing—review and editing. M. Drummond: conceptualization, investigation, methodology, and writing—review and editing. B. Perrin: resources. D.C. Winkler: data curation, investigation, resources, visualization, and writing—review and editing. M.F. Starost: investigation and resources. H. Cheng: formal analysis and writing—review and editing. T. Fitzgerald: formal analysis, investigation, visualization, and writing—review and editing. G.P. Richardson: investigation, supervision, and writing—review and editing. L. Dong: methodology. P.G. Barr-Gillespie: supervision and writing—review and editing. M. Hoa: data curation, formal analysis, funding acquisition, investigation, methodology,

supervision, visualization, and writing—review and editing. G.I. Frolenkov: conceptualization, data curation, formal analysis, funding acquisition, methodology, project administration, supervision, validation, visualization, and writing—original draft, review, and editing. T.B. Friedman: conceptualization, funding acquisition, methodology, project administration, resources, supervision, and writing—review and editing. B. Zhao: conceptualization, data curation, formal analysis, funding acquisition, investigation, methodology, project administration, resources, supervision, validation, visualization, and writing—original draft, review, and editing.

Disclosures: The authors declare no competing interests exist.

Submitted: 5 August 2024

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

# Supplemental material

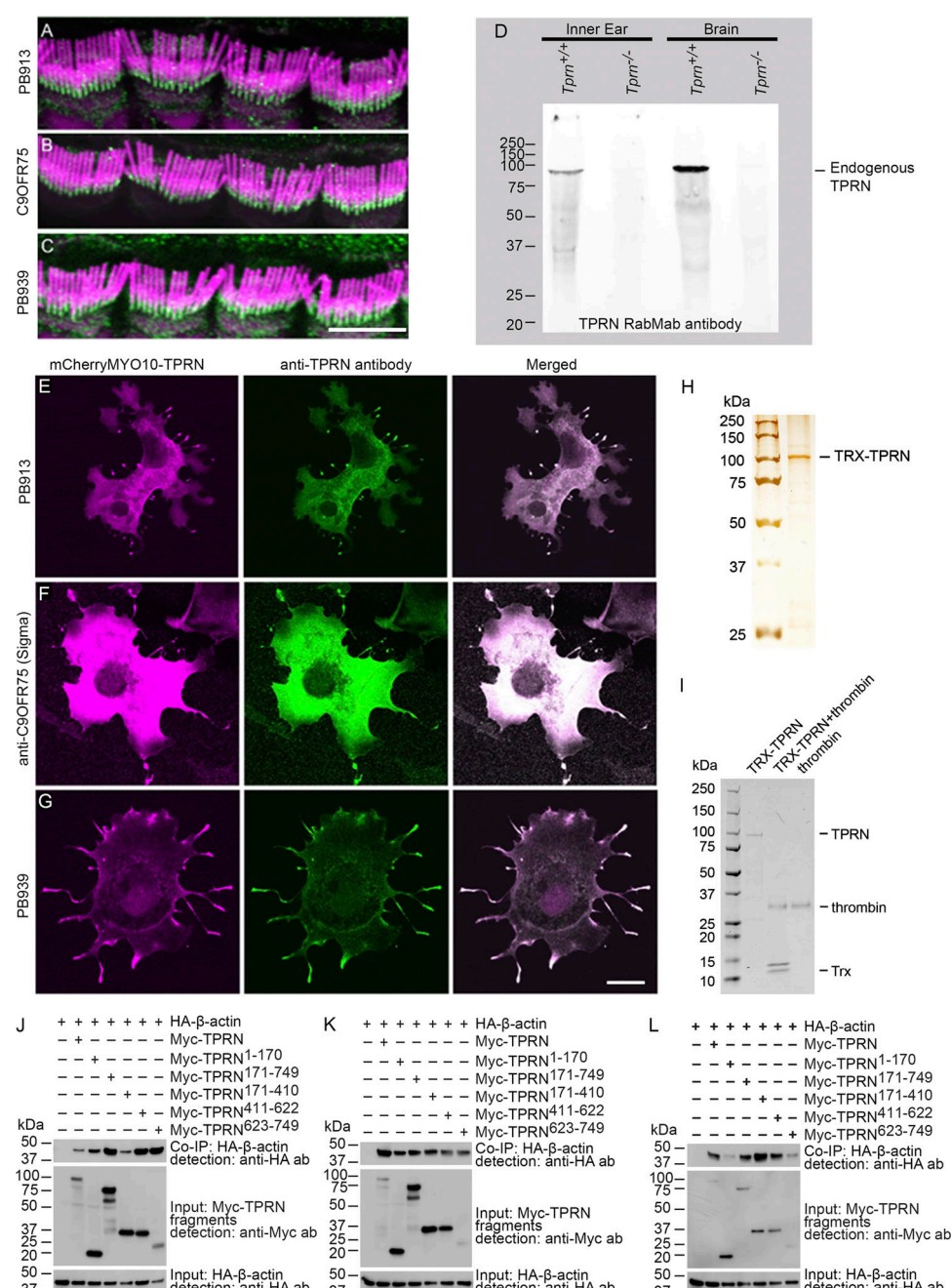

Figure S1.  **Validation of anti-TPRN antibodies used in the study. TRX-TPRN purity and instability without TRX tag and additional examples of Co-IP of TPRN and actin. (A)** Custom-made rabbit polyclonal PB913 antibody against a peptide corresponding to aa residues 516–535 encoded by exon 1 of mouse *Tprn* shows localization of TPRN (green) at the base of hair cell stereocilia of adult mouse IHCs. **(B)** A commercial anti-C9ORF75 (TPRN) antibody (HPA020899, RRID:AB_1845835, aa 446–517, MilliporeSigma), shows localization of TPRN (green) similar to the custom-made PB913 antibody. **(C)** Custom-made C-terminal rabbit polyclonal antibody PB939 against a peptide corresponding to aa 726–749 of mouse TPRN also recognizes TPRN (green) at the base of stereocilia like the antibodies described above. Stereocilia bundles were visualized by counterstaining using rhodamine-phalloidin (shown in magenta). **(D)** Western blot from mouse brain and cochlear tissues using custom TPRN RabmAb against a C-terminal peptide identical to the antigen for PB939 antibody shows a specific band in *Tprn*^+/+ tissues, but not in *Tprn*^−/− tissues, indicating that the antibody specifically recognizes TPRN. **(E–G)** COS-7 cells transfected with mCherry-MYO10-TPRN construct are stained with three antibodies against TPRN, PB913 (E), anti-C9ORF75 (F), and PB939 (G). TPRN fused to mCherry-MYO10-HMM (magenta) was transported to the filopodia tips at the cell periphery and recognized there by anti-TPRN antibodies (green) used to immunostaining transfected cells. Co-localization of mCherry-MYO10-TPRN (magenta) with TPRN antibody immunoreactivity (green) results in white color, confirming that these three anti-TPRN antibodies can specifically recognize TPRN in different cellular compartments, including filopodia. Scale bar in A–C is 5 µm, in E–G is 20 µm. **(H)** Silver-stained acrylamide gel of purified TRX-TPRN shows a major band at about 100 kDa. **(I)** Coomassie blue–stained gel showing TRX-TPRN treated with thrombin protease to remove the TRX-tag. Lane 1—untreated purified TRX-TPRN, lane 2—TRX-TPRN treated with thrombin, and lane 3—thrombin alone. Note, there is no band of TPRN without TRX (∼80 kDa) present in the second lane, but both thrombin-specific band and the TRX band are present, indicating that the removal of the TRX tag resulted in an unstable TPRN protein. **(J–L)** Additional examples of Myc-TPRN and HA–β-actin Co-IP shown in Fig. 4 C. Note the variable amount of HA–β-actin in the Co-IP row for each experiment. Source data are available for this figure: SourceData FS1.

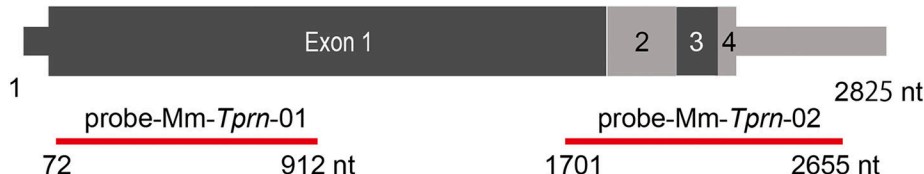

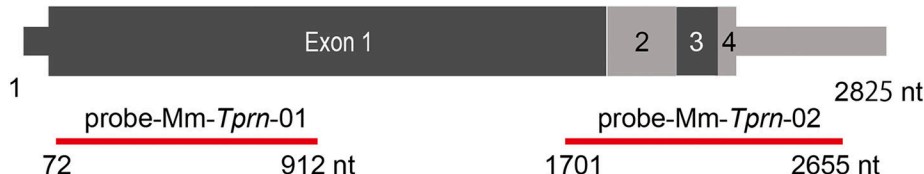

Figure S2. **Tprn mRNA expression in WT and Tprn⁻ᐟ⁻ mouse organ of Corti (OC) and stria vascularis (SV) using RNAscope probes. (A)** Regions of *Tprn* cDNA sequence of two RNAscope probes used to detect mouse *Tprn* mRNA. Probe-Mm-Tprn-01 targets the sequence toward the N-terminus of *Tprn* mRNA. Probe-Mm-Tprn-02 targets sequence for the C-terminus of *Tprn* mRNA. **(B)** Expression of *Tprn* mRNA in hair cells of the OC and SV at P3 *Tprn* mutant mice (*Tprn⁻ᐟ⁻*) and P3 WT littermates (*Tprn⁺ᐟ⁺*) using Probe-Mm-Tprn-01 (red) and Probe-Mm-Tprn-02 (red). Hair cells were highlighted by *Myo7a* (Probe-Mm-Myo7a-C2, magenta). Three OHCs and one IHC are visible in sections and highlighted by *Myo7a* signal (magenta). *Tprn* mRNA signal (red) overlaps with *Myo7a* signal (magenta) in *Tprn⁺ᐟ⁺* but absent in *Tprn⁻ᐟ⁻* OC. SV was highlighted by *Cldn11* mRNA encoding tight junction claudin 11 (Probe-Mm-Cldn11-C3, green), a marker of SV basal cells. *Tprn* mRNA signal (red) overlaps with *Cldn11* mRNA signal (green) and is also present in other SV cells in *Tprn⁺ᐟ⁺* but not in *Tprn⁻ᐟ⁻* SV. Scale bars are 50 μm.

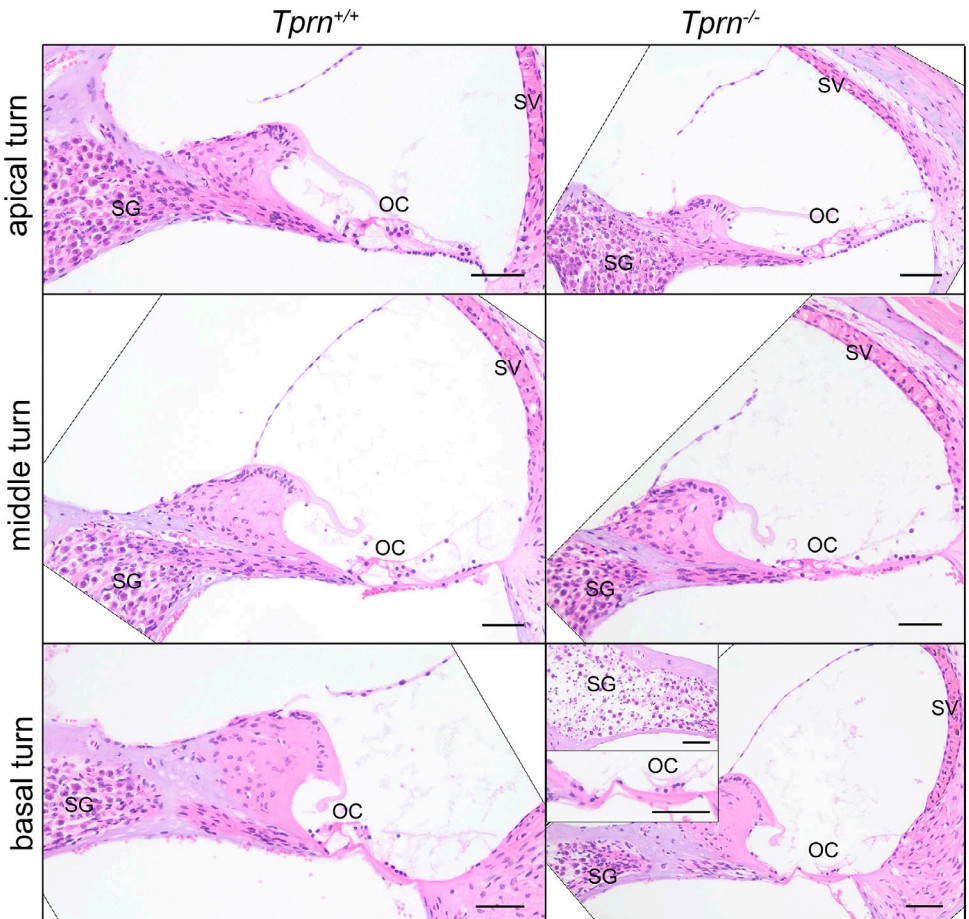

Figure S3. **Histological evaluation of the *Tprn⁻/⁻* mouse organ of Corti (OC) shows progressive degeneration.** Cross section of apical, middle, and basal turns of the cochlea of WT normal hearing *Tprn⁺/⁺* mouse (left panels) and deaf *Tprn⁻/⁻* littermate (right panels) at P60. OC degeneration is evident at basal and middle turns of *Tprn⁻/⁻* cochlea. In the panel of *Tprn⁻/⁻* basal turn the inserts show enlarged views of the degenerated OC and spiral ganglion neurons (SG) with substantial loss of neuronal cell bodies, which is not yet observed in the middle and apical turns of the *Tprn⁻/⁻* cochlea. In the middle turn of *Tprn⁻/⁻* cochlea, the OC shows loss of all OHCs and some supporting cells. In some TPRN-deficient mice, there is a mild increase in melanin pigment in the stria vascularis (SV). All turns of *Tprn⁺/⁺* cochlea show normal OC structure and normal density of spiral ganglion neurons. All scale bars are 50 µm.

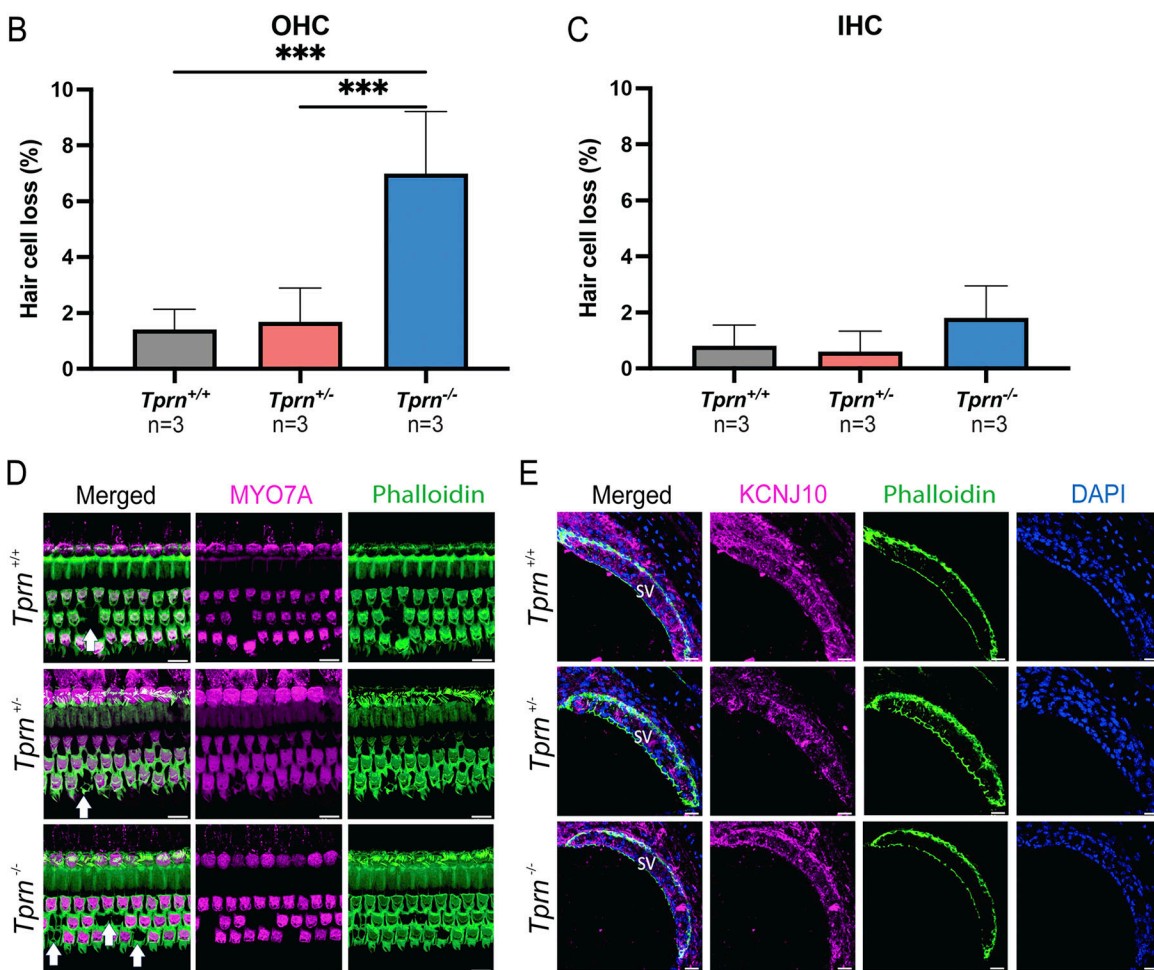

| Tukey's multiple comparisons test | Mean Diff. | 95.00% CI of diff. | Adjusted P Value |
|---|---|---|---|
| *Tprn*+/+ vs. *Tprn*+/- | -5.947 | -30.10 to 18.20 | 0.7879 |
| *Tprn*+/+ vs. *Tprn*-/- | 7.733 | -15.65 to 31.12 | 0.6555 |
| *Tprn*+/- vs. *Tprn*-/- | 13.68 | -6.345 to 33.70 | 0.2006 |

Figure S4. **SV morphology and EP are normal in *Tprn*−/− mouse. (A)** EP of P60 WT (*Tprn*+/+, two males, one female), heterozygous (*Tprn*+/−, two males, three females), and homozygous (*Tprn*−/−, three males, three females) mutant mice was within the WT range and indistinguishable between all three genotypes. Only left ears were tested in all three genotypes. **(B and C)** The inner ears of EP tested mice were immunostained, and the percentage of OHC and IHC loss was quantified and calculated as a fraction of number of missing hair cells out of the total number of this cell type in all images of middle and apical turn of the same ear. Error bars represent SD. One ear from each mouse was imaged and analyzed. Total of three animals were used for each genotype group. **(B)** *Tprn*−/− mice had higher percentage of OHC loss compared with the *Tprn*+/− and *Tprn*+/+. **(C)** No statically significant difference was observed between the percentage of IHC loss of the *Tprn*−/− mice with respect to the *Tprn*+/− and *Tprn*+/+. P values <0.001 were denoted with ***; unpaired two-sided *t* test. **(D)** Representative images of OC of EP tested mice of each genotype showing three rows of OHCs and one row of IHCs stained with antibody to MYO7A as a hair cell marker. Phalloidin is used to visualize the F-actin. Locations of missing OHCs are indicated by white arrows. Scale bars, 10 μm. **(E)** Cross-sections of the SV from the contralateral ear of EP tested animals. Cryosections stained for KCNJ10 (magenta, an intermediate cell marker), DAPI (nuclear marker), and phalloidin (green, F-actin) did not show any differences in SV staining and morphology of the *Tprn*−/− mice compared with the *Tprn*+/− and *Tprn*+/+ littermates. Scale bars 20 μm.

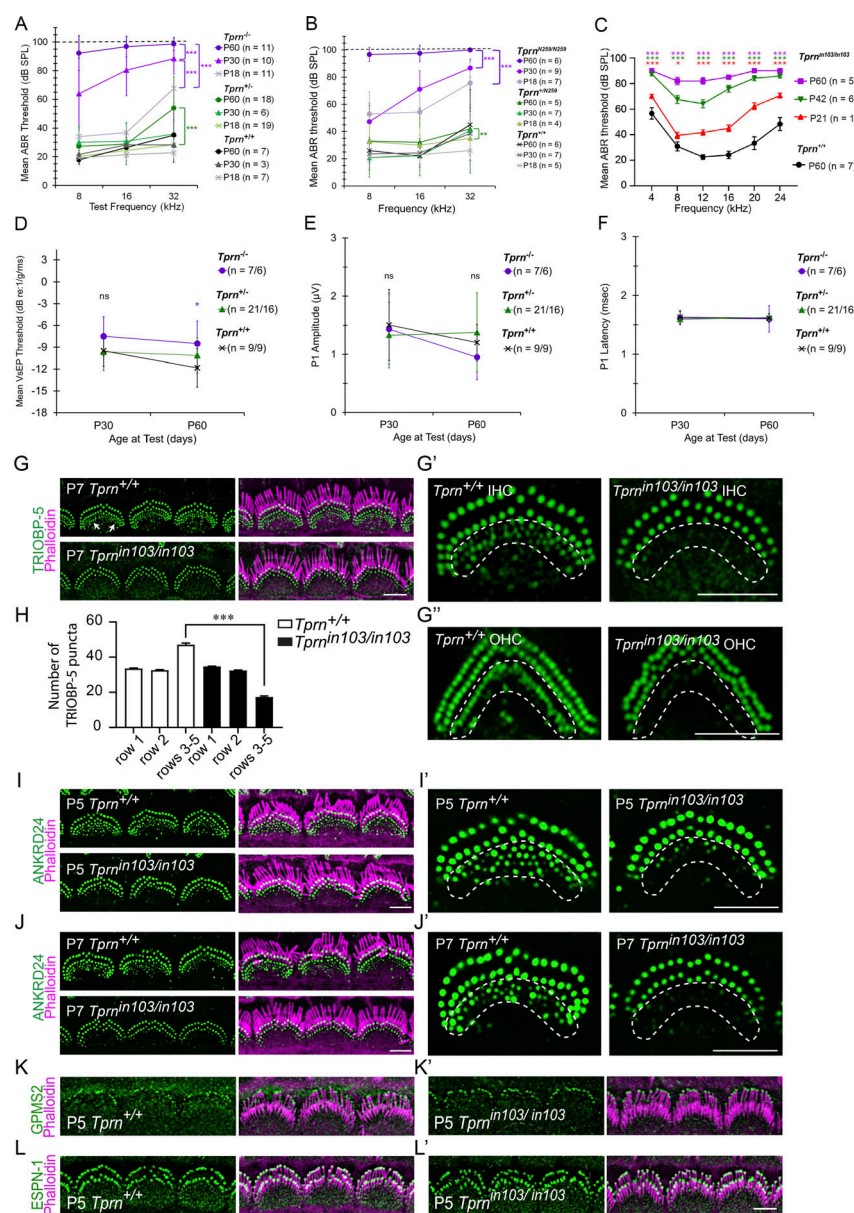

**Figure S5.** **ABR thresholds of *Tprn*[−/−], *Tprn*[N259/N259], and *Tprn*[in103/in103] mice and VsEP measurements of *Tprn*[−/−] mice. ANKRD24, TRIOBP-5, GPSM2, and ESPN-1 localization in WT and *Tprn*[in103/in103] postnatal hair cell stereocilia and hair bundle morphology of TPRN-deficient mice. (A)** Mean ABR thresholds at P18, P30, and P60 of *Tprn*[+/+], *Tprn*[+/−], and *Tprn*[−/−] littermates at 8, 16, and 32 kHz. Using linear mixed effects regression, we found that *Tprn*[−/−] mice had significantly worse hearing overall (Table S2) and exhibited progressive deafness (Table S3) with greater early hearing loss at high frequencies (Tables S4 and S5). **(B)** Mean ABR thresholds of *Tprn*[N259/N259], *Tprn*[+/N259], and *Tprn*[+/+] littermate controls at P18, P30, and P60, *n* = number of animals tested. Similar to *Tprn*[−/−] mice, *Tprn*[N259/N259] mice exhibit worse hearing overall and a significant worsening of hearing over time (Tables S6 and S7). **(C)** Mean ABR thresholds of *Tprn*[in103/in103] and WT control mice (*Tprn*[+/+]) at P21, P42, and 2 mo of age, *n* = number of animals tested. At all ages and frequencies, *Tprn*[in103/in103] mice exhibited worse hearing than P60 *Tprn*[+/+] (Tables S8 and S9). **(D–F)** Mean VsEP threshold (D), P1 amplitude (E), and P1 latency (F) for *Tprn*[+/+], *Tprn*[+/−], and *Tprn*[−/−] littermates at P30 and P60 time points. *Tprn*[−/−] mice did not exhibit significant differences in VsEP thresholds, P1 amplitudes, or P1 latencies except for a significant difference in VsEP threshold at P60 compared with *Tprn*[+/+], P = 0.04 (Tables S10 and S11), which is likely attributable to expected test-retest variation between time points. N indicates the number of animals tested at P30/P60, correspondingly. Graphs A–F display mean ± SD. *, **, ***—significant difference compared with *Tprn*[+/+] animals at P < 0.05, P < 0.01, and P < 0.001, respectively. Color of asterisks indicates the group exhibiting the difference. Brackets indicate differences within a genotype between time points. **(G–G″)** Whole mounts of cochlea from P7 WT and *Tprn*[in103/in103] mice were stained with antibodies against TRIOBP-5 (G–G″) and phalloidin (magenta) to reveal stereocilia. **(G′ and G″)** Enlarged images of a representative IHC (G′) and OHC (G″) at stereocilia level. **(H)** Quantification of TRIOBP-5 puncta at the bases of different rows of OHC stereocilia of P7 *Tprn*[+/+] and *Tprn*[in103/in103] mice. Thirty-four WT and forty-two *Tprn*-deficient OHCs from at least three mice were analyzed per group. Data are means ± SEM. ***P < 0.001 by unpaired two-sided *t* test. **(I–J′)** Whole mounts of cochlea from P5 (I and I′) and P7 (J and J′) WT and *Tprn*[in103/in103] mice were stained with antibodies against ANKRD24, and counterstained by phalloidin (magenta) to show F-actin of stereocilia. **(I′ and J′)** Enlarged images of a representative IHC at stereocilia level. Number of TRIOBP-5 and ANKRD24 puncta at bases of immature short-row stereocilia were significantly reduced in P5 and P7 *Tprn*-deficient hair cells. **(K–L′)** Cochlear whole mounts from P5 WT (K and L) and *Tprn*[in103/in103] mice (K′ and L′) were stained for GPSM2 (K and K′) or ESPN-1 (L and L′), which is localized at tips of stereocilia. No significant change of GPSM2 or ESPN-1 localization was observed in *Tprn*-deficient mice (K′ and L′). Scale bars are 5 μm.

Video 1.   **Overexpression of EGFP-TPRN in a COS-7 cell induces F-actin bundle formation in the nucleus.** Time-lapse recording of a COS-7 cell nucleus expressing mouse full-length EGFP-TPRN, using a PerkinElmer UltraView VoX spinning disk confocal microscope equipped with SCU-X1 spinning disk scan head (Yokogawa) and EMCCD camera (Hamamatsu). COS-7 cells were transfected with EGFP-TPRN (green) using Lipofectamine-2000 (Thermo Fisher Scientific) and incubated for 24 h at 37°C and 5% $CO_2$ before recording. The movie captured the stage of a whorl-like bundle forming when EGFP-TPRN (green) cross-linked F-actin bundles continue to elongate, converge with shorter ones, and fuse with each other. Note that at the end of the movie no short EGFP-TPRN–highlighted filamentous fragments remain. Maximum projections of volume scans acquired every 15 s are shown. Scale bar, 5 µm.

**Provided online are Table S1, Table S2, Table S3, Table S4, Table S5, Table S6, Table S7, Table S8, Table S9, Table S10, and Table S11. Table S1 shows the reported pathogenic or likely pathogenic variants of human *TPRN* and associated phenotypes. Table S2 shows the statistical analysis of ABR data for *Tprn*[−/−], *Tprn*[+/−], and *Tprn*[+/+] mice. Table S3 shows the comparisons between time points of *Tprn*[−/−], *Tprn*[+/−], and *Tprn*[+/+] ABR data within genotypes. Table S4 shows the statistical analysis of *Tprn*[−/−], *Tprn*[+/−], and *Tprn*[+/+] ABR data at each frequency. Table S5 shows the comparisons of *Tprn*[−/−], *Tprn*[+/−], and *Tprn*[+/+] ABR data between genotypes at each frequency and time point. Table S6 shows the statistical analysis of ABR data from *Tprn*[N259/N259], *Tprn*[+/N259], and *Tprn*[+/+] mice. Table S7 shows the comparisons between time points of *Tprn*[N259/N259], *Tprn*[+/N259], and *Tprn*[+/+] ABR data within genotypes. Table S8 shows the statistical analysis of ABR data from *Tprn*[in103/in103] and *Tprn*[+/+] mice. Table S9 shows the comparisons of ABR data from *Tprn*[in103/in103] and *Tprn*[+/+] mice at each frequency between genotype and age groups. Table S10 shows the statistical analyses of *Tprn*[−/−], *Tprn*[+/−], and *Tprn*[+/+] VsEP threshold, P1 amplitude, and P1 latency measures for each genotype and age. Table S11 shows the comparisons of *Tprn*[−/−], *Tprn*[+/−], and *Tprn*[+/+] VsEP thresholds, P1 amplitude, and P1 latency between genotypes at specific ages.**

