## [Peer Review File · The Journal of Cell Biology]

Taperin bundles F-actin at stereocilia pivot-points enabling optimal lifelong mechanosensitivity

Inna Belyantseva, Chang Liu, Abigail Dragich, Takushi Miyoshi, Sayaka Inagaki, Ayesha Imtiaz, Risa Tona, Karen Zuluaga-Osorio, Shadan Hadi, Elizabeth Wilson, Eva Morozko, Rafal Olszewski, Rizwan Yousaf, Yuliya Sokolova, Gavin Riordan, Sean Aston, Atteeq Rehman, Cristina Fenollar Ferrer, Jan Wisniewski, Shoujun Gu, Gowri Nayak, Richard Goodyear, Jinan Li, Jocelyn Krey, Talah Wafa, Rabia Faridi, Samuel Adadey, Meghan Drummond, Benjamin Perrin, Dennis Winkler, Matthew Starost, Hui Cheng, Tracy Fitzgerald, Guy Richardson, Lijin Dong, Peter Barr-Gillespie, Michael Hoa, Gregory Frolenkov, Thomas Friedman, and Bo Zhao

Corresponding Author(s): Bo Zhao, Indiana University School of Medicine and Bo Zhao, Indiana University School of Medicine

Review Timeline:

Submission Date:	2024-08-05
Editorial Decision:	2024-09-24
Revision Received:	2025-02-24
Editorial Decision:	2025-03-24
Revision Received:	2025-05-03

Monitoring Editor: Pekka Lappalainen

Scientific Editor: Dan Simon

Transaction Report:

DOI: <https://doi.org/10.1083/jcb.202408026>

September 24, 2024

Re: JCB manuscript #202408026

Dr. Bo Zhao
Indiana University School of Medicine

Dear Dr. Zhao,

Thank you for submitting your manuscript entitled "Taperin bundles F-actin at stereocilia pivot-points enabling optimal lifelong mechanosensitivity." The manuscript was assessed by expert reviewers, whose comments are appended to this letter. We invite you to submit a revision if you can address the reviewers' key concerns, as outlined here.

All three reviewers found this study very interesting, and appreciated the comprehensive approach to elucidate the function of taperin. While generally very positive about the study, they, however, raised a few important points that should be addressed to further strengthen this manuscript. As you will see from their comments, the reviewers stated that more thorough quantification of the data are needed in several instances, and in some cases better quality images should be provided. Certain figures and figure legends should be also edited to improve the clarity, and the missing controls included when appropriate (e.g. Fig. 4).

If technically possible, some biochemical experiments could be also repeated with untagged proteins as suggested by Reviewer #3. Moreover, as pointed out by Reviewer #2, the MET current data in Fig. 9 should be strengthened. Alternatively, you may consider removing these data from the manuscript. Finally, we ask you to address all minor points by the reviewers to increase the clarity of the manuscript, as well as to better acknowledge earlier papers throughout the manuscript text.

GENERAL GUIDELINES:

Text limits: Character count for an Article is < 40,000, not including spaces. Count includes title page, abstract, introduction, results, discussion, and acknowledgments. Count does not include materials and methods, figure legends, references, tables, or supplemental legends.

Figures: Articles may have up to 10 main text figures. Figures must be prepared according to the policies outlined in our Instructions to Authors, under Data Presentation, <https://jcb.rupress.org/site/misc/ifora.xhtml>. All figures in accepted manuscripts will be screened prior to publication.

*****IMPORTANT:** It is JCB policy that if requested, original data images must be made available. Failure to provide original images upon request will result in unavoidable delays in publication. Please ensure that you have access to all original microscopy and blot data images before submitting your revision. ***

Supplemental information: There are strict limits on the allowable amount of supplemental data. Articles may have up to 5 supplemental figures. Up to 10 supplemental videos or flash animations are allowed. A summary of all supplemental material should appear at the end of the Materials and methods section.

Please note that JCB now requires authors to submit Source Data used to generate figures containing gels and Western blots with all revised manuscripts. This Source Data consists of fully uncropped and unprocessed images for each gel/blot displayed in the main and supplemental figures. Since your paper includes cropped gel and/or blot images, please be sure to provide one Source Data file for each figure that contains gels and/or blots along with your revised manuscript files. File names for Source Data figures should be alphanumeric without any spaces or special characters (i.e., SourceDataF#, where F# refers to the associated main figure number or SourceDataFS# for those associated with Supplementary figures). The lanes of the gels/blots should be labeled as they are in the associated figure, the place where cropping was applied should be marked (with a box), and molecular weight/size standards should be labeled wherever possible. Source Data files will be made available to reviewers during evaluation of revised manuscripts and, if your paper is eventually published in JCB, the files will be directly linked to specific figures in the published article.

The typical timeframe for revisions is three to four months. If you anticipate any difficulties in meeting this aforementioned

revision time limit, please contact us and we can work with you to find an appropriate time frame for resubmission. Please note that papers are generally considered through only one revision cycle, so any revised manuscript will likely be either accepted or rejected.

Thank you for this interesting contribution to Journal of Cell Biology. You can contact us at the journal office with any questions at cellbio@rockefeller.edu.

Sincerely,

Pekka Lappalainen, PhD
Monitoring Editor
Journal of Cell Biology

Dan Simon, PhD
Scientific Editor
Journal of Cell Biology

Reviewer #1 (Comments to the Authors (Required)):

In this interesting paper, the authors report on the function of taperin (TPRN) in the context of mechanosensory hair cells. Mutations in TPRN lead to non-syndromic hearing loss and previous work showed that deficiency of TPRN in mice leads to stereocilia degeneration. Previous studies from the authors and others also localized this factor to the base of stereocilia, where the stereocilia core actin bundle tapers as it enters the apical surface of the hair cell. Despite prior efforts, the molecular function of TPRN and the mechanisms that lead to failure of stereocilia structure with loss of TPRN remain unclear, and these are some of the open questions that the authors sought to address with this work. Using approaches that extend from *in vitro* studies with purified proteins to characterization of genetically modified mice, the authors examine the details of TPRN localization, the actin binding and bundling activities of TPRN, TPRN self-association, and the impact of TPRN KO and overexpression on hair cell morphology and electrophysiological activity. Collectively, the findings presented here indicate that TPRN is an actin bundling protein that localizes to the outside of the tapering stereocilia core bundle, where it also participates in linking the bundle to the overlying plasma membrane. These activities are critical for building and maintaining normal stereocilia bundles and rootlets and preventing stereocilia fusion. I enjoyed reading this strong and comprehensive paper, but there are some points that I hope the authors will consider address toward improving this work before publication.

Major points

(1) Line 164 - "The TPRN staining extends through the entire stereocilia taper region forming a funnel-shaped structure (Fig. 1, L, arrows) enveloping the F-actin core (Fig. 1, L, N)." The funnel-shaped localization of TPRN is not clear from the images provided in this figure. One can almost see what the authors are referring to here, but only one arrow in the L panel shows what looks like a funnel shaped signal. Better examples and some indication of how the optical section is oriented relative to the volume of the cell might help. Also, the authors should reference the STED data in the Qi et al. 2024 paper, which showed clear funnel-shaped TPRN localization in the taper region.

(2) Line 204 - "To investigate if TPRN alone can bundle F-actin, we fused purified full-length mouse TPRN with a thioredoxin (TRX) tag, purified TRX-TPRN from *E. coli* (Fig. 3, A), and used it in actin polymerization assays *in vitro*." Based on the protocol, it is my understanding that this assay is testing how TPRN impacts actin filament bundling rather than polymerization.

(3) Line 208 - "TRX-TPRN was localized along the length of actin bundles or single actin filaments *in vitro*...". The distinction between filament bundles and single filaments is not clear in these images. Does purified TPRN bind to single actin filaments?

(4) Figure 4 - There is a lot of information here on how TPRN binds to actin and to itself, but the narrative glosses over the details. Moreover, it was difficult to interpret the data in this figure, in part because the individual panels are not well described in the legend, but also because not all the blots needed to interpret the various pulldowns are shown. If we start with Fig. 4B, I believe what the authors are trying to communicate here is that they can pulldown HA-beta-actin with an anti-Myc antibody, but only when Myc-TPRN is in the reaction. Here they include both input blots (one for HA and one for Myc) showing the expected bands. However, only one pulldown blot is shown, for Myc-TPRN (which runs right below ~100 kDa); there is no pulldown blot for HA so it is impossible to know if HA-beta-actin bound to Myc-TPRN in these assays. I would encourage the authors to include a different set of blots for each pulldown, top to bottom:

- 1) Pulldown fractions blotted with anti-HA
- 2) Pulldown fractions blotted with anti-Myc
- 3) Input fraction blotted with anti-HA
- 4) Optional: Input fraction blotted with anti-Myc, but this becomes redundant if you are doing anti-Myc pulldowns and show pulldown fractions blotted for anti-Myc.

(5) Related to interpretation of the data in Figure 4, is HA-beta-actin polymerizing in these pulldown assays? In Figure 4C, every fragment tested by the authors appears to bind to actin. Moreover, it looks like the amount of actin coming down in each case is similar, which this reader would not expect given differences in the expression levels of Myc-TPRN constructs. Could the authors be pelleting polymerized HA-beta-actin in these experiments?

(6) Figure 10 - Did the TPRN knockout stereocilia exhibit ultrastructural defects in their tapers and rootlets when imaged with FIB-SIM?

(7) Figure 10D - The authors state that TPRN ... "brings actin filaments of the stereocilia core closer together to traverse the pivot point (Fig. 10, D)." If this statement is based on a specific piece of data in this paper, the authors should point to it here. If not, this sentence should be toned down or removed.

(8) Line 534 - The authors state "...we refined our originally reported localization of TPRN (Rehman et al., 2010) and showed that TPRN is not just a ring around actin filaments as recently reported by others (Qi et al., 2024) Rather, TPRN localizes along the taper region of stereocilia forming a funnel-like shape enveloping minus-ends of peripheral F-actin and bringing more centrally positioned actin filaments closer together to incorporate them into the lower half of rootlets." Qi et al. (2024) used STED to show funnel-like localization for TPRN in the taper region in Figure 1F of their paper. The authors should edit this section to acknowledge that result. Moreover, the statement on TPRN "bringing... actin filaments closer together" is speculation and should be written as such.

(9) Line 596 - The authors state "In summary, our data show that in vitro, purified TPRN alone is able to bundle F-actin into relatively loose and curved bundles." Given that there are no filament spacing measurements in this paper, and no other actin bundlers (e.g. fascin or espin) were examined as points of comparison, the "loose" and "curved" descriptors do not seem justified.

Minor edits

(1) Figure 1 - The cartoon in panel B should have amino acid numbers marked along the length, at the margins of the different motifs.

(2) Figures 4 and 5 - The authors should include a summary cartoon, perhaps as a panel at the end of Figure 5, highlighting the regions that are involved in actin binding and self-association.

(3) The dataset here is expansive and could stand to be consolidated in a few different places to increase paper readability. For example, Figure 2 and Figure 8 both show overexpression of TPRN phenotypes; this could be one figure.

(4) Line 426 - Replace "didn't" with "did not" (contractions might also be used elsewhere in the paper, please check).

Reviewer #2 (Comments to the Authors (Required)):

This study by Inna A. Belyantseva and colleagues elucidates the essential role of Taperin (TPRN) in actin bundling, highlighting its critical importance in maintaining the structural stability and functionality of stereocilia. The study emphasizes that defects in TPRN can lead to hearing loss. They found TPRN to localize to the ankle region of the stereocilia and modify actin filament structures likely through bundling using a series of in vitro and in vivo experiments. Additionally, they mapped the functional domains of the protein that are important for actin binding and actin filament bundling. Functional analysis further revealed the connection between TPRN and actin bundling, utilizing mouse models TPRN^{-/-}, TPRN^{in103/in103}, TPRN^{N259/N259}, and TPRN^{Q259}. This comprehensive investigation examines the functional deficits caused by TPRN defects using hair cell electrophysiology and analyzes changes in stereocilia rootlet ultrastructure. Overall, this is an impressive amount of work using multiple different techniques and approaches to support their hypothesis; however, the experiments and data are not often quantified and the narrative is at times disjointed without a clear line of rationale. To strengthen and complement these findings, the following comments are suggested:

-Major comments

1. A major concern in this paper is the lack of sufficient quantification. In figure 1O, statistical analysis was performed on the fluorescence intensity of ANKRD24, TRIOBP5, and TRIOBP4, but there is no mention of the number of stereocilia, cells, and animals from which the data was collected. Figure 2 presents the results of TPRN transfection using a gene gun, where the images suggest low transfection efficiency, but the frequency of each phenotype upon overexpression in each tissue is not clearly provided (*line 495 - line 497). Additionally, figures 3D and 5A illustrate the effects of co-localization of actin and TPRN

on thickness and bundling, however quantitative analyses were not performed to prove the pattern changes across experiments (*line 565-line 569).

2. In figure 3A, the purity of TRX-TPRN is confirmed using Coomassie blue staining. However, silver staining could potentially reveal higher impurity levels, and these impurities could be affecting the experimental outcome. A minimum of silver staining of the purified protein is needed if multiple purifications steps are not performed.
3. There is no statistical analysis in figure 6B, supplementary figure 5A and 5B.
4. In figures 7H-I, the number of 3rd-5th rows is compared for IHCs between TPRN^{+/+} and TPRN^{in103/in103}. The 3rd row is more clearly observed in OHCs than in IHCs, so a comparative analysis of the changes in the 3rd row observed in OHCs would provide a clearer answer. If this phenotype is specific for IHCs, that would also be useful knowledge. By specifically analyzing how these changes differ between OHCs and IHCs and how they relate to TPRN deficiency, a deeper understanding of TPRN's functional role can be obtained. And in figures 7M-P, TPRN expression in OHCs and IHCs is compared in PTPRQ KO mice. The results show a tendency for reduced localization of TPRN in PTPRQ^{-/-}, but quantitative analysis of the number of puncta or fluorescence intensity across different cells is needed, since it is unclear which is the phenotype the authors are indicating is the cause of the decreased intensity.
5. Additionally, figure 8A shows the results of AAV injection. While this method demonstrates higher efficiency than the gene gun used in figure 2, it would be preferable to present the changes observed in multiple cells rather than just one. Furthermore, quantitative analysis of the frequency of degeneration or over-elongated bundles is necessary to provide the reader with an indication as to the prevalence of these phenotypes. For the changes shown in figures 8H-I, it is also important to quantitatively analyze how many cells, across how many animals, and how frequently overgrown or extra short-thick stereocilia are observed.
6. In Figure 9E - 9F, the peak current size of the outer hair cells is smaller than typically expected and shows a large variance (ranging from at least 0.3 nA to 0.8 nA). This variance is not indicative of the biological factors and likely due to experimenter variability. The variability will make it difficult to assess open probability as well since lower current sizes may bias open probability measurements. This makes it difficult to assess any changes that may be occurring in the mutant lines if there is so much uncontrolled variability in the WT data. Peak current data for TPRN^{N259/N259} and TPRN^{-/-} are different even though both are thought to be knockouts, warranting some explanation. Additionally, the stiffness shown in Figure 9G focuses only on the fluid jet area. One of the factors that significantly influence force on the hair bundle is the Z-axis position of the fluid jet, but these considerations are not mentioned to be controlled. Also, as mentioned in the methods, the distance between the fluid jet and the hair bundle is 6 - 10 μ m. This distance can cause ~ 60% reduction in force (Tobin et al., 2019), further indicating that these experiments are poorly controlled. Considering these factors, additional experiments are required for accurate assessment of the current size and stiffness.
7. In figure 10C, the observed F-actin disruption and kinked rootlet quantification is done only in 2 animals for each genotype and these experiments should be performed on a minimum of 3 animals (biological replicates).
8. In the discussion, there are statements as if the data was shown related to TPRN function, however, many times the data in current form do not prove function. Especially, there are over-interpreted sentences including line 468 - 471, 495 - 497, 536 - 539, 548 - 549, 573 - 575, 598 - 599. Please modify the language to indicate these are hypotheses based on actual data shown in the manuscript. Also, in line 472 - 473, the statement "TPRN deficient rootlets are thinner" lacks quantitative evidence of the rootlet thickness.

Minor comments

1. Line 290-291 should be more precise to avoid confusion. The original sentence could imply both a lack of TPRN and a lack of antibody specificity. Therefore, it should be rewritten to remove any confusion.
2. In figure 10D, tip link breakage is observed in Taperin-deficient cells. Additional information/quantification is needed to determine whether these findings align with results from tip links observed in different biological replicates across various mice. The authors have argued that peak current is not altered in the TPRN^{-/-} (figure 9E), therefore the loss of tip links in the model is surprising.
3. In the description of supplementary figure 1 (line 80), the antibody PB913 was mistakenly labeled as BP913. Also, in line 472, there is a spelling error; "TROBP" should be corrected to "TRIOBP". In line 531, 585, there is a spelling error; "TRPN" should be corrected to "TPRN".
4. In line 422 - 424 (Fig. 9A), the sentence "We observed similar MET responses to step deflections of stereocilia" does not make sense. Fluid jet does not give 'step-deflection'.
5. In line 475 - 477, figure 7 does not mention the gradual disappearance of ANKRD24. And in line 477 - 479, there is a significant loss of MET current in TPRN^{N259/N259} hair cells.

Reviewer #3 (Comments to the Authors (Required)):

The study "Taperin bundles F-actin at stereocilia pivot-points enabling optimal lifelong mechanosensitivity" by Belyantseva* and Liu* et al 2024 investigates the role of taperin (TPRN) in stereocilia structure and its contribution to hearing in mice. Taperin (TPRN) is found at the base of stereocilia and helps to stabilize these structures at pivot points. Lack of TPRN leads to structural degeneration, with warped rootlets and a gradual loss/localization of additional (previously characterized) proteins (PTPRQ, TRIOBPs, and ANKRD24). In contrast, TPRN overexpression causes excessive F-actin bundling and abnormal stereocilia growth. Purified (but still tagged) TPRN protein seems to directly bundle actin filaments in pelleting assays and fluorescence-based assays, through residues present in the first 1-170 amino acids. The authors have done a substantial amount of work

successfully combining assays with purified proteins, cells, and animal models, to provide a very complete picture of TPRN's role in stereocilia structure and function. However, the work is missing several controls as presented and I have a few minor comments as detailed below:

Major comments:

The authors provide a GFP- alone control but unless I missed it, I did not explicitly see a control indicating the functionality of tagged TPRN (GFP- or TRX-tags). The authors provided a GFP- alone control but not the necessary version lacking the tag. It seems very addressable due the prominent bundling activity seen in cells and in vitro - where it may be likely that elevated bundling activity can be seen by actin stain alone. While the authors mention purification issues in the methods, an untagged version of the protein is also important for the interpretation of the results. Can the solubility increasing tag be cleaved at a later purification step to make an untagged version of the protein?

The demonstration that TPRN is overexpressed should be further clarified or validated with a comparison to endogenous levels.

The authors are using this image in Figure 3C to claim the localization of TPRN occurs along filaments rather than at ends, however it is difficult to know for sure since it does not appear to be a single filament. Further, the authors do not specify why they might expect the localization to be on an actin filament end. This image should be replaced with something more representative.

The schematic noting the TPRN constructs capable of binding and bundling actin filaments can be improved by adding + and - notations summarizing the activities of each truncated protein.

The blots present in Figure 4 do not display molecular markings in accordance with JCB guidelines and will need to be re-cropped to have markers above and below the bands of interest to comply.

Minor comments:

Throughout the work the authors refer to "loose" TPRN bundles. While the intention is likely to make a comparison between TPRN and other well characterized bundling proteins like fascin and alpha actinin, this terminology somewhat confusing and should be further clarified.

The color choices and size of the panels in Figure 1, specifically Figure 1D, makes it difficult to discern how much overlap is present with TPRN and ANK. It would be helpful to potential readers to change the colors to better show this overlap and to add additional line scans, particularly for the comparison of signals in 1C-F).

Can the authors comment on whether the nuclear TPRN (particularly in the punctate state) in Figure 2G-K? If it is, perhaps additional biophysical properties are involved in mediating TPRN's bundling activities.

Do the authors know or were they able resolve how many actin filaments are in the TPRN bundles in Figure 3D? What about their orientations (i.e., does TPRN make parallel, anti-parallel, or mixed bundles)

The data present in Figure 9 is cool, but it does not need to be a full-page figure. Perhaps this information can be consolidated with other data to reduce the total number of figures according to the JCB limit/guidelines. In a similar vein, the localization studies with TRIO proteins and ANKRD24 contain stunning images but they do not contribute much to the story beyond putting it in the context of past work and the localization/connection with TPRN. These figures can likely be consolidated or moved into the supplement.

Feb 24, 2025

Pekka Lappalainen, PhD
Monitoring Editor
Journal of Cell Biology

Dan Simon, PhD
Scientific Editor
Journal of Cell Biology

Dear Dr. Lappalainen and Dr. Simon,

Thank you for conducting a thorough assessment of our manuscript entitled "Taperin bundles F-actin at stereocilia pivot-points enabling optimal lifelong mechanosensitivity" by three expert reviewers.

We revised the manuscript and provided all the additional experimental data that reviewers requested by including:

1. quantification of data in updated Figs 1, 3, 6B, 7 C-E and H-J and S5, A-F
2. a detailed method description of statistical analyses in the "Methods" section of the revised manuscript
3. improved image quality of the TPRN "funnel-like staining" in Fig. 1, L with added panel R for enhanced clarity
4. a graphic analysis of the fluorescence intensity along the stereocilia length, Fig. 1, panels E' and F'
5. improved and more detailed figures and figure legends (Figs 4, 7, 8, Fig. S1, S5)
6. an additional control using purified espin (ESPN) protein for experiments with negative stain electron microscopy of purified TPRN (see revised Fig. 3)
7. additional biochemical experiments as requested (Figs 4 and S1)
8. combined Fig. 8 and half of Fig. 7 to clarify data presentation and the take-home messages from these figures
9. new statistical analysis of TPRN puncta in *Ptprq* knockout mouse hair cells (Fig. 7)
10. clarification of fluid-jet positioning and estimates of the accuracy of this positioning in Fig. 9
11. additional new FIB-SEM imaging data incorporated into revised Fig. 10
12. ten Supplemental Tables with detailed statistical analyses of the ABR and VsEP measurements of three different TPRN-deficient mice

All additional data in the revised manuscript further confirm our results and strengthen conclusions in the manuscript. All edits in the revised manuscript are highlighted. All authors agreed to add three co-authors, two statisticians (Drs. Hui Cheng and Andrew Aston) and Dr. Jan Wisniewski who helped with quantitative analyses of the data.

Please find below our point-by-point responses to reviewers' comments.

Reviewer #1 (Comments to the Authors (Required)):

In this interesting paper, the authors report on the function of taperin (TPRN) in the context of mechanosensory hair cells. Mutations in TPRN lead to non-syndromic hearing loss and previous work showed that deficiency of TPRN in mice leads to stereocilia degeneration. Previous studies from the authors and others also localized this factor to the base of stereocilia, were the stereocilia core actin bundle tapers as it enters the apical surface of the hair cell. Despite prior efforts, the molecular function of TPRN and the mechanisms that lead to failure of stereocilia structure with loss of TPRN remain unclear, and these are some of the open questions that the authors sought to address with this work. Using

approaches that extend from in vitro studies with purified proteins to characterization of genetically modified mice, the authors examine the details of TPRN localization, the actin binding and bundling activities of TPRN, TPRN self-association, and the impact of TPRN KO and overexpression on hair cell morphology and electrophysiological activity. Collectively, the findings presented here indicate that TPRN is an actin bundling protein that localizes to the outside of the tapering stereocilia core bundle, where it also participates in linking the bundle to the overlying plasma membrane. These activities are critical for building and maintaining normal stereocilia bundles and rootlets and preventing stereocilia fusion. I enjoyed reading this strong and comprehensive paper, but there are some points that I hope the authors will consider address toward improving this work before publication.

Major points

(1) Line 164 - "The TPRN staining extends through the entire stereocilia taper region forming a funnel-shaped structure (Fig 1, L, arrows) enveloping the F-actin core (Fig 1, L, N)." The funnel-shaped localization of TPRN is not clear from the images provided in this figure. One can almost see what the authors are referring to here, but only one arrow in the L panel shows what looks like a funnel shaped signal. Better examples and some indication of how the optical section is oriented relative to the volume of the cell might help. Also, the authors should reference the STED data in the Qi et al. 2024 paper, which showed clear funnel-shaped TPRN localization in the taper region.

Response:

We agree. The revised manuscript includes an improved STED image as a replacement for original Fig. 1, panel L showing a funnel-like localization of endogenous TPRN in wild type IHC stereocilia. The image was taken along the longitudinal axis of IHC stereocilia using our custom rabbit monoclonal primary antibody to TPRN and an anti-rabbit STAR RED secondary nanobody (Abberior).

It is worth mentioning that we show super-resolution image of native endogenous TPRN. In contrast, Fig. 1 F in Qi et al, 2024 shows detection of an HA-tag (3 x HA) inserted at the C-terminus of TPRN that included the addition of a three-residue linker. The addition of 3 x HA-tag at the carboxy-terminus of TPRN could have altered TPRN localization and/or function because it buries a predicted PDZ ligand (PALYF, see Fig. 1, B) located at the C-terminal end of TPRN. Moreover, the incorporation of the HA-tag was not robust. Qi et al mentioned detecting TPRN only in a few hair cell bundles with various levels of HA-tag expression (see Qi et al, Fig. S1, E). In addition, Qi and coauthors sometimes found a mutation within the HA-tag (their Fig. S1, D). These factors may have altered localization and/or expression of HA-TPRN relatively to the endogenous TPRN. In the revised manuscript, we cited Qi et al, 2024 indicating the similar pattern between our native TPRN immunolocalization and their TPRN-HA. See legend to Fig. 1, panel L.

Our revised sentence on page 23 now reads:

"This pattern is similar to TPRN-HA visualized with an anti-HA-antibody (Qi et al, 2024)."

(2) Line 204 - "To investigate if TPRN alone can bundle F-actin, we fused purified full-length mouse TPRN with a thioredoxin (TRX) tag, purified TRX-TPRN from *E. coli* (Fig 3, A), and used it in actin polymerization assays in vitro." Based on the protocol, it is my understanding that this assay is testing how TPRN impacts actin filament bundling rather than polymerization.

Response:

The reviewer is correct. The revised sentence now reads: "To investigate if TPRN alone can bundle F-actin, we fused FL mouse TPRN with a thioredoxin (TRX) tag, purified TRX-TPRN from *E. coli* (Fig. 3, A and Fig. S1, H, I), and used it in actin-bundling assays *in vitro*."

(3) Line 208 - "TRX-TPRN was localized along the length of actin bundles or single actin filaments in vitro...". The distinction between filament bundles and single filaments is not clear in these images. Does purified TPRN bind to single actin filaments?

Response:

The reviewer raised an excellent question. In the revised manuscript, we included an improved image to replace the original Fig. 3, C. This new image (Fig. 3, D) includes multiple short filaments showing TPRN along their lengths. Additionally, we measured the lengths of the filaments in Fig 3, D and compared them with the length of single actin filaments shown in the right panel of Fig. 3, B. Furthermore, we have revised our statement to make it more cautious, and it now reads: "FL TRX-TPRN was localized along the length of actin bundles (Fig. 3, C, right) or on likely single filaments with the lengths of $10.27 \pm 0.42 \mu\text{m}$ ($n=55$), which are comparable to the lengths of single actin filaments shown in the right panel of Fig. 3, B ($10.34 \pm 0.36 \mu\text{m}$, $n=96$)."

(4) Figure 4 - There is a lot of information here on how TPRN binds to actin and to itself, but the narrative glosses over the details. Moreover, it was difficult to interpret the data in this figure, in part because the individual panels are not well described in the legend, but also because not all the blots needed to interpret the various pulldowns are shown. If we start with Fig. 4B, I believe what the authors are trying to communicate here is that they can pulldown HA-beta-actin with an anti-Myc antibody, but only when Myc-TPRN is in the reaction. Here they include both input blots (one for HA and one for Myc) showing the expected bands. However, only one pulldown blot is shown, for Myc-TPRN (which runs right below ~ 100 kDa); there is no pulldown blot for HA so it is impossible to know if HA-beta-actin bound to Myc-TPRN in these assays. I would encourage the authors to include a different set of blots for each pulldown, top to bottom:

- 1) Pulldown fractions blotted with anti-HA
- 2) Pulldown fractions blotted with anti-Myc
- 3) Input fraction blotted with anti-HA
- 4) Optional: Input fraction blotted with anti-Myc, but this becomes redundant if you are doing anti-Myc pulldowns and show pulldown fractions blotted for anti-Myc.

Response:

Thank you for pointing this out. We replaced Fig. 4, B, incorporated the suggested controls for all the Co-IP experiments and organized blots according to the reviewer's suggestion. A more detailed description of the panels was added to the legend of revised Fig. 4.

(5) Related to interpretation of the data in Figure 4, is HA-beta-actin polymerizing in these pulldown assays? In Figure 4C, every fragment tested by the authors appears to bind to actin. Moreover, it looks like the amount of actin coming down in each case is similar, which this reader would not expect given differences in the expression levels of Myc-TPRN constructs. Could the authors be pelleting polymerized HA-beta-actin in these experiments?

Response:

We agree. Co-IP cannot determine whether all actin molecules bound to TPRN are in a polymerized state or non-polymerized. However, we do not see HA-beta-actin pelleting in the absence of Myc-TPRN

(compare input in lane # 1 and lane #1 of Co-IP with anti-HA-antibody), which confirms that HA-beta-actin was pulled down in a complex with Myc-TPRN.

As requested by the reviewer, we performed additional experiments and replaced Fig. 4, C showing top to bottom: pulldown fraction blotted with anti-HA antibody; pulldown fraction blotted with anti-Myc-antibody; input fraction blotted with anti-HA antibody. In this experiment, the pulldown amount of HA-actin was not identical in every lane. To show variability from experiment to experiment, we provide 3 additional experiments in the supplemental Fig. S1, J-L as examples of the variable amounts of pulldown HA-actin.

(6) Figure 10 - Did the TPRN knockout stereocilia exhibit ultrastructural defects in their tapers and rootlets when imaged with FIB-SIM?

Response:

Yes! In the revised manuscript, we included new FIB-SEM data from age-matched one-month-old wild type and *Tprn*^{-/-} mice. We observed the same ultrastructural defects (breaks in F-actin) at the base of stereocilia in both *Tprn*^{-/-} and *Tprn*^{N259/N259} mice, except that these breaks were observed in every stereocilium in *Tprn*^{-/-} mice vs. ~60% of stereocilia in *Tprn*^{N259/N259} mice (reported in the previous version of the manuscript) and only in a few cells in the wildtype (see new Fig. 10, B). Both these TPRN-deficient mouse strains also have abnormal rootlets. Interestingly, rootlet abnormalities seem to progress from “kinking” at ~120° in *Tprn*^{N259/N259} mice to excessive branching (at a similar angle) in *Tprn*^{-/-} mice (Fig. 10). In the revised version of the manuscript, we concluded that “TPRN deficiency results in abnormalities of stereocilia rootlets in both *Tprn*^{-/-} and *Tprn*^{N259/N259} mice (Fig. 10, D). The differences between the rootlet phenotypes in *Tprn*^{-/-} and *Tprn*^{N259/N259} hair cells may reflect residual actin-bundling capability of just the first 259 TPRN residues (Fig. 5, G) presumably expressed in *Tprn*^{N259/N259} mice (Fig. 5, G).”

(7) Figure 10D - The authors state that TPRN ... "brings actin filaments of the stereocilia core closer together to traverse the pivot point (Fig. 10, D)." If this statement is based on a specific piece of data in this paper, the authors should point to it here. If not, this sentence should be toned down or removed.

Response:

This is our hypothesis, which is based on two observations. First, we found that, *in vitro*, actin filaments bundled by purified TPRN are closer to each other compared to actin filaments bundled by some other actin-bundling proteins, e.g., ESPN-1 (see new Fig. 3, E). Second, in the presence of TPRN, actin filaments merge together in live transfected COS-7 cells (Video 1; Fig. 3, H, white circles). Thus, we hypothesize that a similar mechanism may initiate rootlet formation, but testing this hypothesis requires a separate study. We have also modified the sentence about “bringing... actin filaments closer” accordingly.

(8) Line 534 - The authors state "...we refined our originally reported localization of TPRN (Rehman et al., 2010) and showed that TPRN is not just a ring around actin filaments as recently reported by others (Qi et al., 2024) Rather, TPRN localizes along the taper region of stereocilia forming a funnel-like shape enveloping minus-ends of peripheral F-actin and bringing more centrally positioned actin filaments closer together to incorporate them into the lower half of rootlets." Qi et al. (2024) used STED to show funnel-like localization for TPRN in the taper region in Figure 1F of their paper. The authors should edit this section to acknowledge that result. Moreover, the statement on TPRN "bringing... actin filaments closer together" is speculation and should be written as such.

Response:

We agree and edited the section on page 23, of the revised manuscript to acknowledge the similarity of the pattern between the native TPRN localization in our study and Qi et al (2024) TPRN-HA localization at stereocilia tapers. We have also modified the sentence about 'bringing... actin filaments closer'. Please also see our responses to point #1 and #7 above.

(9) Line 596 - The authors state "In summary, our data show that *in vitro*, purified TPRN alone is able to bundle F-actin into relatively loose and curved bundles." Given that there are no filament spacing measurements in this paper, and no other actin bundlers (e.g. fascin or espin) were examined as points of comparison, the "loose" and "curved" descriptors do not seem justified.

Response:

We agree and added additional control experiments to respond to the reviewer's suggestion. We purified recombinant mouse ESPN isoform 1 and showed additional negative stain TEM data in the revised version of the manuscript. F-actin was polymerized with ESPN and compared to F-actin incubated with recombinant TRX-TPRN or a TRX tag-only control. We also measured the average diameter of the bundles and the distances between actin filaments as shown in the new panel Fig. 3, E. The distances between filaments increase in the following order: control, TRX-TPRN and ESPN. These data indicate that *in vitro* TPRN bundles actin filaments more "tightly" than ESPN.

To clarify these findings, we edited the second paragraph of the "Purified recombinant TPRN bundles F-actin" subsection in the Results. Figure legends and the Method section were also updated.

Minor edits

(1) Figure 1 - The cartoon in panel B should have amino acid numbers marked along the length, at the margins of the different motifs.

Response:

We agree and have added residue numbers to the revised figure 1.

(2) Figures 4 and 5 - The authors should include a summary cartoon, perhaps as a panel at the end of Figure 5, highlighting the regions that are involved in actin binding and self-association.

Response:

We have now included a summary cartoon in Fig. 5, H, highlighting regions critical for F-actin bundling, binding and self-association/oligomerization.

(3) The dataset here is expansive and could stand to be consolidated in a few different places to increase paper readability. For example, Figure 2 and Figure 8 both show overexpression of TPRN phenotypes; this could be one figure.

Response:

Thank you for pointing to a lack of clarity in take-home messages of the original Figures 2 and 8. In fact, Fig. 2 demonstrates that "Exogenous EGFP-TPRN results in F-actin abnormalities in inner ear hair cells and in COS-7 cells", while Fig. 8 shows "Abnormal stereocilia development due to absence or over-expression of TPRN *in vivo*". The quoted sentences are the revised titles of these figures. We also consolidated some data from the original Fig. 7 to the revised Fig. 8 to better deliver the take-home message. By consolidating, we freed space in Fig. 7 to provide quantification of TPRN spots at the base

of control and PTPRQ-deficient stereocilia to show the dependence of TPRN expression on presence of its interacting partner PTPRQ as requested by reviewer # 2 question 4A.

(4) Line 426 - Replace "didn't" with "did not" (contractions might also be used elsewhere in the paper, please check).

Response:

We replaced “didn’t” with “did not” everywhere in the revised manuscript.

Reviewer #2 (Comments to the Authors (Required)):

This study by Inna A. Belyantseva and colleagues elucidates the essential role of Taperin (TPRN) in actin bundling, highlighting its critical importance in maintaining the structural stability and functionality of stereocilia. The study emphasizes that defects in TPRN can lead to hearing loss. They found TPRN to localize to the ankle region of the stereocilia and modify actin filament structures likely through bundling using a series of in vitro and in vivo experiments. Additionally, they mapped the functional domains of the protein that are important for actin binding and actin filament bundling. Functional analysis further revealed the connection between TPRN and actin bundling, utilizing mouse models TPRN^{-/-}, TPRN^{in103/in103}, TPRN^{N259/N259}, and PTPRQ^{-/-}. This comprehensive investigation examines the functional deficits caused by TPRN defects using hair cell electrophysiology and analyzes changes in stereocilia rootlet ultrastructure. Overall, this is an impressive amount of work using multiple different techniques and approaches to support their hypothesis; however, the experiments and data are not often quantified and the narrative is at times disjointed without a clear line of rationale. To strengthen and complement these findings, the following comments are suggested:

-Major comments

1. A major concern in this paper is the lack of sufficient quantification.

In figure 1O, statistical analysis was performed on the fluorescence intensity of ANKRD24, TRIOBP5, and TRIOBP4, but there is no mention of the number of stereocilia, cells, and animals from which the data was collected.

Response:

We added the requested numbers to the graphs in Fig. 1, O. The corresponding part of the legend to Fig. 1 now reads:

“(O) Diameters of ring staining for ANKRD24, TRIOBP-4 and TRIOBP-5 in the longest row (Row 1) stereocilia of OHCs. Number of stereocilia/cells: ANKRD24, n = 39/3; TRIOBP-5, n = 57/3; TRIOBP-4, n = 54/4. Asterisks show statistical significance of the differences between proteins: overall one-way ANOVA (p < 0.0001), followed by Tukey’s post-hoc comparison test (***: p < 0.0001).”

Figure 2 presents the results of TPRN transfection using a gene gun, where the images suggest low transfection efficiency, but the frequency of each phenotype upon overexpression in each tissue is not clearly provided (*line 495 - line 497).

Response:

By design, gene gun transfection is not supposed to transfect a significant number of hair cells. Instead, this technique provides unsurpassed comparison of the phenotypes in a transfected cell and a neighboring non-transfected cell in the same experimental conditions and the same location within the cochlea. As to the frequency of the effects observed in the transfected cells, we revised our text as follows (Discussion, second paragraph):

“Our data show that TPRN overexpression is cytotoxic to hair cells. In 90% of the transfected hair cells, we observed excessive actin bundling leading to over-elongated, thickened, and misshaped stereocilia and subsequent hair cell degeneration (Fig. 2, B-F). In addition, 67% of hair cells with bundle abnormalities also had abnormal nuclear F-actin structures (Fig. 2, G-J).”

Additionally, figures 3D and 5A illustrate the effects of co-localization of actin and TPRN on thickness and bundling, however quantitative analyses were not performed to prove the pattern changes across experiments (*line 565-line 569).

Response:

Following this comment, we quantified the diameters of F-actin bundles in the presence and the absence of different concentrations of TPRN and a new control, ESPN isoform 1. These new results are summarized in three new panels (Fig. 3, E-G) and described now in the second paragraph of the “Purified recombinant TPRN bundles F-actin” subsection in the Results. We repeated the experiments shown in Figure 5A at least three times and consistently obtained similar results, demonstrating that the N-terminal TPRN fragments of TPRN efficiently bundle actin filaments. Additionally, Fig. 5, G provides quantitative analysis of the ability of different TPRN fragments to crosslink actin filaments.

2. In figure 3A, the purity of TRX-TPRN is confirmed using Coomassie blue staining. However, silver staining could potentially reveal higher impurity levels, and these impurities could be affecting the experimental outcome. A minimum of silver staining of the purified protein is needed if multiple purifications steps are not performed.

Response:

We added silver staining data in Supplemental Fig. S1, H. The quality of the purified TPRN is sufficient to meet the requirements for the actin-binding assays. Based on the reviewer’s suggestion, we will use a silver staining protocol to characterize FPLC-purified TPRN for future structural analysis.

3. There is no statistical analysis in figure 6B, supplementary figure 5A and 5B

Response:

We added statistical analyses data in the revised Fig. 6, B [ABR graph] and Fig. S5, A, B, C-F. We also provided a detailed description of the statistical analyses in the Method section under subsection “Statistical analyses”. This analysis was performed by two NIDCD statisticians, Drs. Cheng and Aston, whom we now include as co-authors.

4. In figures 7H-I, the number of 3rd-5th rows is compared for IHCs between TPRN+/+ and TPRN^{in103/in103}. The 3rd row is more clearly observed in OHCs than in IHCs, so a comparative analysis of the changes in the 3rd row observed in OHCs would provide a clearer answer. If this phenotype is specific for IHCs, that would also be useful knowledge. By specifically analyzing how these changes differ

between OHCs and IHCs and how they relate to TPRN deficiency, a deeper understanding of TPRN's functional role can be obtained.

Response:

Indeed, the original Fig 7 H-I (now Fig. 8, H-I) show SEM images of stereocilia bundles in wild type and TPRN deficient (*Tprn*^{in103/in103}) IHCs and the numbers of stereocilia in different rows are quantified in Fig. 8, J also for IHCs. That said, we also reported the retraction of third row stereocilia in *Tprn*^{-/-} OHCs (Fig. 6, C-D"). Unfortunately, 3rd row stereocilia in OHCs were often behind the tallest stereocilia due to the viewing angle on SEM imaging, which prevented us from quantifying OHC stereocilia in SEM images. Instead, we quantified the number of TRIOBP spots in the third row of stereocilia (Fig. 7, A-C, current Fig. 8, A-C and Fig. S5 G-H), confirming a significant reduction of them in TPRN-deficient hair cells. More than 34 cells per group were quantified and more than three mice per group were characterized. These results demonstrate similar (nearly identical!) morphological defects in both IHCs and OHCs. We added this quantification data to the revised Fig. S5, H.

4A) And in figures 7M-P, TPRN expression in OHCs and IHCs is compared in PTPRQ KO mice. The results show a tendency for reduced localization of TPRN in PTPRQ^{-/-}, but quantitative analysis of the number of puncta or fluorescence intensity across different cells is needed, since it is unclear which is the phenotype the authors are indicating is the cause of the decreased intensity.

Response:

We quantified the fluorescence intensity across different cells, incorporated the results in revised Fig. 7, C-E and H-J and its legend and describe the quantification procedure in the revised Method section under a subtitle "Quantification of immunofluorescent puncta using Imaris and ImageJ".

5. Additionally, figure 8A shows the results of AAV injection. While this method demonstrates higher efficiency than the gene gun used in figure 2, it would be preferable to present the changes observed in multiple cells rather than just one. Furthermore, quantitative analysis of the frequency of degeneration or over-elongated bundles is necessary to provide the reader with an indication as to the prevalence of these phenotypes.

Response:

The original Fig. 8, A is now Fig. 8, L-L'. We agree and provide an additional panel K-K' to the updated Fig. 8 displaying more wild-type OHCs and IHCs after AAV infection with full-length TPRN and added the following quantification of the results in revised Fig. 8, K-K' legend. "On average, 64.9 ± 6.5% of IHCs exhibited excessive stereocilia elongation. A total of three mice were analyzed and 27-64 IHCs were assessed in the apical region of the cochlea of each mouse."

5A) For the changes shown in figures 8H-I, it is also important to quantitatively analyze how many cells, across how many animals, and how frequently overgrown or extra short-thick stereocilia are observed.

Response:

The original Fig. 8, H-I is now Fig. 8, R-S. We have added the quantification data to the legend of Fig. 8 as follows:

"On average, 5.5 ± 2.1% of IHCs exhibited negligible changes to the hair bundle, 87.4 ± 2.7% exhibited supernumerary rows of short stereocilia with increased thickness (R'), and 7.0 ± 2.9% exhibited over-

elongated stereocilia (S). A total of five mice were analyzed and 32-41 IHCs were assessed in the apical region of the cochlea of each mouse. Scale bars: 5 μm in A-Q, except 1 μm in H-I and R-S.”

6. In Figure 9E - 9F, the peak current size of the outer hair cells is smaller than typically expected and shows a large variance (ranging from at least 0.3 nA to 0.8 nA). This variance is not indicative of the biological factors and likely due to experimenter variability.

Response:

Response: Yes, and this is exactly why we have shown all the data! Cell-to-cell variability of the maximal MET current is a common phenomenon of all MET current studies (see, for example, even larger variability of MET currents from almost zero to ~ 1 nA in 52 different inner hair cells in Figure 4b from Beurg et al., *Nat. Neurosci.*, 2009). The most likely reason for this variability is the mechanical damage to the bundle (and/or tip links) during dissection and/or bundle deflection by the stimulating device. Therefore, in many reports, various groups (including us, see Stepanyan and Frolenkov, *J. Neurosci.*, 2009) select only the cells with a MET current above a certain threshold. This is an acceptable practice but only when changes in bundle fragility are not expected. Since TPRN deficiency may change the fragility of the hair bundle, we believe that group-to-group comparison in this study requires showing all technically successful MET recordings. On page 47, the revised Methods section now describes it as following: “Since TPRN deficiency could make the hair bundles more fragile resulting in increased cell-to-cell variability in the MET current amplitudes, our criteria for a “technically successful” recording did not include minimal MET current. We included in statistical analysis all recordings from hair cells that had uncompensated series resistance of <25 MOhm and membrane resistance of >250 MOhm (determined by pClamp “membrane test”).” See comment 6.2 below for clarification what our data means for the hair bundle fragility.

6.1 The variability will make it difficult to assess open probability as well since lower current sizes may bias open probability measurements. This makes it difficult to assess any changes that may be occurring in the mutant lines if there is so much uncontrolled variability in the WT data.

Response:

We are puzzled by the comment. We are not aware of any study that would experimentally demonstrate the link between the amplitude of MET current and open MET channel probability at resting bundle position (Popen). Maximal MET current usually indicates how many functional MET channels operate in a bundle, while Popen indicates the state (resting tension) of these remaining functional MET channels. Those are two independent measurements, often uncorrelated. There are several reports demonstrating unchanged or even increased Popen in hair cells with almost ten times decreased MET current, see some recent examples in Fig. 8, A-B (Liang et al., *Neuron*, 2021) or in Fig. 3, G-H (Underhill et al., *PNAS*, 2024).

6.2 Peak current data for TPRNN259/N259 and TPRN^{-/-} are different even though both are thought to be knockouts, warranting some explanation.

Response:

We would also like to know the answer to this question but, unfortunately, we do not. In the revised version of the manuscript, we discussed these data as follows: “As an F-actin bundler at the base of stereocilia, TPRN is well suited for controlling actin/myosin-based transport of various stereocilia proteins, including components of MET machinery. However, our data show that MET currents are largely

unaffected, at least in *Tprn*^{-/-} OHCs. However, it may be still possible in *Tprn*^{N259/N259} OHCs that have decreased MET current, perhaps due to abnormal F-actin bundling in the rootlets of these mice by the truncated TPRN. The alternative explanation of the decreased MET current in *Tprn*^{N259/N259} hair cells would be an increased fragility of their hair bundles. However, it is not supported by the fact that we were not able to detect any obvious effects of TPRN deficiency on pivot stiffness of stereocilia, at least in young postnatal OHCs. Furthermore, in contrast to our previous observations in TRIOBP-4/5 deficient hair cells (Kitajiri et al., 2010), TPRN-deficient OHC stereocilia are not excessively fragile and can withstand intensive repetitive deflections.”

6A) Additionally, the stiffness shown in Figure 9G focuses only on the fluid jet area. One of the factors that significantly influence force on the hair bundle is the Z-axis position of the fluid jet, but these considerations are not mentioned to be controlled. Also, as mentioned in the methods, the distance between the fluid jet and the hair bundle is 6 - 10 μm . This distance can cause $\sim 60\%$ reduction in force (Tobin et al., 2019), further indicating that these experiments are poorly controlled. Considering these factors, additional experiments are required for accurate assessment of the current size and stiffness.

Response:

We apologize for a generic description of the experimental procedure in our original submission. In fact, positioning of the pipette relative to the hair bundle was carefully controlled. We have added a new inset in Fig. 9, A to illustrate the accuracy of fluid-jet positioning in our experiments. We also revised the Methods section on page 48 to describe positioning of the fluid-jet pipette as follows: “Before the experiment, the fluid-jet was positioned in front of a hair bundle at distance of $\sim 4.5 \mu\text{m}$ and in such a way that the diameter of the pipette was aligned in Z axis with the image of the tip of the bundle (see Fig. 9, A, inset). This ensured Z positioning of the fluid-jet with an error of $\pm 340\text{nm}$, the half-width of the point spread function of the microscope objective in the Z direction. During the experiment, high-speed video recordings of the bundle movements always included the image of the fluid-jet pipette (as shown in Fig. 9, A, inset), which allowed us to measure the actual distance between the pipette and the bundle (3.6-5.0 μm throughout all experiments).”

In addition, on page 49, we added a new section in Methods describing the measurements of hair bundle displacements and their relationships with fluid-jet positioning as follows: “Within variabilities of fluid-jet stimulation parameters in our experiments, we did not find a correlation of hair bundle compliance with the fluid-jet pipette-to-bundle distance (Pearson coefficient = 0.39, $p=0.08$) but did find a strong correlation with the area of the fluid-jet pipette (Pearson coefficient = 0.66, $p=0.001$).” The bottom line, we did try our best to control fluid-jet stimulation as accurately as possible and even obtained some data indicating potential decrease in hair bundle compliance in *Tprn*^{-/-} OHCs (note that all magenta points are at the bottom of Fig. 9, G). However, we believe that fluid-jet stimulation is too crude to accurately assess potential effects of TPRN deficiency on hair bundle stiffness and, therefore, are developing different techniques for future studies. This conclusion is now explicitly stated in the Results as follows: “We concluded that any potential effects of TPRN-mediated F-actin crosslinking at the stereocilia pivot points on mechanical stiffness of the hair bundles may be too subtle to be detected by a fluid-jet technique”.

7. In figure 10C, the observed F-actin disruption and kinked rootlet quantification is done only in 2 animals for each genotype and these experiments should be performed on a minimum of 3 animals (biological replicates).

Response:

Also following comments of Reviewer #1, we now provide new FIB-SEM data on the age-matched wild type and *Tprn*^{-/-} IHCs (see revised Fig. 10). These new data bring the number of control and mutant animals to three. That said, we should point out that EM imaging of serial sections in fast-frozen and

freeze-substituted samples is a time-consuming and expensive technique, independent on whether it is performed with “classical” ultramicrotome sectioning or with FIB-SEM. Even with FIB-SEM, each sample requires at least a couple of months of work and costs several thousands of dollars in FIB-SEM user fees (unless imaged at a more typical low resolution). This is why the focus of these EM techniques is always on abnormal phenotypes that are rarely seen in the wildtype, i.e. broken and/or abnormal rootlets in Fig. 10. Thus, serial section EM studies very often have n=1 animals (for examples, please see Wood et al., *JARO*, 2021; or Kantardzhieva et al., *J. Comp. Neurol.*, 2013).

8. In the discussion, there are statements as if the data was shown related to TPRN function, however, many times the data in current form do not prove function. Especially, there are over-interpreted sentences including line 468 - 471, 495 - 497, 536 - 539, 548 - 549, 573 - 575, 598 - 599. Please modify the language to indicate these are hypotheses based on actual data shown in the manuscript.

Response:

The sentences were modified accordingly in the revised manuscript.

8A) Also, in line 472 - 473, the statement "TPRN deficient rootlets are thinner" lacks quantitative evidence of the rootlet thickness.

Response:

We have quantified the diameters of the rootlets and provided the data in new panel (Fig. 10, C).

Minor comments

1. Line 290-291 should be more precise to avoid confusion. The original sentence could imply both a lack of TPRN and a lack of antibody specificity. Therefore, it should be rewritten to remove any confusion.

Response:

We thank the reviewer for pointing to a confusing sentence. It has been revised as follows: “This 80 kDa band was absent in tissues from *Tprn*^{-/-} littermates, confirming a lack of TPRN and establishing antibody specificity (Fig. S1, D).”

2. In figure 10D, tip link breakage is observed in Taperin-deficient cells. Additional information/quantification is needed to determine whether these findings align with results from tip links observed in different biological replicates across various mice. The authors have argued that peak current is not altered in the TPRN^{-/-} (figure 9E), therefore the loss of tip links in the model is surprising.

Response:

The reviewer correctly noticed that the peak current was not altered in the *Tprn*^{-/-} mice (Fig. 9, E) as measured at P4-P7. However, the schematic illustrates the adult stereocilia bundle changes in *Tprn*^{-/-} mice with progressive degeneration of hair bundles due to abnormalities of rootlets. Later in the adult, the loss of tip-links is expected due to shortening and even the disappearance of some stereocilia from the shorter rows (see also Fig. 6, E-F) which is reflected in our schematic in Fig. 10, D. To avoid confusion, we added “adult wild type and “adult TPRN deficient hair cell” to panel 10, D and in the figure legend.

3. In the description of supplementary figure 1 (line 80), the antibody PB913 was mistakenly labeled as BP913. Also, in line 472, there is a spelling error; "TROBP" should be corrected to "TRIOBP". In line 531, 585, there is a spelling error; "TRPN" should be corrected to "TPRN".

Response:

Spelling errors were corrected.

4. In line 422 - 424 (Fig. 9A), the sentence "We observed similar MET responses to step deflections of stereocilia" does not make sense. Fluid jet does not give 'step-deflection'.

Response:

We are puzzled. A fluid-jet generates a stimulus proportional to the driving voltage. If this command voltage is step-like, a fluid-jet does generate a step-like bundle deflection. Obviously, the rise time of these "steps" is slower using a fluid-jet compared to responses evoked by a piezo-driven rigid probe, but they are still "steps". Many groups use fluid-jet to generate step-like deflections of the hair bundle (for examples, please see Mecca et al., *PNAS*, 2022; or Corns et al., *PNAS*, 2014).

5. In line 475 - 477, figure 7 does not mention the gradual disappearance of ANKRD24.

Response:

Thank you for noticing this omission. The correct figure to show the disappearance of ANKRD24 is Figure S5, panels I-J. The revised sentence now reads "The loss of F-actin bundling by TPRN results in a gradual disappearance of TRIOBP-5 and ANKRD24 and retraction of the shortest stereocilia rows (Fig. 8, A-J and Fig. S5, G-J', correspondingly)."

And in line 477 - 479, there is a significant loss of MET current in TPRN^{N259/N259} hair cells.

Response:

The sentence was corrected to clarify the loss of MET current in P7 *Tprn*^{N259/N259} hair cells while no changes of MET currents were observed in P7 *Tprn*^{-/-} hair cells.

Reviewer #3 (Comments to the Authors (Required)):

The study "Taperin bundles F-actin at stereocilia pivot-points enabling optimal lifelong mechanosensitivity" by Belyantseva* and Liu* et al 2024 investigates the role of taperin (TPRN) in stereocilia structure and its contribution to hearing in mice. Taperin (TPRN) is found at the base of stereocilia and helps to stabilize these structures at pivot points. Lack of TPRN leads to structural degeneration, with warped rootlets and a gradual loss/localization of additional (previously characterized) proteins (PTPRQ, TRIOBPs, and ANKRD24). In contrast, TPRN overexpression causes excessive F-actin bundling and abnormal stereocilia growth. Purified (but still tagged) TPRN protein seems to directly bundle actin filaments in pelleting assays and fluorescence-based assays, through residues present in the first 1-170 amino acids. The authors have done a substantial amount of work successfully combining assays with purified proteins, cells, and animal models, to provide a very complete picture of TPRN's role in stereocilia structure and function. However, the work is missing several controls as presented and I have a few minor comments as detailed below:

Major comments:

The authors provide a GFP- alone control but unless I missed it, I did not explicitly see a control indicating the functionality of tagged TPRN (GFP- or TRX-tags). The authors provided a GFP- alone control but not the necessary version lacking the tag. It seems very addressable due the prominent bundling activity seen in cells and in vitro - where it may be likely that elevated bundling activity can be seen by actin stain

alone. While the authors mention purification issues in the methods, an untagged version of the protein is also important for the interpretation of the results. Can the solubility increasing tag be cleaved at a later purification step to make an untagged version of the protein?

Response:

The Editor and Reviewer #3 raised a question on solubility and stability of untagged full-length mouse TPRN that we tried to solve for a long time. Over a four-year period, two of our laboratories independently tried to purify full-length mouse taperin (TPRN) using various expression systems. We tried expressing full-length mouse TPRN with or without various fusion proteins and tags in Sf9 insect cells, HEK293 cells, and various strains of *E. coli*. Multiple tags and fusion proteins were used with and without codon optimization: EGFP and FLAG tag in Sf9 and HEK 293 cells; maltose binding protein (MBP), glutathione S-transferase (GST), Strep-tag, EGFP, His₆-tag, HA-tag, and thioredoxin (TRX) in *E. coli*. Except for TRX-TPRN expression in *E. coli*, no other technique provided enough stable and functional TPRN protein for experiments *in vitro*. We hypothesized that untagged TPRN is likely to be unstable *in vitro*. To test this hypothesis, we removed the TRX tag from TRX-fused taperin using a thrombin protease (see new panel I in Fig S1) and no untagged TPRN was detected, suggesting that it is indeed unstable *in vitro*.

In the revised Method section, we added the following sentence. "TRX tag removal using thrombin protease (69-022-3, Fisher Scientific) failed to obtain untagged FL TPRN, suggesting that untagged TPRN is unstable *in vitro*."

The demonstration that TPRN is overexpressed should be further clarified or validated with a comparison to endogenous levels.

Response:

We would like to point out that epitope-tagged TPRN (GFP-TPRN or HA-TPRN) was introduced via either a biolistic Helios gene gun transfection or AAV delivery into wild type hair cells. Therefore, by design, these techniques overexpress TPRN, since EGFP-TPRN is expressed under a very potent CMV promoter of the expression vector. Of course, there is a theoretical possibility of compensatory downregulation of endogenous TPRN after introduction of a TPRN expression vector. Two of our laboratories independently overexpressed TPRN in hair cells using different methods (gene gun transfection or AAV delivery) and observed similar results: abnormal actin remodeling, F-actin over-elongation, and degeneration of stereocilia. Our data are in line with observations reported by Qi et al (2024) showing that their AAV delivered TPRN-HA expressed under a potent CMV promoter leading to TPRN overexpression does not restore hearing but causes stereocilia degeneration.

The authors are using this image in Figure 3C to claim the localization of TPRN occurs along filaments rather than at ends, however it is difficult to know for sure since it does not appear to be a single filament. Further, the authors do not specify why they might expect the localization to be on an actin filament end. This image should be replaced with something more representative.

Response:

Following the Reviewer's comment, we've replaced original Fig. 3C with an image that includes more short actin filaments (now Fig. 3D). We also modified the sentence and added length measurements to this sentence as follows: "TRX-TPRN was localized along the length of actin bundles (Fig. 3, C, right) or thinner filamentous structures (Fig. 3, D) likely representing single filaments with lengths (10.27 ± 0.42 μm , $n=55$) comparable to those of single actin filaments shown in the left panel of Fig. 3, B (10.34 ± 0.36 μm , $n=96$). This localization pattern differs from that of phostensin, a minus-end actin-capping protein that shares some sequence similarity with TPRN (Kao et al., 2007; Lai et al., 2009)."

The schematic noting the TPRN constructs capable of binding and bundling actin filaments can be improved by adding + and - notations summarizing the activities of each truncated protein.

Response:

We have now included a summary cartoon as in Fig. 5H and use "+" notation to indicate the strength of bundling activity of different TPRN fragments.

The blots present in Figure 4 do not display molecular markings in accordance with JCB guidelines and will need to be re-cropped to have markers above and below the bands of interest to comply.

Response:

Done. We indicated the molecular weights above and below the bands on the gel images and provided uncropped gel images submitted together with the revised manuscript to the Journal of Cell Biology.

Minor comments:

Throughout the work the authors refer to "loose" TPRN bundles. While the intention is likely to make a comparison between TPRN and other well characterized bundling proteins like fascin and alpha actinin, this terminology somewhat confusing and should be further clarified.

Response:

Thank you for your comments. We addressed this issue in response to question 9 of Reviewer #1 evaluating the gross morphology, the average diameter of bundles and the distances between actin filaments with a new control, ESPN isoform 1. The results are described in the second paragraph of the "Purified recombinant TPRN bundles F-actin" subsection in the Results section. The Fig. 3 legend and Method section were also updated.

The color choices and size of the panels in Figure 1, specifically Figure 1D, makes it difficult to discern how much overlap is present with TPRN and ANK. It would be helpful to potential readers to change the colors to better show this overlap and to add additional line scans, particularly for the comparison of signals in 1C-F).

Response:

We agree. To clarify the overlap of TPRN and ANKRD24, we added two of line scans to show overlap of TPRN and ANKRD24 (Fig. 1, E') and ANKRD24 and TRIOBP-5 (Fig. 1, F'). We also changed the colors in Fig. 1, F. For consistency with other panels in this figure, we would like to preserve green and magenta for TPRN and F-actin, respectively. In this case, only replacing blue for ANKRD24 to red color in panel E would give a clear distinction between colors as many other combinations of colors did not work. However, this is against the JCB rule to avoid green-red combinations. So, we kept the colors for Fig. 1, E but added a line scan (panel E') to show the overlap.

Can the authors comment on whether the nuclear TPRN (particularly in the punctate state) in Figure 2G-K? If it is, perhaps additional biophysical properties are involved in mediating TPRN's bundling activities.

Response:

This is indeed an interesting question. TPRN is predicted to be a largely disordered protein. Therefore, it may form a multivalent molecular condensate in the nucleus, interacting with CHD4 (Bird et al., 2017) and forming protein complexes for DNA damage repair (Ferrar et al., 2012). It was reported that DNA damage stimulates actin filament formation in the nucleus (Belin, Lee and Mullins, eLife, 2015), and TPRN, having bundling ability, may promote the formation of actin bundles. Obviously, this possibility requires its own investigation. On page 21, we have briefly discussed it in the subsection entitled "Possible function of TPRN in hair cell nuclei".

Do the authors know or were they able resolve how many actin filaments are in the TPRN bundles in Figure 3D? What about their orientations (i.e., does TPRN make parallel, anti-parallel, or mixed bundles)

Response:

The reviewer raised another good point. In our *in vitro* experiments we do not know if the bundles are parallel, anti-parallel or mixed filaments. We can visualize coalescence of short actin filamentous structures in COS-7 cell nuclei (See Video 1) to form thicker bundles, but we cannot discriminate single actin filaments. However, in hair cell stereocilia, actin filaments are known to be organized in parallel unidirectionally oriented bundles with barbed ends at the distal ends (tips) of stereocilia. TPRN concentrates at the base of stereocilia where the pointed ends of the peripheral actin filaments are located. TPRN has two regions of similarities to phostensin, a minus-end actin capper. This similarity leads to expectation that TPRN might bind preferentially at minus-end of actin filaments/bundles. However, TPRN localizes along the polymerized filamentous actin *in vitro*. Even in hair cell stereocilia where endogenous TPRN localizes at the tapered base, when overexpressed, TPRN can mislocalize to stereocilia tips and along abnormal thin filamentous structures extending from the tips of stereocilia (Fig. 2, B-F). It is hard to discern if there are short actin filaments with pointed ends at stereocilia tips (Liao et al., Research Square, preprint, 2024) or filaments that emanate from the tips have pointed ends distributed along the length, or if some thin filamentous structures decorated by TPRN *in vitro* represent the single actin filaments or thin bundles. A separate study is required to answer these questions.

Thus far, regarding the number of actin filaments, it is challenging to count filaments because actin filaments can form three-dimensional bundles. Instead, we measured the diameters of bundles and distances between filaments as described above for question 9 of the Reviewer 1 and minor point 1 of the Reviewer 2. We also agree that the orientation of filaments is a crucial parameter that can be evaluated by decorating with myosin heads (Tilney et al, JCB, 1980). We will explore this parameter in a future study.

The data present in Figure 9 is cool, but it does not need to be a full-page figure. Perhaps this information can be consolidated with other data to reduce the total number of figures according to the JCB limit/guidelines.

Response:

We are not planning to have Fig. 9 as a full-page. We are preparing it as a half-page figure. It does show two crucial take-home messages: i) the effects of TPRN deficiency on stability of shorter row stereocilia are MET-independent; and ii) the effects of TPRN deficiency on mechanical properties of stereocilia bundles are different from the previously published effects of a lack of TRIOBP-4/5 or TRIOBP-5.

In a similar vein, the localization studies with TRIO proteins and ANKRD24 contain stunning images but they do not contribute much to the story beyond putting it in the context of past work and the

localization/connection with TPRN. These figures can likely be consolidated or moved into the supplement.

Response:

We thank the reviewer for his suggestion. To clarify the take-home messages from different figures and to consolidate the data, we combined Fig. 8 with some panels from Fig. 7 and moved some panels to the supplement. That said, we believe it is important to show the localization of TPRN relative to TRIOBP and ANKRD24 because:

1. The absence of TPRN causes a loss of TRIOBP5 from rootlets of third row stereocilia early in development (Fig. 8).
2. TRIOBP recruits ANKRD24 to stereocilia rootlets (Krey et al., 2022) and ANKRD24 is also lost from the third row stereocilia in TPRN-deficient mice (Fig. S5).
3. This is the first report of TRIOBP-4 localizing to the smaller diameter rings compared to TRIOBP-5.
4. The ring-like pattern of TRIOBP-5 and ANKRD24 localization was previously shown only in IHCs (Krey et al., 2022). However, in *Tprn*^{-/-} mice, TRIOBP-5 loss occurs in both IHCs and OHC stereocilia. Therefore, it is crucial to show localization of these proteins in OHC rootlets that was not reported previously.

March 24, 2025

RE: JCB Manuscript #202408026R

Bo Zhao
Indiana University School of Medicine

Dear Dr. Zhao,

Thank you for submitting your revised manuscript entitled "Taperin bundles F-actin at stereocilia pivot-points enabling optimal lifelong mechanosensitivity." We would be happy to publish your paper in JCB pending final revisions necessary to meet our formatting guidelines and address the remaining reviewer comments (see details below).

As you will see Reviewer #2 asks for several text changes to tone down a few claims and more clearly acknowledge discrepancies and caveats. The reviewer also notes that the data does not show that TRIOBP-4, TRIOBP-5, ANKRD24, & F-actin colocalize but rather that these protein have similar signal patterns in different hair bundles. If it is possible we encourage you to add a final experiment showing co-localization of these proteins. But if this is not feasible or technically challenging then please revise statements about these colocalizations in the text.

A. MANUSCRIPT ORGANIZATION AND FORMATTING:

1) Text limits: Character count for Articles is < 40,000, not including spaces. Count includes title page, abstract, introduction, results, discussion, and acknowledgments. Count does not include materials and methods, figure legends, references, tables, or supplemental legends.

2) Figure formatting: Articles may have up to 10 main text figures. Scale bars must be present on all microscopy images, including inset magnifications. Molecular weight or nucleic acid size markers must be included on all gel electrophoresis. Please add scale bars to Figures 1D/N and the inset magnifications in S3.

Also, please avoid pairing red and green for images and graphs to ensure legibility for color-blind readers. If red and green are paired for images, please ensure that the particular red and green hues used in micrographs are distinctive with any of the colorblind types. If not, please modify colors accordingly or provide separate images of the individual channels.

3) Statistical analysis: Error bars on graphic representations of numerical data must be clearly described in the figure legend. The number of independent data points (n) represented in a graph must be indicated in the legend. Please, indicate whether 'n' refers to technical or biological replicates (i.e. number of analyzed cells, samples or animals, number of independent experiments). If independent experiments with multiple biological replicates have been performed, we recommend using distribution-reproducibility SuperPlots (please see Lord et al., JCB 2020) to better display the distribution of the entire dataset, and report statistics (such as means, error bars, and P values) that address the reproducibility of the findings.

Statistical methods should be explained in full in the materials and methods. For figures presenting pooled data the statistical measure should be defined in the figure legends. Please also be sure to indicate the statistical tests used in each of your experiments (both in the figure legend itself and in a separate methods section) as well as the parameters of the test (for example, if you ran a t-test, please indicate if it was one- or two-sided, etc.). Also, if you used parametric tests, please indicate if the data distribution was tested for normality (and if so, how). If not, you must state something to the effect that "Data distribution was assumed to be normal but this was not formally tested."

4) Materials and methods: Should be comprehensive and not simply reference a previous publication for details on how an experiment was performed. Please provide full descriptions (at least in brief) in the text for readers who may not have access to referenced manuscripts. The text should not refer to methods "...as previously described." Please also indicate the acquisition and quantification methods for immunoblotting/western blots.

5) For all cell lines, vectors, constructs/cDNAs, etc. - all genetic material: please include database / vendor ID (e.g. Addgene, ATCC, etc.) or if unavailable, please briefly describe their basic genetic features, even if described in other published work or gifted to you by other investigators (and provide references where appropriate). Please be sure to provide the sequences for all of your oligos: primers, si/shRNA, RNAi, gRNAs, etc. in the materials and methods. You must also indicate in the methods the

source, species, and catalog numbers/vendor identifiers (where appropriate) for all of your antibodies, including secondary. If antibodies are not commercial, please add a reference citation if possible. Please double check the list of antibodies, it does not include an anti-TRIOBP-4 antibody currently.

6) Microscope image acquisition: The following information must be provided about the acquisition and processing of images:

- a. Make and model of microscope
- b. Type, magnification, and numerical aperture of the objective lenses
- c. Temperature
- d. Imaging medium
- e. Fluorochromes
- f. Camera make and model
- g. Acquisition software
- h. Any software used for image processing subsequent to data acquisition. Please include details and types of operations involved (e.g., type of deconvolution, 3D reconstitutions, surface or volume rendering, gamma adjustments, etc.).

7) References: There is no limit to the number of references cited in a manuscript. References should be cited parenthetically in the text by author and year of publication. Abbreviate the names of journals according to PubMed.

8) Supplemental materials: Articles may have up to 5 supplemental figures and 10 videos.

Please also note that tables, like figures, should be provided as individual, editable files. A summary of all supplemental material should appear at the end of the Materials and methods section. Please include one brief sentence per item. In Figure S1 panels J-L are very small and may be difficult to read, please consider enlarging these.

9) Video legends: Should describe what is being shown, the cell type or tissue being viewed (including relevant cell treatments, concentration and duration, or transfection), the imaging method (e.g., time-lapse epifluorescence microscopy), what each color represents, how often frames were collected, the frames/second display rate, and the number of any figure that has related video stills or images.

10) eTOC summary: A ~40-50 word summary that describes the context and significance of the findings for a general readership should be included on the title page. The statement should be written in the present tense and refer to the work in the third person. It should begin with "First author name(s) et al..." to match our preferred style.

11) Conflict of interest statement: JCB requires inclusion of a statement in the acknowledgements regarding competing financial interests. If no competing financial interests exist, please include the following statement: "The authors declare no competing financial interests." If competing interests are declared, please follow your statement of these competing interests with the following statement: "The authors declare no further competing financial interests."

12) A separate author contribution section is required following the Acknowledgments in all research manuscripts. All authors should be mentioned and designated by their first and middle initials and full surnames. We encourage use of the CRediT nomenclature (<https://casrai.org/credit/>).

13) ORCID IDs: ORCID IDs are unique identifiers allowing researchers to create a record of their various scholarly contributions in a single place. Please note that ORCID IDs are required for all authors. At resubmission of your final files, please be sure to provide your ORCID ID and those of all co-authors.

14) JCB requires authors to submit Source Data used to generate figures containing gels and Western blots with all revised manuscripts. This Source Data consists of fully uncropped and unprocessed images for each gel/blot displayed in the main and supplemental figures. For assays performed using capillary electrophoresis and/or immunoassay-based detection, authors should instead provide the electropherogram graph(s) for each experiment, plotting fluorescence/chemiluminescence intensity vs. molecular weight/size. Since your paper includes cropped gel and/or blot images, please be sure to provide one Source Data file for each figure gels, blots, and/or capillary electrophoresis assays along with your revised manuscript files. File names for Source Data figures should be alphanumeric without any spaces or special characters (i.e., SourceDataF#, where F# refers to the associated main figure number or SourceDataFS# for those associated with Supplementary figures). For traditional gels and blots, the lanes of the gels/blots should be labeled as they are in the associated figure, the place where cropping was applied should be marked (with a box), and molecular weight/size standards should be labeled wherever possible. For capillary electrophoresis assays, each trace in the graph should be color-coded and labeled to indicate which protein, gene, or sample is being measured (please try to avoid red/green combinations to accommodate our color-blind readers).

Source Data files will be directly linked to specific figures in the published article. Source Data Figures should be provided as individual PDF files (one file per figure). Authors should endeavor to retain a minimum resolution of 300 dpi or pixels per inch. Please review our instructions for export from Photoshop, Illustrator, and PowerPoint here: <https://rupress.org/jcb/pages/submission-guidelines#revised>

15) Journal of Cell Biology now requires a data availability statement for all research article submissions. These statements will

be published in the article directly above the Acknowledgments. The statement should address all data underlying the research presented in the manuscript. Please visit the JCB instructions for authors for guidelines and examples of statements at (<https://rupress.org/jcb/pages/editorial-policies#data-availability-statement>).

B. FINAL FILES:

-- A response to reviewer comments document (PDF is acceptable).

Thank you for your attention to these final processing requirements. Please revise and format the manuscript and upload materials when ready.

Thank you for this very interesting contribution, we look forward to publishing your paper in Journal of Cell Biology.

Sincerely,

Pekka Lappalainen, PhD
Monitoring Editor
Journal of Cell Biology

Dan Simon, PhD
Scientific Editor
Journal of Cell Biology

Reviewer #1 (Comments to the Authors (Required)):

The authors did a nice job addressing my concerns and suggestions with edits to narrative, additional explanation, reorganization of figure panels, and in several places, new data. The new FIB-SEM images are stunning. Overall, I view this as a strong revision, and I have no further suggestions for improving the work.

Reviewer #2 (Comments to the Authors (Required)):

In the revised manuscript, Belyantseva et al. have made some admirable improvements to the manuscript. The added control in Figure 3 is a strong addition, the new data regarding Tprn -/- in the figure 10 is excellent, and the increased quantification adds clearer support of the data presented. The authors provided many great improvements and additions, however, my impression

is that the overall conclusion of the manuscript is still not following all the data presented, likely due to what appears to be over interpretation of the data in some cases. The authors are proposing the idea that Taperin is required for the proper formation of the rootlet in development, but no data are provided that at the EM level for rootlets when they are initially formed (~P7 based on Kitajiri et al., 2010). I do appreciate that the authors are careful in stating that much of the ideas come from their in vitro work. However, the data rather appear to support the other part of their hypothesis/conclusion that taperin is required for the proper maintenance of rootlets and for stability of the rootlets in vivo. Even at the age of P17 for N259 IHC (figure 10A), the rootlet is normally formed at the upper portion and traverses the narrowing of the stereocilia pivots. All the abnormalities occur at the pivot point (cracks) and lower rootlet, suggesting other functions besides bringing the rootlet down from the espin cross-linked inter-filament distance to the rootlet filament density. These data all suggest that the authors second conclusion of "durable deflections of stereocilia by sound throughout a lifetime" is more fitting. Overall, the data are strong, but the interpretation and the conclusions drawn are often claimed as definitive, when there are discrepancies and irregularities. The text needs to be carefully edited to not overstate what can be concluded from the data.

Major:

The text is still a bit overinterpreted in my opinion. Line 149: This summary is not quite a demonstration, since stereocilia structure is not necessarily stabilized by taperin when overexpression also causes stereocilia instability. How was it shown that the stereocilia changes are not secondary to other dysfunction that causes homeostatic changes in the stereocilia structure? Optimizing mechanosensitivity and sound transduction is also not demonstrated, since Tprn ^{-/-} mice show no changes in mechanosensitivity. How was it shown that sound transduction is optimized for "sustained sound-induced stimuli?" Some of these are not really supported by the data and others are hypotheses of the functional model, not a "demonstration."

Figure 1G-H: In the rebuttal, the authors make a point about the TRIOBP-4 staining being important and new, but the images presented here are not even co-localized with each other and are of different hair bundles. The conclusion that the F-actin filaments are at the center of the "ring-like" structure of TRIOBP-4 is not fully supported in this imaging since there is no co-localization. In terms of referring to G-J and O-P, it should be stated that the assumption is that these are rings forming around the F-actin rootlet, but there is currently no data to confirm the strict co-localization/structure of TRIOBP 4 and 5 with each other or with the F-actin and ANKRD24. The only data showing the co-staining appears to be with TPRN and F-actin.

Figure 2: It may actually be more helpful to state the numbers of cells exhibiting each of the displayed morphologies shown as well. The conclusions drawn from this figure (line 204-206) still appear to be an over-interpretation, since the data themselves about the 1-260 residues being sufficient for actin bundling do not observe that same behavior in the AAV infected hair cells in figure 8. The statement regarding targeting sequence between 260 and 730 is based on constructs 1-260 and 730-749 not having specific rootlet targeting, but a "targeting sequence" could exist straddling those constructs and the 260-730aa region or could require the first 260 aa and the last 19 aa connected by some linker. Additionally, full length expression of TPRN does not only localize the protein at the taper region, overexpressed FL TPRN also localized elsewhere, so one could argue that the 1-260 aa construct also targets the protein to the taper, but the overexpression causes localization to other parts of the stereocilia as well. The data more aptly "suggests" rather than "concludes."

Figure 3: The addition of the quantification and control make these experiments much more complete. It is interesting that the distance between filaments is similar in the TRX only as compared to the TPRN. The images in Panel E make it look as if filaments are further apart in TRX only and closer together in TPRN expressed. I wonder if the separation of the filaments is more difficult when they are closer together.

Figure 9: The increase in Popen is exactly the bias that would be expected. As calcium is a key regulator of MET, a reduction in the peak current reduces the resting current and hence calcium in the stereocilia, where the cell compensates by increasing the open probability to increase the intracellular calcium concentration (like a homeostasis). While I can understand the value in including all the data when indicating the peak current, the variability in the number of intact tip-links makes it difficult to assess whether any other changes are actually occurring. So analysis of other aspects need to account for the peak current variability. For instance, the stiffness of the hair bundle is affected by the number of intact tip-link. When breaking all tip-links with a calcium chelator, the stiffness of the hair bundle is significantly reduced (see Tobin et al., 2019). When comparing stiffness across hair bundles, it would have been better to break all tip-links such that there is a cleaner measure of the stiffness of the stereocilia pivots without the variability of the number of tip links contributing. There is also still a remaining issue with the positioning of the fluid jet in relation to the hair bundle. The distance from the hair bundle ranges from 3.6-5um, but from Tobin et al. (2019) a 1.4 fold change in the distance results in a 30% difference in the force delivered, contributing more uncontrolled variability in the data. If improvements to the quality of the experiments is not done, it would be better to move this data to the supplement since there are clear caveats in the interpretation of the data as presented.

Figure 10: It is interesting that the phenotype looks more severe for the Tprn^{-/-} IHCs, but the ABR at P18 suggests N259 is more severe, the MET measurements suggest that the N259 is more severe as well, and the SEM data in figure 6 suggests that N259 is more severe since 6E is of P30 Tprn^{-/-} and it is comparable to 6F of P17 N259. A more thorough discussion of how these phenotypes may differ based on the potential function of the 1-259 version of taperin may be helpful to put all of these results in context. How can that expression potentially improve the results in Figure 10, but potentially be detrimental in the other situations?

Line 515-517: It is not clear that this is what the data indicate, since if it is involved in the formation of the rootlet, then one would hypothesize that rootlets may be missing in knockout animals. Data seemed to indicate instead that they are involved in the stability and/or maintenance of the rootlet from figure 8.

Minor:

In the legend for Figure 5A, it should be stated that the experiment was performed at least 3 times with similar results, or something to that effect.

Line 280: I like how the statement conveys "these data suggest," but my impression is that residues 1-170 actually has the actin binding ability as well based on the data in 5A, D, and G. Affinity may be reduced, but still in the sub-micromolar range.

Line 449-451: There was a peak current decline in the N259/N259 mice, so in those mice is the row stability due to reduced peak current? Please clarify the rationale.

Line 598: please reference figure 4C if that is the appropriate reference.

Line 606-607: Data from Kitajiri et al, suggests TrioBP bundling with an interfilament distance of about 8nm, which is similar or less than what Tprn appears to have in figure 3G.

Line 1840: The fluid jet does not deliver a step-like deflection. The term deflection implies displacement. Your reference to Mecca et al., PNAS, 2022; or Corns et al., PNAS, 2014 as well as Tobin et al. 2019, all show that the bundle displacement is not step-like and there are continued motions with a fluid jet. The hair bundle is not a simple spring, so a force stimulus of the hair bundle is not equivalent to a displacement stimulus. The displacement response is not likely a square step like the voltage command in your setup unless you can show it with displacement measurements.

Line 1876: Please indicate the ages of the WT that were combined.

Reviewer #3 (Comments to the Authors (Required)):

The study "Taperin bundles F-actin at stereocilia pivot-points enabling optimal lifelong mechanosensitivity" by Belyantseva* and Liu* et al 2024 investigates the role of taperin (TPRN) in stereocilia structure and its contribution to hearing in mice. Upon first submission the authors performed an incredible amount of work and should be further commended on their efforts in this improved submission. I very much appreciate the hardships and huge efforts that were undertaken to address the issue of the tagged/untagged taperin. I believe the revised methods and new figure that now show that the untagged version is less stable can help readers understand why this experiment was performed as it was and also will be helpful to inform future taperin curious people of necessary protein handling instructions. I also appreciate that the authors now include some comments in the discussion about the potential of taperin to form condensates (and why that might be interesting in this context) and the improvements to quantification and image quality throughout this revised work. I look forward to this group's future plans to explore bundle orientation and agree that it is beyond the scope of this current work. The authors successfully addressed all my concerns with very reasonable responses.

May 3, 2025

Pekka Lappalainen, PhD
Monitoring Editor
Journal of Cell Biology

Dan Simon, PhD
Scientific Editor
Journal of Cell Biology

Dear Drs. Lappalainen and Simon,

Thank you for conducting a thorough assessment of our revised #202408026R manuscript entitled "Taperin bundles F-actin at stereocilia pivot-points enabling optimal lifelong mechanosensitivity" by three expert reviewers. Also, thank you for provisionally accepting our study as an article in the Journal of Cell Biology.

We revised the manuscript and addressed all additional comments and requests made by the Editor and Reviewer #2.

Editorial comments:

1A. "Reviewer #2 asks for text changes to tone down a few claims and more clearly acknowledge discrepancies and caveats."

Response: We modified the manuscript by toning down some conclusions as suggested by the reviewer.

1B. "The reviewer also notes the data does not show that TRIOBP-4, TRIOBP-5, ANKRD24, & F-actin colocalize but rather that these proteins have similar signal patterns in different hair bundles. If it is possible we encourage you to add a final experiment showing co-localization of these proteins. But if this not feasible or technically challenging then please revise statements about these co-localizations in the text."

Response: We performed some additional STED experiments as suggested by the editor and reviewer #2. To clarify, we did not claim or use the word "co-localization" for these three proteins. The title of the relevant section of the manuscript is "**Partially overlapping localizations** of TPRN, TRIOBP, and ANKRD24...". Our data do not indicate a perfect co-localization but rather concentric and partially overlapping rings similar to that reported by us in Krey et al., 2022 for TRIOBP-5 and ANKRD24 localization. We now provide additional new STED images of rootlet F-actin encircled by TRIOBP-4 or TRIOBP-5 or ANKRD24 (new inserts in Fig 1, H-J) in response to the reviewer statement that "The conclusion that the F-actin filaments are at the center of the "ring-like" structure of TRIOBP-4 is not fully supported in this imaging...". Unfortunately, all currently available antibodies to these proteins are rabbit polyclonal and cannot be combined for STED co-localization. The goat ANKRD24 antibody (Santa Cruz) used in this study was discontinued, and we are out of this antibody but are in the process of requesting from Covance (LabCorp) custom production of this antibody should we be asked by investigators to provide some.

We revised the text to highlight the fact that our data shows localization of each of these proteins around the rootlets separately to each other: “Super-resolution STED imaging revealed unreported ring patterns of TRIOBP-4 in cross-sections at the stereocilia pivot points of the outer hair cells (OHCs) (Fig. 1, G-H), consistent with our model postulating that TRIOBP-4 “wraps around” actin filaments along the length of the rootlets (Katsuno et al., 2019; Kitajiri et al., 2010). TRIOBP-5 and ANKRD24 also form rings around the rootlets similar to our reported labeling in IHCs (Krey et al., 2022). TRIOBP-5 and TRIOBP-4 rings were similar in diameter (Fig. 1, I and O-P), while ANKRD24 diameter is larger than the TRIOBP-5 diameter (Fig. 1, J and O-P). TPRN also forms rings around the rootlets at stereocilia bases of both OHCs and IHCs (Fig. 1, K-N and Q). However, TPRN localization extends throughout the entire taper region of stereocilia forming a funnel-shaped structure (Fig. 1, L, arrowheads) enveloping the F-actin core (Fig. 1, L, R). We hypothesize that TPRN interacts with F-actin and influences localization and/or functions of other proteins in its immediate proximity such as ANKRD24 and TRIOBP-5.”

2. “Text limits: Character count for Articles is <40,000, not including spaces. Count includes title page, abstract, introduction, results, discussion, and acknowledgments. Count does not include materials and methods, figure legends, references, tables, or supplemental legends.”

Response: After incorporating the requested additional experimental data and addressing the extensive reviewer comments, the current version of the manuscript is approximately 45,000 characters. In an email dated April 01, 2025, Dr. Lindsey Hollander kindly acknowledged that this length is acceptable, considering the challenges associated with further reducing the length of the manuscript.

3. “Figure formatting: Articles may have up to 10 main text figures. Scale bars must be present on all microscope images, including inset magnifications. Molecular weight or nucleic acid size markers must be included on all gel electrophoresis. Please add scale bars to Figure 1D/N and the inset magnifications in S3.”

Response: Done. Scale bars were added to Fig 1, panel D and N and in the new inserts in panels H-J and indicated in the Fig 1 legend. For better clarity we added the scale bars to the insert in Fig S3 and all the panels.

4. “Also, please avoid pairing red and green for images and graphs to ensure legibility for colorblind readers. If red and green are paired for images, please ensure that the particular red and green hues used in micrographs are distinctive with any of the colorblind types. If not, please modify colors accordingly or provide separate images of the individual channels.”

Response: We changed the red color for TRIOBP-5 to orange and the green color for TRIOBP-4 to brown in O and P panels of Fig 1 to avoid the red-green combination. We changed the magenta color of TPRN fluorescence in panel Q to the green to be consistent with the color for TPRN in panels E' and R (graphs on the right).

5. “Statistical analysis: Error bars on graphic representations of numerical data must be clearly described in the figure legend. The number of independent data points (n) represented in a graph must be indicated in the legend. Please, indicate whether 'n' refers to technical or biological replicates (i.e. number of analyzed cells, samples or animals, number of independent experiments). If independent experiments with multiple biological replicates have been performed, we recommend using distribution-reproducibility SuperPlots (please see Lord et al., JCB 2020) to better display the distribution of the entire dataset, and report statistics (such as

means, error bars, and P values) that address the reproducibility of the findings.”

Response: We added the requested “n” with an explanation for each “n” in the Fig. 2 and Figure 7 legends and corrected a typo in Fig. 7A image labels and corresponding text in the legend. We also replaced panel M of Figure 2 with better quality images showing similar nuclear localization of TPRN²⁶⁰⁻⁷⁴⁹ but now counterstained the nuclei with DAPI.

6. “Statistical methods should be explained in full in the materials and methods. For figures presenting pooled data the statistical measure should be defined in the figure legends. Please also be sure to indicate the statistical tests used in each of your experiments (both in the figure legend itself and in a separate methods section) as well as the parameters of the test (for example, if you ran a t-test, please indicate if it was one- or two-sided, etc.). Also, if you used parametric tests, please indicate if the data distribution was tested for normality (and if so, how). If not, you must state something to the effect that “Data distribution was assumed to be normal but this was not formally tested.”

Response: Information about the two-sided t-test was added to Figure 8 legend.

7. “Materials and methods: Should be comprehensive and not simply reference a previous publication for details on how an experiment was performed. Please provide full descriptions (at least in brief) in the text for readers who may not have access to referenced manuscripts. The text should not refer to methods “...as previously described.” Please also indicate the acquisition and quantification methods for immunoblotting/western blots.

Response: The requested information was added to the Materials and methods. The acquisition and quantification information is also added to the Co-immunoprecipitation section on page 36.

8. “For all cell lines, vectors, constructs/cDNAs, etc. - all genetic material: please include database / vendor ID (e.g. Addgene, ATCC, etc.) or if unavailable, please briefly describe their basic genetic features, even if described in other published work or gifted to you by other investigators (and provide references where appropriate). Please be sure to provide the sequences for all of your oligos: primers, si/shRNA, RNAi, gRNAs, etc. in the materials and methods. You must also indicate in the methods the source, species, and catalog numbers/vendor identifiers (where appropriate) for all of your antibodies, including secondary. If antibodies are not commercial, please add a reference citation if possible. Please double check the list of antibodies, it does not include an anti-TRIOBP-4 antibody currently.”

Response: We thank the reviewer and the editor for pointing out this omission. Our TRIOBP4/5 antibody was custom developed against a common sequence of both TRIOBP-4 and TRIOBP-5 splice isoforms (evaluated in Kitajiri et al., 2010) and reported that it predominantly recognizes TRIOBP-4. This antibody and a reference were added to the Materials and methods section.

9. “Microscope image acquisition: The following information must be provided about the acquisition and processing of images:

- a. Make and model of microscope
- b. Type, magnification, and numerical aperture of the objective lenses
- c. Temperature
- d. Imaging medium
- e. Fluorochromes
- f. Camera make and model

- g. Acquisition software
h. Any software used for image processing subsequent to data acquisition. Please include details and types of operations involved (e.g., type of deconvolution, 3D reconstitutions, surface or volume rendering, gamma adjustments, etc.).”

Response: The requested information above was added to the Methods.

10. “References: There is no limit to the number of references cited in a manuscript. References should be cited parenthetically in the text by author and year of publication. Abbreviate the names of journals according to PubMed.”

Response: All references conform to JCB formatting.

11. “Supplemental materials: Articles may have up to 5 supplemental figures and 10 videos. Please also note that tables, like figures, should be provided as individual, editable files. A summary of all supplemental material should appear at the end of the Materials and methods section. Please include one brief sentence per item. In Figure S1 panels J-L are very small and may be difficult to read, please consider enlarging these.”

Response: Done.

12. “Video legends: Should describe what is being shown, the cell type or tissue being viewed (including relevant cell treatments, concentration and duration, or transfection), the imaging method (e.g., time-lapse epifluorescence microscopy), what each color represents, how often frames were collected, the frames/second display rate, and the number of any figure that has related video stills or images.”

Response: Done.

13. eTOC summary: A ~40-50 word summary that describes the context and significance of the findings for a general readership should be included on the title page. The statement should be written in the present tense and refer to the work in the third person. It should begin with "First author name(s) et al..." to match our preferred style.

Response: The eTOC summary was modified and reads:

Belyantseva and Liu report that taperin encircles and stabilizes F-actin at auditory hair cell stereocilia bases and bundles F-actin. Taperin deficiency causes stereocilia disassembly, rootlet breakage at pivots and other abnormalities culminating in deafness. Thus, taperin bundles F-actin to maintain rootlet integrity and lifelong hearing.

14. “Conflict of interest statement: JCB requires inclusion of a statement in the acknowledgements regarding competing financial interests. If no competing financial interests exist, please include the following statement: "The authors declare no competing financial interests." If competing interests are declared, please follow your statement of these competing interests with the following statement: "The authors declare no further competing financial interests.”

Response: Done.

15. “A separate author contribution section is required following the Acknowledgments in all research manuscripts. All authors should be mentioned and designated by their first and middle

initials and full surnames. We encourage use of the CRediT nomenclature (<https://casrai.org/credit/>).”

Response: The author contributions section was added after the Acknowledgements according to CRediT nomenclature. We also add Dr. Erich Boger and Mr. Spencer Goodman to the acknowledgement to thank them for technical assistance.

16. “ORCID IDs: ORCID IDs are unique identifiers allowing researchers to create a record of their various scholarly contributions in a single place. Please note that ORCID IDs are required for all authors. At resubmission of your final files, please be sure to provide your ORCID ID and those of all co-authors.”

Response: ORCID IDs for all co-authors are provided after the author contribution section.

17. “JCB requires authors to submit Source Data used to generate figures containing gels and Western blots with all revised manuscripts. This Source Data consists of fully uncropped and unprocessed images for each gel/blot displayed in the main and supplemental figures. For assays performed using capillary electrophoresis and/or immunoassay-based detection, authors should instead provide the electropherogram graph(s) for each experiment, plotting fluorescence/chemiluminescence intensity vs. molecular weight/size. Since your paper includes cropped gel and/or blot images, please be sure to provide one Source Data file for each figure gels, blots, and/or capillary electrophoresis assays along with your revised manuscript files. File names for Source Data figures should be alphanumeric without any spaces or special characters (i.e., SourceDataF#, where F# refers to the associated main figure number or SourceDataFS# for those associated with Supplementary figures). For traditional gels and blots, the lanes of the gels/blots should be labeled as they are in the associated figure, the place where cropping was applied should be marked (with a box), and molecular weight/size standards should be labeled wherever possible. For capillary electrophoresis assays, each trace in the graph should be color-coded and labeled to indicate which protein, gene, or sample is being measured (please try to avoid red/green combinations to accommodate our color-blind readers). Source Data files will be directly linked to specific figures in the published article. Source Data Figures should be provided as individual PDF files (one file per figure). Authors should endeavor to retain a minimum resolution of 300 dpi or pixels per inch. Please review our instructions for export from Photoshop, Illustrator, and PowerPoint here: <https://rupress.org/jcb/pages/submission-guidelines#revised>”

Response: Source data files are provided according to the instructions.

18. “Journal of Cell Biology now requires a data availability statement for all research article submissions. These statements will be published in the article directly above the Acknowledgments. The statement should address all data underlying the research presented in the manuscript. Please visit the JCB instructions for authors for guidelines and examples of statements at (<https://rupress.org/jcb/pages/editorial-policies#data-availability-statement>).”

Response: The data availability statement is located directly above Acknowledgments.

19. “Please upload the following materials to our online submission system. These items are required prior to acceptance. If you have any questions, contact JCB’s Managing Editor, Lindsey Hollander (lhollander@rockefeller.edu).”

-- A response to reviewer comments document (PDF is acceptable).

Response: We provided all the above-mentioned information and files in the requested resolution and format.

Reviewer #1 (Comments to the Authors (Required)):

20. "The authors did a nice job addressing my concerns and suggestions with edits to narrative, additional explanation, reorganization of figure panels, and in several places, new data. The new FIB-SEM images are stunning. Overall, I view this as a strong revision, and I have no further suggestions for improving the work."

Response: We are very grateful to the reviewer for acknowledging our effort.

Reviewer #2 (Comments to the Authors (Required)):

21. "In the revised manuscript, Belyantseva et al. have made some admirable improvements to the manuscript. The added control in Figure 3 is a strong addition, the new data regarding Tprn -/- in the figure 10 is excellent, and the increased quantification adds clearer support of the data presented. The authors provided many great improvements and additions, however, my impression is that the overall conclusion of the manuscript is still not following all the data presented, likely due to what appears to be over interpretation of the data in some cases. The authors are proposing the idea that Taperin is required for the proper formation of the rootlet in development, but no data are provided that at the EM level for rootlets when they are initially formed (~P7 based on Kitajiri et al., 2010). I do appreciate that the authors are careful in stating that much of the ideas come from their in vitro work. However, the data rather appear to support the other part of their hypothesis/conclusion that taperin is required for the proper maintenance of rootlets and for stability of the rootlets in vivo. Even at the age of P17 for N259 IHC (figure 10A), the rootlet is normally formed at the upper portion and traverses the narrowing of the stereocilia pivots. All the abnormalities occur at the pivot point (cracks) and lower rootlet, suggesting other functions besides bringing the rootlet down from the espin cross-linked inter-filament distance to the rootlet filament density. These data all suggest that the authors second conclusion of "durable deflections of stereocilia by sound throughout a lifetime" is more fitting. Overall, the data are strong, but the interpretation and the conclusions drawn are often claimed as definitive, when there are discrepancies and irregularities. The text needs to be carefully edited to not overstate what can be concluded from the data."

Response: We thank the reviewer for acknowledging the improvements in our manuscript. The Reviewer raised an interesting question about interpretation of our results, specifically, whether TPRN deficiency results in abnormal rootlet development as we proposed in the manuscript or increased damage to the rootlets as argued by the Reviewer. To clarify our point of view and acknowledge the reviewer's alternative interpretation, we revised the first paragraph of the Discussion as follows:

“The effects of TPRN deficiency on stereocilia bundles could be seen as early as P1-P7 (Fig. 6, C-D; Fig. 8, B-J; Fig. S5, G-J’), which argues in favor of abnormal development of the rootlets rather than an accelerated damage, although, the latter supposition cannot be excluded as we observed increased F-actin breakages at P17 *Tprn*^{N259/N259} and at P32 in *Tprn*^{-/-} stereocilia (Fig. 10B).”

Major:

22. “The text is still a bit overinterpreted in my opinion. Line 149: This summary is not quite a demonstration, since stereocilia structure is not necessarily stabilized by taperin when overexpression also causes stereocilia instability. How was it shown that the stereocilia changes are not secondary to other dysfunction that causes homeostatic changes in the stereocilia structure? Optimizing mechanosensitivity and sound transduction is also not demonstrated, since *Tprn*^{-/-} mice show no changes in mechanosensitivity. How was it shown that sound transduction is optimized for “sustained sound-induced stimuli?” Some of these are not really supported by the data and others are hypotheses of the functional model, not a “demonstration.”

Response: We agree and modified the text on page 6:

“Based on these data, we propose a model, which assumes that F-actin bundling by TPRN at stereocilia pivot points brings actin filaments together to reinforce rootlets, stabilizing stereocilia structure and optimizing mechanosensitivity and sound transduction for sustained sound-induced stimuli”.

23. “Figure 1G-H: In the rebuttal, the authors make a point about the TRIOBP-4 staining being important and new, but the images presented here are not even co-localized with each other and are of different hair bundles. The conclusion that the F-actin filaments are at the center of the “ring-like” structure of TRIOBP-4 is not fully supported in this imaging since there is no co-localization. In terms of referring to G-J and O-P, it should be stated that the assumption is that these are rings forming around the F-actin rootlet, but there is currently no data to confirm the strict co-localization/structure of TRIOBP 4 and 5 with each other or with the F-actin and ANKRD24. The only data showing the co-staining appears to be with TPRN and F-actin.”

Response: At the resolution of a confocal light microscope, we reported co-localization of F-actin with TRIOBP-4 (Kitajiri et al., 2010, Cell; Katsuno et al., 2019, JCI Insight), TRIOBP-5 (Kitajiri et al., 2010, Cell), and ANKRD24 (Krey et al., 2022, JCB) in the lower rootlets of stereocilia. It may seem redundant to reproduce the results in the current study, which refined the localization patterns of each of three proteins using super-resolution STED microscopy in cross-section of OHC rootlets. We agree with the Reviewer that it would be nice to refine the localization of these proteins relative to each other with STED microscopy. However, it would require three-color (due to including F-actin labeling) STED imaging with the same depletion laser (to avoid chromatic shifts inevitable at this resolution), which is extremely challenging. In addition, all three antibodies are rabbit polyclonal and cannot be combined. Unfortunately, goat ANKRD24 antibody from Santa Cruz that we used here and in Krey et al, 2022 are discontinued and we do not have any aliquots left. However, to confirm the “assumption that the rings forming around the F-actin rootlet” as stated by reviewer #2, we now include the two-color images as inserts showing F-actin of rootlets with each corresponding protein (TRIOBP4, TRIOBP5 and ANKRD24). See inserts in Figure 1 panels H, I, J.

We revised a few statements in the first paragraph of the Results to highlight the fact that our data show individual localization of each of these proteins around the rootlets: “Super-resolution STED imaging revealed previously unreported ring patterns of TRIOBP-4 in cross-sections of the lower rootlet of outer hair cells (OHCs) (Fig. 1, G-H), consistent with our model postulating that TRIOBP-4 “wraps around” actin filaments along the length of the rootlets (Katsuno et al., 2019; Kitajiri et al., 2010). TRIOBP-5 and ANKRD24 also form rings around the rootlets similar to our reported labeling in IHCs (Krey et al., 2022). TRIOBP-5 and TRIOBP-4 rings were similar in diameter (Fig. 1, I and O-P), while ANKRD24 diameter is larger than the TRIOBP-5 diameter (Fig.1, J and O-P). TPRN also forms rings around the rootlets at stereocilia bases of both OHCs and IHCs (Fig. 1, K-N and Q). TPRN also forms rings around the rootlets at stereocilia bases of both OHCs and IHCs (Fig. 1, K-N and Q). However, TPRN localization extends throughout the entire taper region of stereocilia forming a funnel-shaped structure (Fig. 1, L, arrowheads) enveloping the F-actin core (Fig. 1, L, R). We hypothesize that TPRN interacts with F-actin and influence localization and/or function of other proteins in its immediate proximity such as ANKRD24 and TRIOBP-5.”

24. “Figure 2: It may actually be more helpful to state the numbers of cells exhibiting each of the displayed morphologies shown as well. The conclusions drawn from this figure (line 204-206) still appear to be an over-interpretation, since the data themselves about the 1-260 residues being sufficient for actin bundling do not observe that same behavior in the AAV infected hair cells in figure 8. The statement regarding targeting sequence between 260 and 730 is based on constructs 1-260 and 730-749 not having specific rootlet targeting, but a “targeting sequence” could exist straddling those constructs and the 260-730aa region or could require the first 260 aa and the last 19 aa connected by some linker. Additionally, full length expression of TPRN does not only localize the protein at the taper region, overexpressed FL TPRN also localized elsewhere, so one could argue that the 1-260 aa construct also targets the protein to the taper, but the overexpression causes localization to other parts of the stereocilia as well. The data more aptly “suggests” rather than “concludes.”

Response: We now indicate the number of hair cells with each displayed morphological abnormalities in the Figure 2 legend and replaced “We conclude” with “These data suggest” softening the statement on page 9 as requested by the reviewer.

25. Figure 3: “The addition of the quantification and control make these experiments much more complete. It is interesting that the distance between filaments is similar in the TRX only as compared to the TPRN. The images in Panel E make it look as if filaments are further apart in TRX only and closer together in TPRN expressed. I wonder if the separation of the filaments is more difficult when they are closer together.”

Response: We agree that resolving inter-filament distances is more difficult when filaments are close together and form three-dimensional bundles. We clearly acknowledged this potential drawback and describe our measurement procedure details including the limitations in the Methods.

26a. Figure 9: “The increase in Popen is exactly the bias that would be expected. As calcium is a key regulator of MET, a reduction in the peak current reduces the resting current and hence calcium in the stereocilia, where the cell compensates by increasing the open probability to increase the intracellular calcium concentration (like a homeostasis).

Response: To our knowledge, there are no published reports demonstrating this “homeostasis” effect. Popen increase is not observed during partial blockage of MET channels by

tubocurarine. There is no precedent in cell biology for actively maintaining micromolar concentrations of free Ca^{2+} like the ones expected at the tips of stereocilia – usually, the cells maintain significantly lower concentrations of free Ca^{2+} around 100 nM.

26b. While I can understand the value in including all the data when indicating the peak current, the variability in the number of intact tip-links makes it difficult to assess whether any other changes are actually occurring. So analysis of other aspects need to account for the peak current variability. For instance, the stiffness of the hair bundle is affected by the number of intact tip-link. When breaking all tip-links with a calcium chelator, the stiffness of the hair bundle is significantly reduced (see Tobin et al., 2019). When comparing stiffness across hair bundles, it would have been better to break all tip-links such that there is a cleaner measure of the stiffness of the stereocilia pivots without the variability of the number of tip links contributing. There is also still a remaining issue with the positioning of the fluid jet in relation to the hair bundle. The distance from the hair bundle ranges from 3.6-5um, but from Tobin et al. (2019) a 1.4 fold change in the distance results in a 30% difference in the force delivered, contributing more uncontrolled variability in the data.

Response: We do agree with the reviewer that extended study is needed to reveal potential subtle effects of TPRN deficiency on mechanical properties of the hair bundles. This is specifically acknowledged in our manuscript on page 19: “Our data suggest that any potential effects of TPRN-mediated F-actin crosslinking at stereocilia pivot points on mechanical stiffness of hair bundles may be too subtle to be detected by a fluid-jet technique or not present in young postnatal hair cells before the onset of hearing.”

26c. If improvements to the quality of the experiments is not done, it would be better to move this data to the supplement since there are clear caveats in the interpretation of the data as presented.”

Response: Since we reached the limit of Supplementary figures allowed by JCB, we decided to keep this figure in the main text as agreed upon by the editor. We believe that the data provides essential information for interpreting the results – the fact that TPRN deficiency affects stereocilia bundles already at an age when there is not yet significant loss of MET current.

27. “Figure 10: It is interesting that the phenotype looks more severe for the *Tprn*^{-/-} IHCs, but the ABR at P18 suggests N259 is more severe, the MET measurements suggest that the N259 is more severe as well, and the SEM data in figure 6 suggests that N259 is more severe since 6E is of P30 *Tprn*^{-/-} and it is comparable to 6F of P17 N259. A more thorough discussion of how these phenotypes may differ based on the potential function of the 1-259 version of taperin may be helpful to put all of these results in context. How can that expression potentially improve the results in Figure 10, but potentially be detrimental in the other situations?”

Response: We refrained from judging these phenotypes as more or less severe based solely on FIB-SEM images. Some abnormalities are more prominent in *Tprn*^{-/-} IHCs (F-actin breaks) but the other ones are more prominent in *Tprn*^{N259/N259} IHCs (kinked rootlets). Additionally, “branched” rootlets in *Tprn*^{-/-} IHCs may be more stable than “kinked” rootlets in *Tprn*^{N259/N259} IHCs. We agree that these are important considerations but too speculative to mention in the text.

28. “Line 515-517: It is not clear that this is what the data indicate, since if it is involved in the formation of the rootlet, then one would hypothesize that rootlets may be missing in knockout animals. Data seemed to indicate instead that they are involved in the stability and/or

maintenance of the rootlet from figure 8.”

Response: We respectfully question the logic of the statement that if protein “X” is involved in formation of rootlets, then rootlets may be missing in “X”-knockout animals. There are many cases where some protein can play a critical role in the formation of a subcellular organelle without being absolutely required for the initial appearance of the structure. For example, myosin XVa that is clearly involved in the development of wild type stereocilia. Yet, in the absence of myosin XVa, stereocilia do form, albeit they are abnormally short. Similarly, rootlets are present in the absence of TPRN, but they appear structurally abnormal.

Minor:

29. “In the legend for Figure 5A, it should be stated that the experiment was performed at least 3 times with similar results, or something to that effect.”

Response: Thank you for pointing out this omission. The relevant statement is now included in Figure 5A legend.

30. “Line 280: I like how the statement conveys “these data suggest,” but my impression is that residues 1-170 actually has the actin binding ability as well based on the data in 5A, D, and G. Affinity may be reduced, but still in the sub-micromolar range.”

Response: We agree. We state “TRX-TPRN¹⁻³⁰⁰ and TRX-TPRN¹⁻¹⁷⁰ effectively bundle F-actin; nevertheless, the rightward shift of the curves indicates that the crosslinking activity of purified TPRN is reduced as the length of TPRN is decreased (Fig. 5, G, H).” Experiments are needed to determine the exact role of the first 170 residues of mouse TPRN. This question will be explored in a future project.

31. “Line 449-451: There was a peak current decline in the N259/N259 mice, so in those mice is the row stability due to reduced peak current? Please clarify the rationale.”

Response: We have modified the text to clarify that our conclusions can be drawn only from the data obtained from knockout mice lacking TPRN without potential confounding effects of the decreased MET current in *Tprn*^{N259/N259} mice. The modified text on page 21 reads:

“The loss of F-actin bundling by TPRN in mice lacking TPRN also results in a gradual disappearance of TRIOBP-5 and ANKRD24 and retraction of the shortest stereocilia rows during postnatal development (Fig. 8, A-G’ and Fig. S5, G-J’, correspondingly). All these effects of a TPRN deficiency in postnatal hair cells before the onset of hearing are MET independent, e.g. occur without significant loss of MET current in *Tprn*^{-/-} hair cells (Fig. 9).”

32. “Line 598: please reference figure 4C if that is the appropriate reference.”

Response: Done. The sentence in the first paragraph of the “Novel function of TPRN as an F-actin bundling protein at stereocilia pivot points” subsection of the Discussion now reads: “When TPRN is divided into four consecutive fragments, each of them binds to actin indicating multiple actin-binding regions (Fig. 4, C).”

33. "Line 606-607: Data from Kitajiri et al, suggests TrioBP bundling with an interfilament distance of about 8nm, which is similar or less than what Tprn appears to have in figure 3G."

Response: Yes, interfilamentous distance between F-actin bundled by TRX-TPRN is similar or slightly larger than reported for TRIOBP-4 actin-bundling by us in Kitajiri et al, 2010. We do not have any *a priori* reasons to believe that the distance between actin filaments bundled by TPRN should be the same or different from the distance between the filaments bundled by TRIOBP-4. Even if this distance is the same, there are still plenty of possibilities for different types of bundling, e.g. a similar type of bundling but less bundling points per micrometer of F-actin filament.

34. Line 1840: The fluid jet does not deliver a step-like deflection. The term deflection implies displacement. Your reference to Mecca et al., PNAS, 2022; or Corns et al., PNAS, 2014 as well as Tobin et al. 2019, all show that the bundle displacement is not step-like and there are continued motions with a fluid jet. The hair bundle is not a simple spring, so a force stimulus of the hair bundle is not equivalent to a displacement stimulus. The displacement response is not likely a square step like the voltage command in your setup unless you can show it with displacement measurements.

Response: The figure legend has been modified as follows: "Representative MET currents to fluid-jet stimuli evoked by step-like voltage commands..."

35. Line 1876: Please indicate the ages of the WT that were combined.

Response: Done. The sentence now reads: "Since no statistically significant differences were found between P17 and P34 for the wild type, the data for wild type were combined".

Reviewer #3 (Comments to the Authors (Required)):

The study "Taperin bundles F-actin at stereocilia pivot-points enabling optimal lifelong mechanosensitivity" by Belyantseva* and Liu* et al 2024 investigates the role of taperin (TPRN) in stereocilia structure and its contribution to hearing in mice. Upon first submission the authors performed an incredible amount of work and should be further commended on their efforts in this improved submission. I very much appreciate the hardships and huge efforts that were undertaken to address the issue of the tagged/untagged taperin. I believe the revised methods and new figure that now show that the untagged version is less stable can help readers understand why this experiment was performed as it was and also will be helpful to inform future taperin curious people of necessary protein handling instructions. I also appreciate that the authors now include some comments in the discussion about the potential of taperin to form condensates (and why that might be interesting in this context) and the improvements to quantification and image quality throughout this revised work. I look forward to this group's future plans to explore bundle orientation and agree that it is beyond the scope of this current work. The authors successfully addressed all my concerns with very reasonable responses.

Response: We are grateful for the complimentary comments.